# An optogenetic approach for regulating human parathyroid hormone secretion

Yunhui Liu [1,2,8], Lu Zhang [1,3,8], Nan Hu [4,8], Jie Shao[1,2], Dazhi Yang[5], Changshun Ruan [6], Shishu Huang[7], Liping Wang [1], William W. Lu [3], Xinzhou Zhang [4✉] & Fan Yang [1,2✉]

Parathyroid hormone (PTH) plays crucial role in maintaining calcium and phosphorus homeostasis. In the progression of secondary hyperparathyroidism (SHPT), expression of calcium-sensing receptors (CaSR) in the parathyroid gland decreases, which leads to persistent hypersecretion of PTH. How to precisely manipulate PTH secretion in parathyroid tissue and underlying molecular mechanism is not clear. Here, we establish an optogenetic approach that bypasses CaSR to inhibit PTH secretion in human hyperplastic parathyroid cells. We found that optogenetic stimulation elevates intracellular calcium, inhibits both PTH synthesis and secretion in human parathyroid cells. Long-term pulsatile PTH secretion induced by light stimulation prevented hyperplastic parathyroid tissue-induced bone loss by influencing the bone remodeling in mice. The effects are mediated by light stimulation of opsin expressing parathyroid cells and other type of cells in parathyroid tissue. Our study provides a strategy to regulate release of PTH and associated bone loss of SHPT through an optogenetic approach.

[1] The Brain Cognition and Brain Disease Institute (BCBDI), Shenzhen Institute of Advanced Technology, Chinese Academy of Sciences (CAS), Shenzhen-Hong Kong Institute of Brain Science-Shenzhen Fundamental Research Institutions, Shenzhen, China. [2] University of Chinese Academy of Sciences, Beijing, China. [3] Department of Orthopaedics and Traumatology, The University of Hong Kong, Hong Kong SAR, China. [4] Department of Nephrology and Shenzhen Key Laboratory of Kidney Diseases, Shenzhen People's Hospital, The Second Clinical Medical College of Jinan University, Shenzhen, China. [5] Department of Orthopedics, Union Shenzhen Hospital, Huazhong University of Science and Technology, Shenzhen, China. [6] Research Center for Human Tissue and Organs Degeneration, Institute of Biomedicine and Biotechnology, Shenzhen Institute of Advanced Technology, Chinese Academy of Sciences (CAS), Shenzhen, China. [7] Department of Orthopaedic Surgery and Orthopaedic Research Center, West China Hospital of Sichuan University, Chengdu, China. [8] These authors contributed equally: Yunhui Liu, Lu Zhang, Nan Hu. ✉email: xinzhouzhang1946@163.com; fan.yang@siat.ac.cn

Parathyroid hormone (PTH) is the most important endocrine regulator involved in maintaining calcium levels in humans. The PTH peptide family include PTH and the PTH-related peptide (PTHrP), both of which play an important roles in maintaining homeostasis of calcium-phosphate and bone metabolism[1]. The parathyroid gland (PTG) secrets PTH to regulate serum calcium, which effects bone, kidney, and the intestines. Parathyroid disorder can cause excessive secretion of PTH and hyperparathyroidism, which either occurs within the parathyroid gland (primary) or outside the gland (secondary), resulting in elevated serum PTH levels[2]. The most common symptoms of hyperparathyroidism include fragile bones, joint pain, and kidney stones[3].

The regulatory secretion of PTH is determined by serum ionized calcium concentration through negative feedback loops. Calcium sensing receptors (CaSR) are expressed in parathyroid chief cells and control the rapid release of PTH[4]. High concentrations of calcium ions may activate CaSR in the parathyroid gland, induce intracellular calcium changes and inhibit the secretion of PTH[5]. In the progression of secondary hyperparathyroidism (SHPT), the expression of CaSR in the parathyroid gland decreases and thus limits the regulation of PTH secretion based on calcium levels[6]. In secondary hyperparathyroidism patients, CaSR and vitamin D receptor (VDR)-transcripts are both down-regulated in parathyroid tissue[7], and the lack of CaSR expression in the parathyroid gland of some patients also leads to the progression of hyperparathyroidism and resistance to cinacalcet[8,9]. In severe hyperparathyroidism that is refractory to medical treatment, total parathyroidectomy with forearm transplantation has been used to treat SHPT[10]. However, elevated PTH after curative parathyroidectomy still occurs and results in a delay in post-operative symptom improvement[11]. Although PTH secretion is partially restored following autotransplantation, the capacity of the transplanted tissue to adapt to serum calcium changes remain profoundly disturbed[12]. The current situation is that a method of precise inhibition of PTH secretion in both in situ and transplanted human PTG tissues, especially secretion based on serum calcium concentration, is lacking.

Optogenetics is a promising and powerful technology that allows reversible control of neuronal activity and cellular processes using light[13]. An optogenetic approach has successfully been used to control the activity of gonadotropin-releasing hormone (GnRH) neurons and the pulsatile secretion of luteinizing hormone[14]. The combined use of optogenetic perturbation and imaging offers new tools for precise control of rhythmic cardiac electrical activity[15]. Closed-loop automated optogenetic neuromodulation of bladder sensory afferents can normalize bladder function in a rat model[16]. Based on these findings, we hypothesized that optogenetic technology could be used as a potential method to regulate the secretion of parathyroid hormones. However, until now, the effects of optogenetic regulation of human parathyroid cells are largely unknown, as is the underlying molecular mechanism.

In this study, we used optogenetic tools to investigate the effects and mechanisms underlying the optical activation of human parathyroid cells. Our data show that parathyroid cells isolated from patients with SHPT displayed an impaired response to extracellular calcium. Light activation of the parathyroid inhibits human PTH secretion by depolarizing the membrane potential, elevating intracellular calcium, and modulating cell signaling pathways. Importantly, we also demonstrate that optogenetic activation of human parathyroid tissue can be automatically controlled using ionized calcium concentrations. The in vivo long-term inhibition of human PTH secretion can substantially enhance bone formation and repress bone reabsorption by influencing the bone remodeling process.

## Results

**Parathyroid cells from patients with secondary hyperparathyroidism do not respond to changes in extracellular calcium.** Parathyroid cells are able to detect serum ionized calcium concentration and regulate PTH secretion through negative feedback loops. To evaluate the calcium-responding capacity of parathyroid cells in secondary hyperparathyroidism (SHPT), we isolated and cultured parathyroid cells from patients with SHPT, then used electrophysiological recordings, calcium fluorescence and a PTH assay to assess the calcium-induced response, including membrane potential, intracellular $Ca^{2+}$ and PTH levels, respectively (Fig. 1a).

Parathyroid chief cells were isolated from eight patients with a diagnosis of SHPT (Table 1 and Supplementary Fig. 1a). Hematoxylin and Eosin (HE) staining of the hyperplastic parathyroid tissue showed typical histological features of secondary hyperparathyroidism: asymmetric enlargement, nodularity and heterogeneous distribution of chief cells (Fig. 1b). By the seventh day of cultivation, parathyroid cells appeared uniform and maintained epithelial-cell-like morphology (Supplementary Fig. 1b). Immunofluorescent staining of the cultured cells revealed that most of the isolated cells expressed parathyroid hormone (PTH), calcium-sensing receptors (CaSR) and Vitamin D receptors (VDR) (Fig. 1c), and there were few signals in the negative control group and the adjacent thyroid gland tissue (Supplementary Fig. 1c, d).

Membrane potential is a key factor in the cellular signaling involving PTH secretion[17]. We used patch-clamp techniques to record the membrane potential of parathyroid chief cells during an increase of extracellular $Ca^{2+}$ from 0.5 to 2.6 mM. In four recorded parathyroid cells (Cell 1 from patient1, Cell 2 from patient 2, Cell 3 and 4 from patient 3), only one (Cell 4) showed fluctuations of the membrane potential, whereas the other three cells displayed stable membrane potentials during the increase of extracellular $Ca^{2+}$ (Fig. 1d). Of 30 cells, only six were responsive to high extracellular calcium and showed fluctuations of the membrane potential; the other 24 cells showed less of a response to extracellular calcium with stable membrane potentials (Fig. 1e).

Intracellular calcium levels can be increased by CaSR through the release of $Ca^{2+}$ from the endoplasmic reticulum and the opening of plasma membrane calcium channels to inhibit PTH secretion[4]. We used the calcium indicator dye Fura-2-AM to evaluate intracellular $Ca^{2+}$ levels and measured the intensity of the fluorescence in parathyroid cells during an increase in extracellular $Ca^{2+}$ from 0.5 to 2.6 mM (Fig. 1f, g). Calcium imaging revealed relatively stable fluorescence over time during the increase of extracellular $Ca^{2+}$ from 0.5 to 2.6 mM in four parathyroid cells (Cell 1 from patient1, Cell 2 from patient 2, Cell 3 and 4 from patient 3; Fig. 1f), and quantification confirmed that there were no notable fluorescence fluctuations in any parathyroid cells (Fig. 1g). Statistics revealed that only five of 53 cells from eight patients were responsive to extracellular $Ca^{2+}$, showing fluctuations in fluorescence (Fig. 1h). We then quantified the level of parathyroid hormone (PTH) secreted from cultured human parathyroid cells at both low and high $Ca^{2+}$ concentrations, and we observed no difference in PTH levels between 2.6 mM $Ca^{2+}$ and 0.5 mM $Ca^{2+}$ (Fig. 1i).

In order to demonstrate the normal response to extracellular calcium, we also isolated and cultured normal rat parathyroid cells (Supplementary Fig. 2a). Patch-clamp recordings showed that the membrane potential of normal rat parathyroid chief cells fluctuated during an increase of extracellular $Ca^{2+}$ from 0.5 to 2.6 mM (Supplementary Fig. 2b). Of ten recorded cells, eight cells were responsive to high extracellular calcium and showed fluctuations of the membrane potential (Supplementary Fig. 2c). Calcium imaging and quantification showed that normal rat

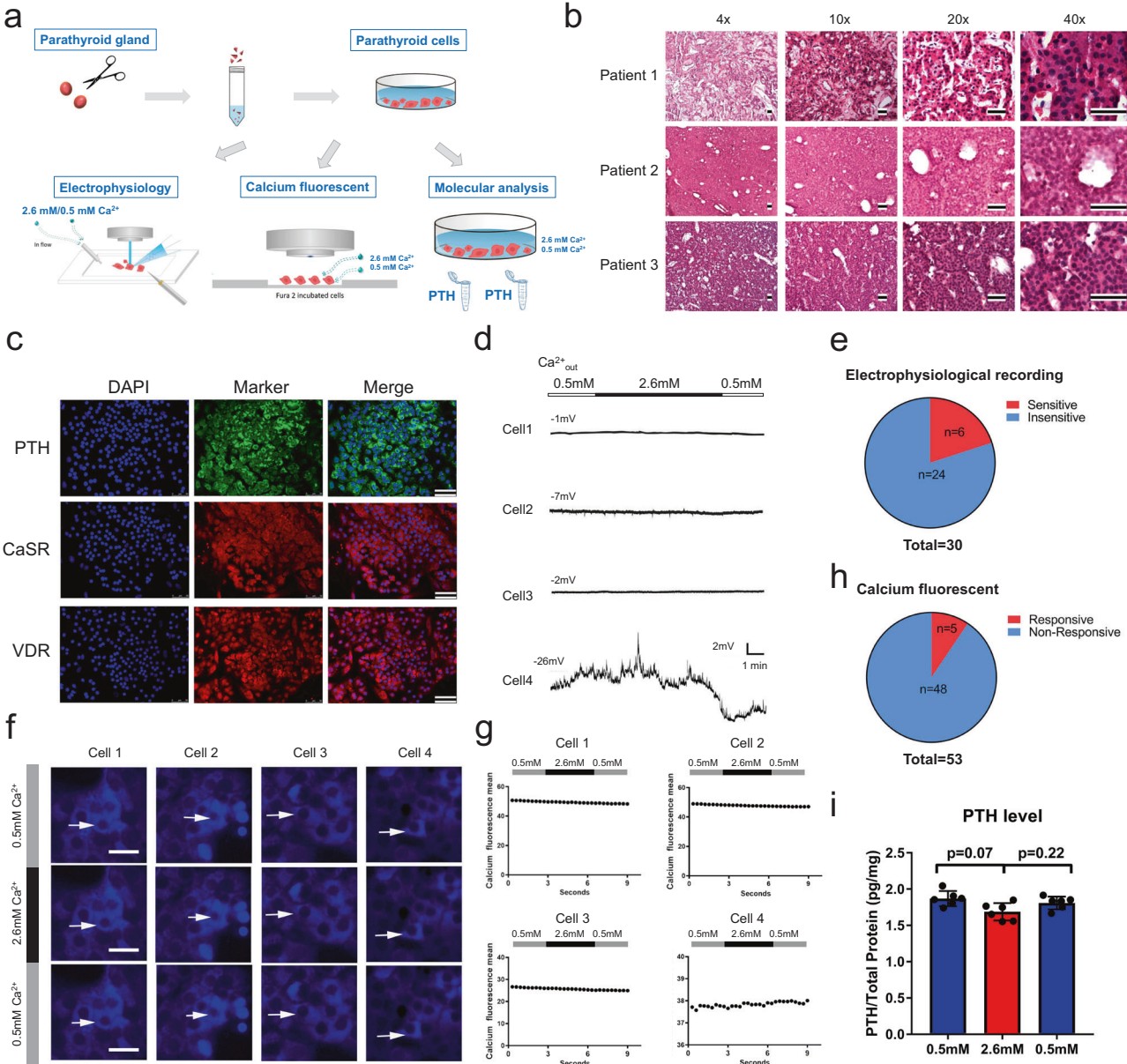

**Fig. 1 Parathyroid cells from patients with secondary hyperparathyroidism do not respond to changes in extracellular calcium. a** Schematic protocol showing the isolation and culture of the parathyroid cells from patients with secondary hyperparathyroidism (SHPT) and techniques used to assess the calcium-induced response of the parathyroid cells. **b** Hematoxylin & Eosin (HE) staining of representative parathyroid tissue from three patients with SHPT at different magnifications. Scale bar = 100 μm. **c** Immunostaining of parathyroid hormone (PTH), calcium sensing receptor (CaSR) and Vitamin D receptor (VDR) on the representative cultured human parathyroid cells. Scale bar = 50 μm. **d** Electrophysiological recordings of the membrane potential of four representative parathyroid chief cells during the change of extracellular $Ca^{2+}$ from 0.5 mM to 2.6 mM. **e** Summary of electrophysiological recording statistics from 30 parathyroid cells from eight patients. **f** Calcium fluorescence assay reflecting the change of the intracellular $Ca^{2+}$ concentrations in four parathyroid cells responding to extracellular $Ca^{2+}$. Scale bar = 20 μm. **g** Quantification of the calcium fluorescence intensity over time in the four representative parathyroid chief cells. **h** Only 5 of 53 cells from eight patients were sensitive to extracellular $Ca^{2+}$ and showed fluctuations of fluorescence during extracellular $Ca^{2+}$ changes. **i** Quantification of the level of PTH secreted from cultured human parathyroid cells at 0.5 mM extracellular $Ca^{2+}$ and 2.6 mM extracellular $Ca^{2+}$ respectively ($n = 6$ per group). Two-tailed unpaired $t$-test, $p = 0.07$, $p = 0.22$, respectively. Values represent mean ± SEM. Source data are provided as a Source Data file.

parathyroid cells were sensitive to extracellular $Ca^{2+}$ and showed fluorescence fluctuations during extracellular $Ca^{2+}$ changes (Supplementary Fig. 2e, f). Statistics revealed that 12 of 12 cells were responsive to extracellular $Ca^{2+}$, showing fluctuations in fluorescence (Supplementary Fig. 2d). Our data also demonstrate that increasing extracellular calcium leads to lower PTH levels in normal rat parathyroid cells (Supplementary Fig. 2g).

Taken together, these data consistently demonstrate that there was a decreased response of membrane potential and intracellular calcium in human parathyroid cells isolated from SHPT patients compared with normal parathyroid cells. The lack of response to extracellular calcium limited the ability of hyperplastic parathyroid cells to effectively regulate PTH secretion in response to calcium concentration.

**Table 1 Summary of patient information participating in this study.**

| No | PTH (pg/ml) | Serum calcium (mM) | Creatinine (μM) | Glomerular filtration rate (ml/min) |
|----|-------------|--------------------|-----------------|--------------------------------------|
| 1 | 775 | 1.91 | 740 | 5.19 |
| 2 | 578 | 2.52 | 934 | 3.50 |
| 3 | 517 | 2.42 | 1371.5 | 3.53 |
| 4 | 1477 | 2.15 | 862 | 4.91 |
| 5 | 1174 | 2.50 | 539.2 | 7.14 |
| 6 | 1200 | 2.62 | 1017 | 5.40 |
| 7 | 443 | 2.39 | 966.3 | 5.06 |
| 8 | 380 | 2.19 | 1234.2 | 3.39 |

Serum PTH level, serum calcium level and serum creatinine level and glomerular filtration rate in eight patients diagnosed with secondary hyperparathyroidism (SHPT). The age of the patients participating in the study was 26–68 years old, and the study consisted of three female and five male participants. All tissue specimens were collected from August 2017 to February 2019 with the written consent of secondary hyperparathyroidism patients and approved by Ethics Committee of Shenzhen People's Hospital.

**Light stimulation of the opsin ChETA effectively depolarizes membrane potential and inhibits PTH secretion from parathyroid cells.** We used ChETA, an engineered opsin gene which has specific advantages in optogenetic control over the ChR2 gene. These include a reduced level of undesirable extra spiking seen in ChR2 animals, and the ability to drive temporally-stationary spiking up to around 200 Hz[18]. We used lentivirus carrying the CMV-ChETA-eYFP construct to transfect cultured human parathyroid cells (Fig. 2a). Forty-eight hours after transfection, most chief cells were successfully labeled and expressed green fluorescence (Fig. 2b). We then used patch-clamp techniques to investigate the electrophysiological properties of ChETA-expressing human parathyroid cells (Fig. 2c). Stimulating ChETA-expressing parathyroid cells with blue light (450–490 nm, 20 Hz) for 10 s induced a depolarizing membrane potential with a steady-state amplitude at $2.8 \pm 0.9$ mV (Fig. 2d, e). We then used 10 s of light stimuli with a time interval of 30 s to illuminate the human parathyroid cells and found that light stimulation successfully induced ChETA depolarization spike trains (Fig. 2f), suggesting that this protocol using blue light stimulation was sufficient to induce the reversible membrane depolarization of ChETA-expressing parathyroid cells.

To investigate the effects of membrane depolarization on PTH release, we constructed a blue-light-emitting diode (LED) to illuminate the transfected human parathyroid cells for 0.5 h (20 Hz with stimulus intervals of 50 ms) (Fig. 2g–i). Conditional medium was collected at 1, 2, 4, and 24 h after light stimulation (Fig. 2g). Quantification of PTH in the medium was performed from parathyroid cells of patients with SHPT (Fig. 2j–l and Supplementary Fig. 3a). In patient 1, there was no significant difference between ChETA and eYFP groups before the light stimulation, however, PTH was significantly lower 1 h after blue light stimulation in ChETA group compared with the eYFP group, yet not different at 2, 4, and 24 h after light stimulation (Fig. 2j). In patient 2 and 3, PTH levels in the ChETA group were significantly lower than the eYFP group 1 and 2 h after light stimulation and recovered to a level comparable with the eYFP group at 4 and 24 h after stimulation (Fig. 2k, l). Quantification of the samples from the eight patients revealed that PTH levels were unanimously lower 1 h after light stimulation (Supplementary Fig. 3a). Perfusion of cinacalcet, which is indicated for the treatment of hyperparathyroidism, also induced a decrease in PTH level from SHPT cells 1 h following treatment, which is evidence that these cells were healthy and responded to the stimuli (Supplementary Fig. 3c). We also determined the level of PTH secreted from normal rat parathyroid cells after optogenetic

stimulation and quantification showed that PTH levels also decreased after the light stimulation (Supplementary Fig. 3b). Taken together, the data from electrophysiological recordings and the PTH quantification assay indicate that optogenetic stimulation of human parathyroid cells can effectively depolarize the membrane potential and effectively inhibit the level of secreted PTH.

**Optogenetic treatment elevates intracellular calcium in human parathyroid cells and inhibits both PTH synthesis and secretion.** We investigated the molecular mechanism underlying optogenetic inhibition of PTH. We first used a calcium fluorescence assay to reveal light-induced changes of intracellular $Ca^{2+}$. In ChETA-expressing chief cells, blue light stimulation induced a significant increase in fluorescence signal, which then decreased when the light was turned off, and then increased again when the light was turn on again in a rhythmical manner (Fig. 3a); however, this pattern was not seen in the control group that expressed eYFP. The fluorescence signal from this group remained stable when the blue light was turned on and off (Fig. 3b). Quantification of the fluorescence signals confirmed that blue light stimulation did indeed increase the concentration of intracellular $Ca^{2+}$ in the ChETA group, indicated by the elevation and return to baseline of calcium fluorescence (Fig. 3c and Supplementary Fig. 4b, c). However, in the eYFP control group cells, there was no significant elevation of the signals following blue light exposure (Fig. 3d and Supplementary Fig. 4a, c). The fluorescence intensity ratio $\Delta F/F0$ was significantly higher in the ChETA group than in the eYFP group (Fig. 3e).

Because delicate intracellular signaling pathways determine the synthesis and secretion of human parathyroid hormone[19,20], we first treated human parathyroid cells with forskolin, which activates the enzyme adenylyl cyclase, and we found that the levels of PKA and cAMP in parathyroid cells were significantly increased at 5 min, and then recovered 30 min after the treatment (Supplementary Fig. 5a, b). Then, we determined cAMP and PKA levels in human parathyroid cells both before and after light stimulation. We found that PKA levels were significantly lower in the ChETA group than both control and eYFP groups immediately after light stimulation. However, this had recovered 1 h after light stimulation (Fig. 3f). PKA is the downstream target of cyclin AMP (cAMP) and we found that cAMP levels were also significantly lower in the ChETA group following light stimulation, which then recovered to a level comparable to those in the eYFP group 1 h after light stimulation (Fig. 3g).

Next, we investigated the molecular pathway regulating the PTH secretion process. Because the PLA2-arachidonic acid (AA) intracellular signaling pathway may contribute to regulation of PTH secretion[19], we first treated human parathyroid cells with PLA2 activator, which activated the PLA2-arachidonic acid pathway, and we found that the levels of Phospholipases A2 (PLA2), arachidonic acid (AA), 12-lipoxygenase (12-LO), and 15-lipoxygenase (15-LO) in parathyroid cells were significantly increased at both 5 and 30 min after the treatment (Supplementary Fig. 5c–f). We further determined the levels of PLA2, AA, 12-LO, and 15-LO level in parathyroid cells (Fig. 3h–k) after light stimulation. The analysis showed that PLA2 and AA levels were significantly higher in the ChETA group compared with eYFP group immediately after light stimulation, then had recovered by 1 h after light stimulation (Fig. 3h–i). Levels of 12-LO and 15-LO, important enzymes in PLA2-arachidonic acid (AA) signaling pathways, were also significantly higher in the ChETA group and were restored to a level comparable to the eYFP group 1 h after light stimulation (Fig. 3j, k). Taken together, these data suggest that optogenetic regulation of parathyroid cells can reversibly

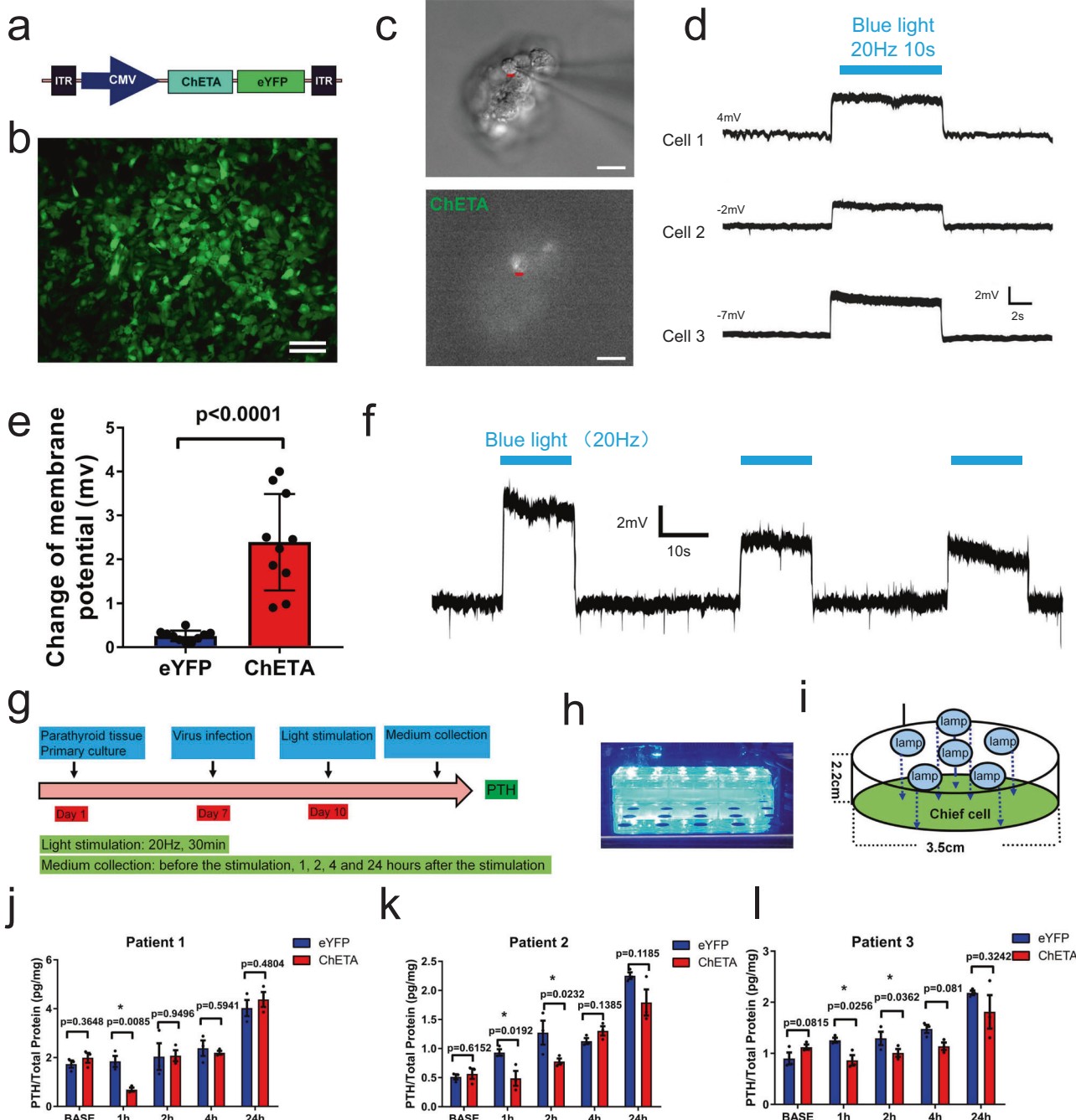

**Fig. 2 Blue light stimulation depolarizes membrane potential and inhibits secretion of PTH in parathyroid cells from patients with secondary hyperparathyroidism. a** Schematic showing lentivirus carrying the CMV-ChETA-eYFP construct. **b** Representative image of parathyroid cells expressing ChETA-eYFP. Scale bar = 50 μm. **c** Whole-cell patch-clamp was used to investigate the function of ChETA in human parathyroid cells. Scale bar = 20 μm. **d** Blue light stimulation induced membrane depolarization of ChETA-expressing parathyroid cells. **e** Change of depolarized membrane potential in the eYFP and ChETA groups ($n = 10$ cells). Two-tailed unpaired $t$-test, $p < 0.0001$. Values represent mean ± SEM. **f** Blue light stimulation induced ChETA depolarization spike trains. **g** Schematic diagram of PTH secretion at different time points after light stimulation of ChETA-expressing parathyroid cells. **h** A blue light-emitting diode (LED) was used to illuminate the cultured cells in 6-well plates. **i** Schematic showing the arrangement of the lamps in the LED constructed for light illumination. **j** Quantification of PTH was performed on the stimulated human parathyroid cells from SHPT patient 1 ($n = 3$ times per group). Two-tailed unpaired $t$-test, $p$ values as indicated. Values represent mean ± SEM. **k** Quantification of PTH was performed on stimulated human parathyroid cells from SHPT patient 2 ($n = 3$ times per group). Two-tailed unpaired $t$-test, $p$ values as indicated. Values represent mean ± SEM. **l** Quantification of PTH was performed on stimulated human parathyroid cells from patient 3 with SHPT ($n = 3$ times per group). Two-tailed unpaired $t$-test, $p$ values as indicated. Values represent mean ± SEM. Source data are provided as a Source Data file.

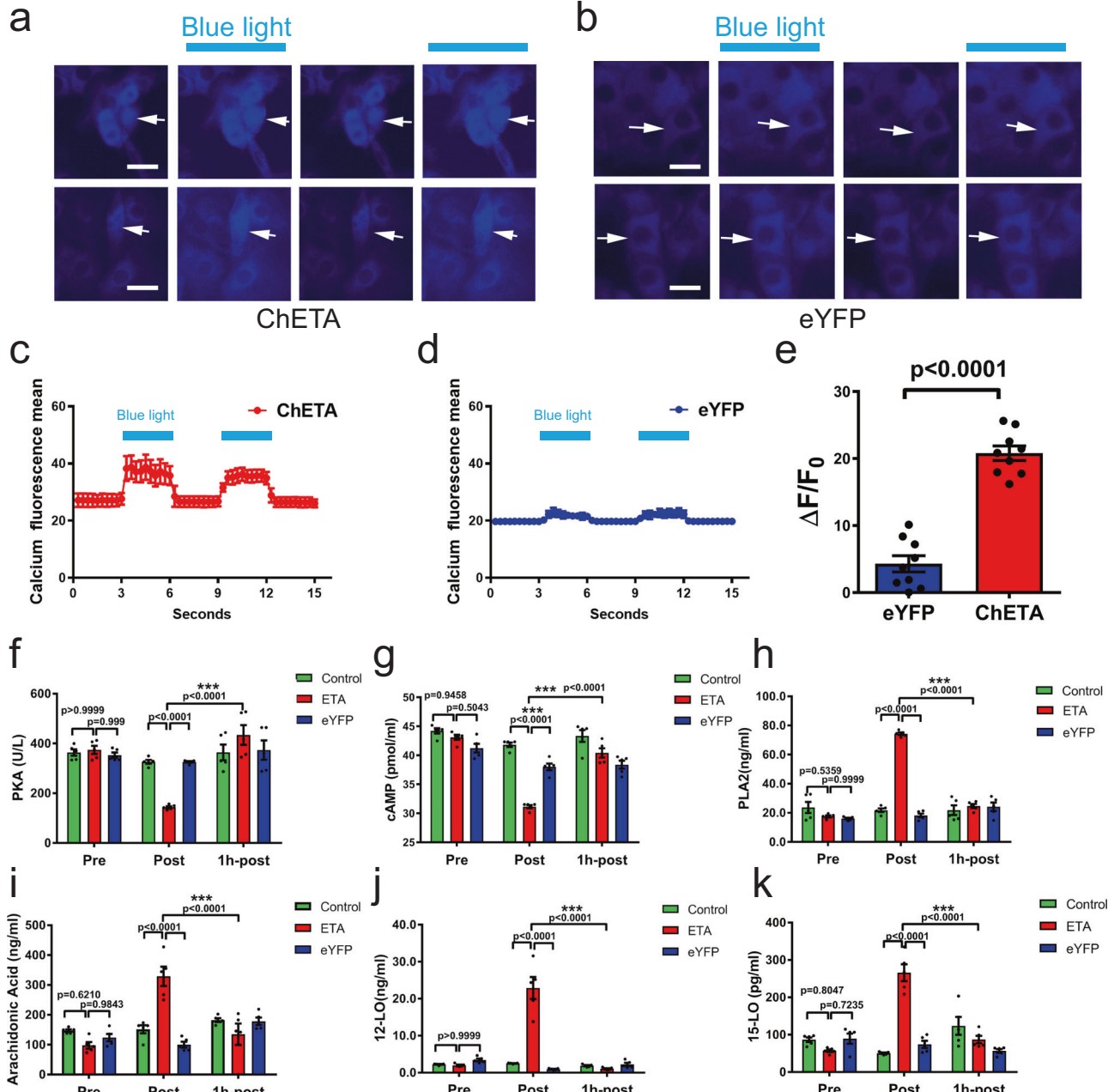

**Fig. 3 Optogenetic regulation elevates intracellular Ca$^{2+}$ in human parathyroid cells and inhibits both synthesis and secretion of PTH. a** Blue light stimulation induced a significant increase in calcium fluorescence signal in ChETA-expressing cells, and this increase is reproducible. Scale bar = 20 μm. **b** Fluorescence signals in control eYFP-expressing cells remained relatively stable when blue light was turned on and off. Scale bar = 20 μm. **c** Quantification of the intensity of calcium fluorescence signals over time in ChETA-expressing cells ($n = 3$ cells). Values represent mean ± SEM. **d** Quantification of calcium fluorescence signals with time in control eYFP-expressing cells ($n = 3$ cells). Values represent mean ± SEM. **e** Fluorescence intensity ratios $\Delta F = F/F0$ in the ChETA and control eYFP groups ($n = 9$ per group). Two-tailed unpaired $t$-test, $p < 0.0001$. Values represent mean ± SEM. **f** Quantification of PKA levels in the control, eYFP and ChETA groups before (Pre), immediate after (Post) and 1 h after light stimulation (1 h-post) ($n = 5$ per group). **g** Quantification of cAMP levels in the control, eYFP, and ChETA groups before, immediate after and 1 h after the light stimulation ($n = 5$ per group). **h** Quantification of Phospholipases A2 (PLA2) in the control, eYFP and ChETA groups before, immediate after and 1 h after the light stimulation ($n = 5$ per group). **i** Quantification of arachidonic acid (AA) in the control group, eYFP group and ChETA groups before, immediate after and 1 h after the light stimulation ($n = 5$ per group). **j** Quantification of 12-lipoxygenase (12-LO) in the control, eYFP, and ChETA groups before, immediate after and 1 h after the light stimulation ($n = 5$ per group). **k** Quantification of 15-lipoxygenase (15-LO) in the control, eYFP, and ChETA groups before, immediate after and 1 h after the light stimulation ($n = 5$ per group). All statistical tests in **f-k** used: Two-tailed unpaired $t$-test, $p$ values as indicated. Values represent mean ± SEM. Source data are provided as a Source Data file.

increase intracellular calcium levels in addition to inhibiting the secretion of human PTH through the PKA/cAMP and PLA2-arachidonic acid (AA) signaling pathways.

To further study whether light stimulation can regulate PTH production, we evaluated changes in *GCM2*, *MAFB*, and *PTH*, genes closely related to PTH production, before and after the light illumination. We found that light stimulation effectively inhibited expression levels of *GCM2*, *MAFB*, and *PTH*, and this inhibition returns to normal levels within 1 h after cessation of light stimulation (Supplementary Fig. 6a–c). In addition, we also tested the effect of light stimulation on expression levels of genes related to intracellular membrane vesicle transport in PTG cells, such as *Rab8a* and *Rab11a*. The results show that light stimulation can effectively inhibit expression levels of *Rab8a* and *Rab11a*, and that this inhibition returns to normal levels within one hour after the light stimulation (Supplementary Fig. 6d, e). These data show that

levels of mRNA for genes involved in PTH synthesis and vesicle transport were suppressed in the optogenetic stimulated group.

**Optogenetic treatment inhibits ex vivo PTH secretion in an ex vivo organoid culture model of rat parathyroid gland.** To further investigate the effects of optogenetic inhibition of PTH release, we constructed an organoid culture model using rat parathyroid gland (Fig. 4a–c). Histology and immunofluorescence showed that the predominant cell type in rat parathyroid is chief cells (Fig. 4a), and most chief cells expressed PTH (Fig. 4b). We then injected a lentivirus carrying the CMV-ChETA-eYFP construct into the rat parathyroid gland; the CMV-eYFP construct was similarly injected in a control group (Fig. 4d). Parathyroid glands were dissected and cultured in a petri dish 21 days after the in vivo transfection (Fig. 4e, f). Over the next seven-day

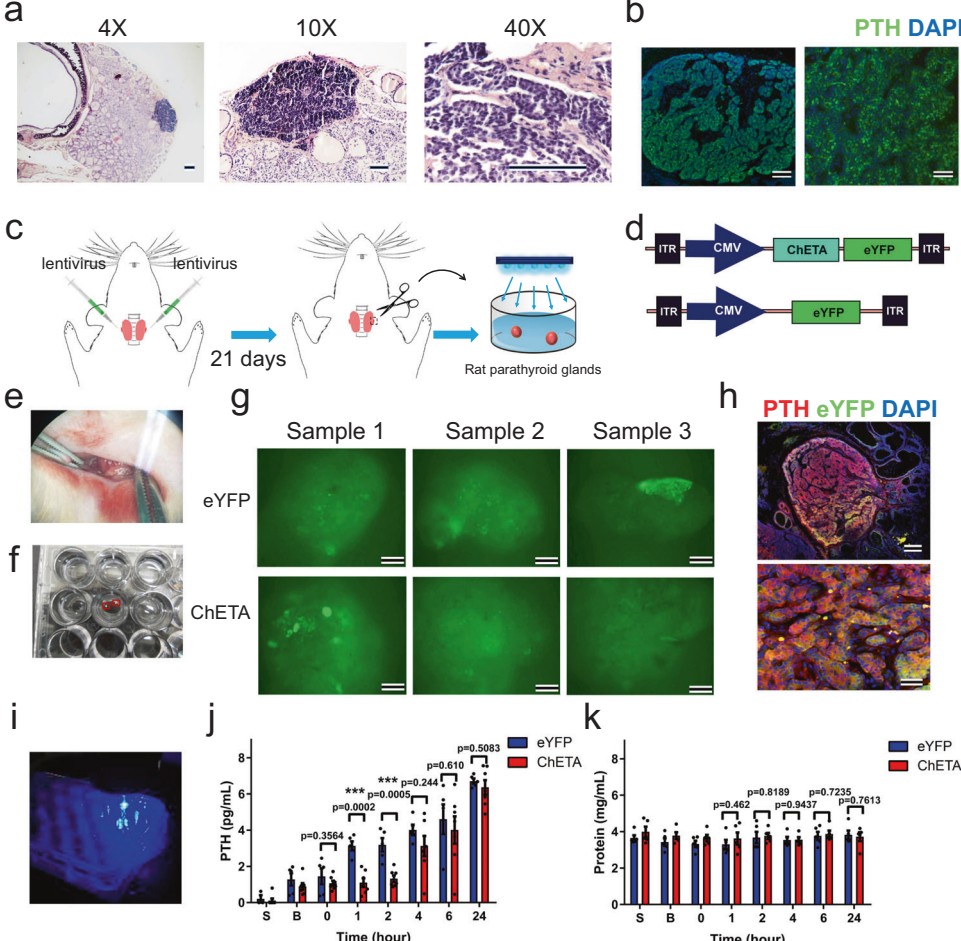

**Fig. 4 Light stimulation inhibits ex vivo PTH secretion in an organoid culture model of the rat parathyroid gland. a** Hematoxylin & Eosin (HE) staining of the rat parathyroid gland at different magnifications. Scale bar = 100 μm. **b** Immunofluorescence of the rat parathyroid gland showing that most chief cells expressed parathyroid hormone (PTH). Left scale bar = 100 μm. Right scale bar = 50 μm. **c** Schematic protocol showing the lentivirus carrying the CMV-ChETA-eYFP construct injected into the rat parathyroid gland, and rat parathyroid culture model. **d** Schematic of the lentivirus carrying the CMV-ChETA-eYFP construct and the control CMV-eYFP construct. **e** Dissection of the rat parathyroid after in vivo transfection. **f** Petri dish ex vivo culture of the rat parathyroid gland (red box). **g** Green fluorescence was continuously observed in the parathyroid gland in both ChETA and control eYFP groups. Scale bar = 100 μm. **h** Double staining of PTH and eYFP was performed on sections of the cultured gland and co-localization of ChETA-eYFP and PTH were observed in the parathyroid gland. Upper scale bar = 100 μm. Lower scale bar = 50 μm. **i** Illumination of transfected rat parathyroid gland using an LED. **j** Quantification of PTH levels in the conditional medium from both ChETA and eYFP groups before (B) and at 0, 1, 2, 4, 6, and 24 h after light stimulation. S saturated phase of parathyroid cell plating (*n* = 7 for ChETA, *n* = 5 for eYFP). Two-tailed unpaired *t*-test, *p* values as indicated. Values represent mean ± SEM. **k** Quantification of the total protein level in cultured rat parathyroid gland in ChETA and eYFP groups before (B) and at 0, 1, 2, 4, 6, and 24 h after the light stimulation (*n* = 5 per group). Two-tailed unpaired *t*-test, *p* values as indicated. Values represent mean ± SEM. Source data are provided as a Source Data file.

culture period, green fluorescence was continuously observed in parathyroid glands in both the ChETA and control eYFP groups (Fig. 4g). Double staining of PTH and eYFP was performed on sectioned parathyroid gland, which showed that ChETA expression (green) was mainly restricted to the parathyroid gland (Fig. 4h), whereas few fluorescent cells were observed outside the parathyroid region, illustrating the successful expression of the optogene ChETA in the cultured parathyroid gland.

We then used a blue light-emitting diode to illuminate the transfected parathyroid organoid for 0.5 h (Fig. 4i) and collected the conditioned medium to determine PTH levels in both ChETA and eYFP groups. We found that there was no significant difference in PTH levels between ChETA and eYFP groups before light stimulation but PTH levels were significantly lower in the ChETA group at 1 and 2 h following blue light stimulation and then returned such that there was no difference at 4, 6, and 24 h (Fig. 4j). During light stimulation, the total protein level remained stable and there was no difference between ChETA and eYFP groups (Fig. 4k). These data suggest that the optogenetic treatment specifically and effectively inhibited the release of the parathyroid hormone in an established rat parathyroid organoid culture model.

Since extracellular calcium and optogenetic stimulation both successfully induced a decrease in PTH levels in normal parathyroid cells, we compared the inhibitory effects of light stimulation and raising extracellular calcium on PTH level. Quantification of the level of PTH secreted from rat parathyroid cells was performed at 0.5, 1.5, and 2.5 mM extracellular $Ca^{2+}$, and we found that the gradually increased extracellular $Ca^{2+}$ level significantly inhibited PTH secretion from parathyroid cells (Supplementary Fig. 7a). The level of PTH at 2.5 mM extracellular $Ca^{2+}$ was significantly lower that the PTH at 0.5 mM and 1.5 mM $Ca^{2+}$. Next, we investigated the effects of light stimulation duration and frequency on inhibition of secreted PTH and found that increased light stimulation duration significantly enhanced the inhibitory effects on PTH levels. Thirty minutes of blue light stimulation (20 Hz) induced a significantly lower PTH level than 15 min of stimulation did. However, there was no significant differences in PTH levels between 30 min and 1 h of light stimulation (Supplementary Fig. 7b). It is interesting to note that PTH levels were comparable between 10, 20, and 30 Hz of light stimulation, suggesting that changing the light frequency may not enhance the inhibitory effects on PTH (Supplementary Fig. 7c).

**Optogenetic treatment effectively inhibits in vivo release of PTH in an established rat model of secondary hyperparathyroidism**. To further investigate the in vivo inhibitory effect of the optogenetic approach on the elevated release of PTH, we established a rat model of secondary hyperparathyroidism (SHPT) (Fig. 5a–c and Supplementary Fig. 8a). In the control group, hematoxylin and eosin (HE) staining of the parathyroid gland showed a symmetric distribution of chief cells and homogenous cellularity. However, in the parathyroid of SHPT rats, we observed typical histological features of secondary hyperparathyroidism: heterogeneous distribution of the chief cells in addition to nodularity and enlargement of the connective fibrous tissue (Fig. 5a and Supplementary Fig. 8b–e). Quantification of PTH levels in serum showed that PTH in the SHPT group was significantly higher than that of the control group (Fig. 5b and Supplementary Fig. 8f).

We then investigated the effects of light stimulation on the rat parathyroid gland in vivo (Fig. 5c). Strong signals were observed in the parathyroid glands in the eYFP and ChETA group one week after injecting the CMV-ChETA-eYFP lentivirus into the parathyroid gland (Fig. 5d and Supplementary Fig. 9). To evaluate

the efficiency of virus transfection, we counted the PTH positive cells expressing green fluorescent protein. On average, 46.69% of PTH[+] cells expressed ChETA-eYFP (Fig. 5f). To reveal the transfection specificity, we double stained ChETA-eYFP and PTH and found that ChETA expression (green) was mainly restricted to the parathyroid gland (red) (Fig. 5e). Quantification revealed that around 86% of the ChETA-positive cells expressed PTH (Fig. 5f), suggesting that we successfully achieved in situ expression of the optogene ChETA in the hyperplastic rat parathyroid gland.

We then illuminated the transfected rat parathyroid gland in vivo for 0.5 h and determined PTH levels in both the ChETA and eYFP groups. We found that serum PTH levels were comparable between the ChETA and eYFP groups before blue light stimulation. However, PTH levels were significantly lower in the ChETA group than in the eYFP group 5 and 15 min after light stimulation, a difference that lasted until 30 min after stimulation (Fig. 5g). We also determined calcium levels in the serum of treated rats and interestingly, we found that light-induced calcium responses occurred after the decrease in PTH. There was no significant difference in calcium signal between the ChETA group and eYFP groups immediately after stimulation, however, serum calcium decreased significantly in the ChETA group 30 min after blue light stimulation (Fig. 5h). To investigate the cellular mechanism of optogenetic inhibition of PTH decrease after the transfection and light stimulation, we compared the expression of PTH and the ChETA in situ on the rat parathyroid gland chief cells using immunofluorescence staining. We found that the cells with high expression of ChETA had low expression of PTH after light stimulation. Chief cells with high PTH expression were typically not transfected with ChETA. However, there were also cells that expressed low PTH without expression of ChETA (Supplementary Fig 10). These cells might have been regulated by paracrine signaling from neighboring ChETA-transfected cells through non-cellular autonomous effects. Taken together, these data suggest that optogenetic treatment effectively inhibited PTH secretion in vivo and also led to decreased calcium levels in an established rat model of SHPT.

**Optogenetic regulation of human PTH can be controlled automatically using ionized calcium concentration**. We show above the decreased response of parathyroid cells from SHPT patients to extracellular $Ca^{2+}$ (Fig. 1); the ideal physiological regulatory secretion of human PTH is determined by serum $Ca^{2+}$ concentration through negative feedback loops. To achieve this goal, we first established an ex vivo human parathyroid organoid culture model and regulated PTH secretion using optogenetics (Fig. 6a). Parathyroid tissues were obtained from patients with secondary hyperparathyroidism and cultured ex vivo and then a lentivirus carrying the CMV-ChETA-eYFP construct was used to transfect this cultured human parathyroid tissue. Forty-eight hours after the transfection, most of the parathyroid tissue were successfully labeled, expressing green fluorescence (Fig. 6b). Calcium monitoring using an electrode and blue light stimulation (20 Hz and stimulus intervals of 50 ms) was used to examine the human parathyroid culture model (Fig. 6c). When $Ca^{2+}$ reached a level of 50 mg/L, blue light was automatically turned on and we found that the PTH level in the conditional medium decreased 2 h after light stimulation in the ChETA group (Fig. 6d). However, when $Ca^{2+}$ level was reduced below 50 mg/L, blue light was turned off automatically and there was no difference in PTH levels between ChETA and eYFP groups (Fig. 6d). This demonstrates that light-induced inhibition of PTH can be controlled automatically using the level of extracellular calcium in the human parathyroid organoid culture model.

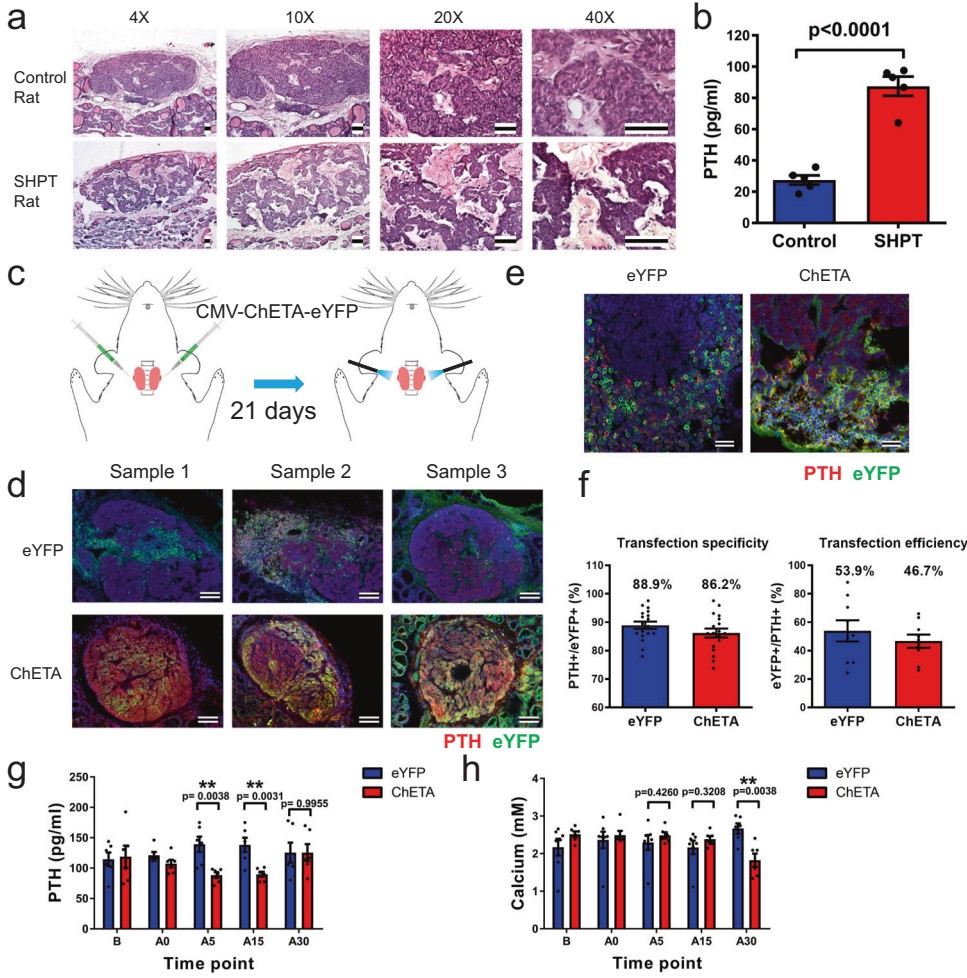

**Fig. 5 Optogenetic regulation effectively inhibits the in vivo release of PTH in an established rat model of secondary hyperparathyroidism. a** HE staining of control rat parathyroid gland and hyperplastic parathyroid from secondary hyperparathyroidism (SHPT) rats at different magnifications. Heterogeneous distribution of chief cells, nodularity, and enlargement of connective fibrous tissue was observed in SHPT rat parathyroid. Scale bar = 100 μm. **b** Quantification of serum levels of PTH in the SHPT and control groups ($n = 5$ rats per group). Two-tailed unpaired $t$-test, $p < 0.0001$. Values represent mean ± SEM. **c** Schematic showing the injection of the CMV-ChETA-eYFP lentivirus into the parathyroid gland and blue light stimulation (470 nm, 20 Hz) of the gland for 30 min. **d** Immunofluorescence of representative parathyroid glands from the eYFP and ChETA groups. Strong florescence signals were observed in the ChETA group. Scale bar = 100 μm. **e** Double staining of ChETA-eYFP and PTH of the sectioned rat parathyroid gland. Scale bar = 50 μm. **f** Quantification of the number of PTH+/ChETA+ double positive cells per section. A total of 86% (865/1000) of ChETA-positive cells expressed PTH from 20 sections from three rats per group. Quantification of the number of ChETA+/ PTH+ double positive cells per section. A total of 46.7% of PTH-positive cells expressed ChETA+. Values represent mean ± SEM. **g** Quantification of serum PTH levels in the ChETA and eYFP groups before (B) and at 0, 5, 15, and 30 min after light stimulation. PTH levels in the ChETA group were lower than those of the control group at 5 and 15 min following light stimulation ($n = 6$ rats per group). Two-tailed unpaired $t$-test, $p$ values as indicated. Values represent mean ± SEM. **h** Quantification of calcium levels in the serum of rats before (B) and at 0, 5, 15, and 30 min after light stimulation. Serum calcium was lower in ChETA group than the control group 30 min after blue light stimulation ($n = 6$ rats per group). Two-tailed unpaired $t$-test, $p$ values as indicated. Values represent mean ± SEM. Source data are provided as a Source Data file.

Precise regulation of in vivo PTH release based on serum $Ca^{2+}$ level is required for transplanted parathyroid tissue[12]. To achieve the goal of precise regulation of human PTH in vivo, we transplanted the ChETA-transfected human parathyroid tissue into the backs of nude mice (Supplementary Fig. 12a). Seven days after transplantation, the transplanted human parathyroid tissue remained intact (Fig. 6f) and double immunostaining revealed that ChETA expression (green) was mainly restricted to the parathyroid chief cells (red) (Fig. 6g and Supplementary Fig. 11a, b). Positive expression of tyrosine hydroxylase (TH) and alpha-smooth muscle actin were also observed in the transplanted tissue (Supplementary Fig. 12b, c), suggesting nerve and blood vessel ingrowth in the transplanted human parathyroid gland.

We then combined blue-light illumination with simultaneous monitoring of serum $Ca^{2+}$ using microdialysis probes (Fig. 6e and Supplementary Fig. 13a, b). Mouse tail veins were injected with $CaCl_2$ and elevated levels of serum $Ca^{2+}$ automatically triggered blue light to turn on. Levels of PTH were lower in the ChETA group compared to the control group at 5 and 15 min after the 0.5 h light stimulation (Fig. 6h) and serum calcium was lower at 15 and 30 min (Fig. 6i). To investigate the compensatory effects of unaffected parathyroid tissue on PTH and calcium level, we performed graft removal in nude mice bearing human PTG graft. It takes 15 min for endogenous PTH to compensate for the decrease in PTH after the graft removal (Supplementary Fig 14a). Furthermore, unilaterally removal of PTG in C57 mice and sHPT

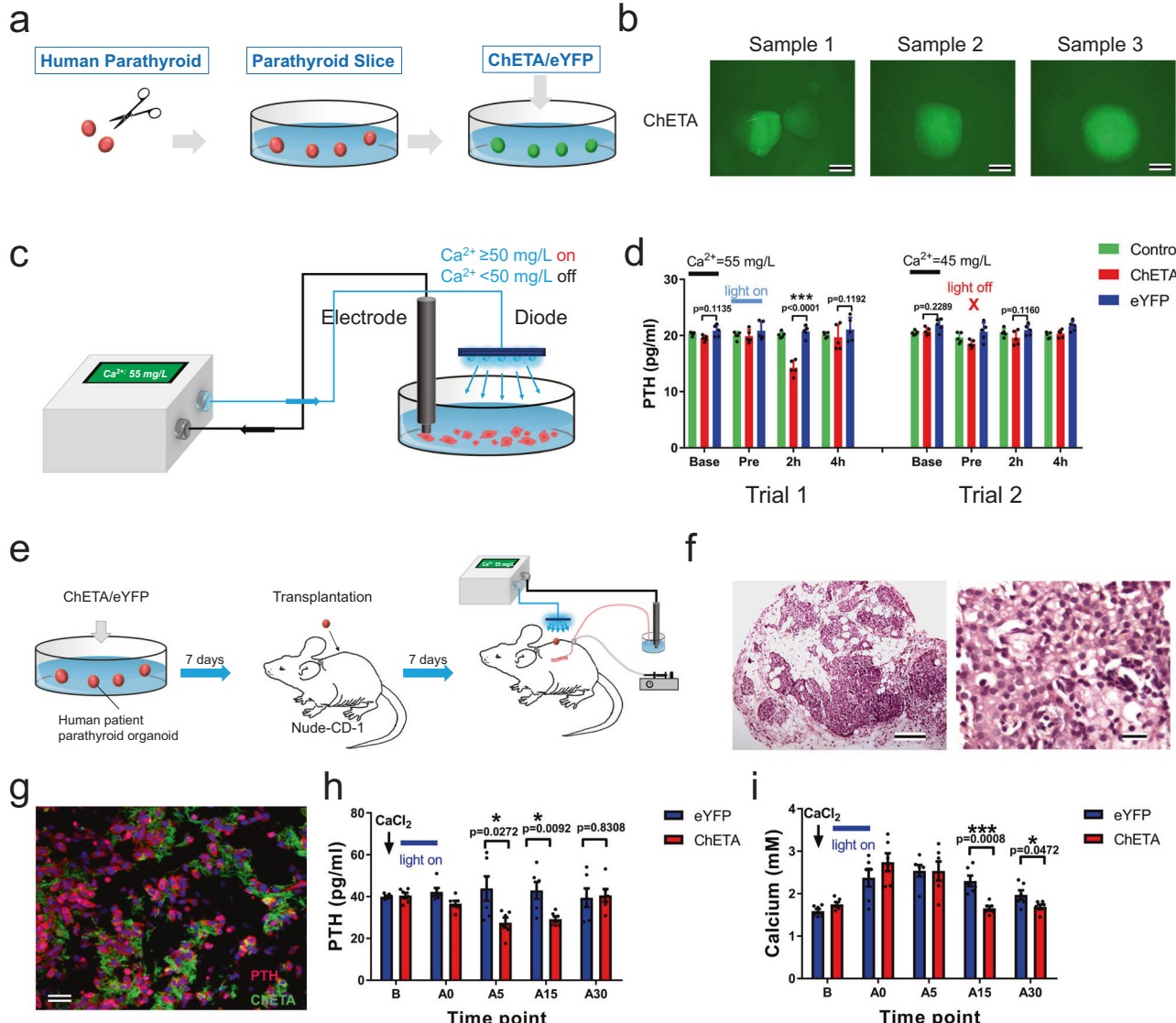

**Fig. 6 Optogenetic regulation of PTH was automatically controlled by monitoring ionized calcium concentration ex vivo and in vivo. a** Schematic showing the establishment of a human parathyroid organoid culture model. The CMV-ChETA-eYFP construct was used to transfect cultured human parathyroid tissue. **b** Most human parathyroid tissue was successfully labeled with ChETA gene 48 h after the transfection. Scale bar = 100 μm. **c** Schematic showing the combination of calcium monitoring with electrode and the optogenetic regulation of human parathyroid tissue cultured in a petri dish. **d** Blue light was automatically turned on when $Ca^{2+}$ reached above the level of 50 mg/L (Trial 1) and was turned off when $Ca^{2+}$ was below 50 mg/L (Trial 2). PTH level was quantified for the ChETA and eYFP groups before (Pre) and 2 and 4 h after the light stimulation ($n = 5$ samples per group). Two-tailed unpaired t-test, p values as indicated. Values represent mean ± SEM. **e** Schematic showing serum calcium monitoring using microdialysis probes and illumination using blue light in the established human parathyroid transplantation model in nude mice. **f** HE staining showing the intact structure of the human parathyroid organoid transplanted into nude mice. Scale bar = 50 μm. **g** Double staining showed that ChETA expression (green) was mainly restricted to the parathyroid chief cells expressing PTH (red). Scale bar = 50 μm. **h** Quantification of the PTH level in the eYFP and ChETA groups before (B) and at 0, 5, 15, and 30 min after light stimulation. Blue light was automatically turned on when the $CaCl_2$ was injected into the mouse tail vein ($n = 6$ mice per group). Two-tailed unpaired t-test, p values as indicated. Values represent mean ± SEM. **i** Quantification of calcium levels in the eYFP and ChETA groups before (B) and at 0, 5, 15, and 30 min after light stimulation. Serum calcium was lower in the ChETA group than the eYFP group at 15 and 30 min following the light stimulation ($n = 6$ mice per group). Two-tailed unpaired t-test, p values as indicated. Values represent mean ± SEM. Source data are provided as a Source Data file.

rats induced decreased serum PTH and did not recover to pre-operation level within 30 min (Supplementary Fig 14b, c). Additionally, intravenous injection of calcium and calcium antagonists did not cause dramatic fluctuations in PTH levels in a short period of time (Supplementary Fig 14d). All these results suggested that the compensatory capacity of unaffected parathyroid cells in mice, rats or endogenous parathyroid glands

from nude mice is not instantaneous and limited for the recovery of PTH and calcium.

Taken together, these ex vivo and in vivo experiments consistently demonstrate the feasibility of the combination of $Ca^{2+}$ monitoring and optogenetic regulation of the transplanted human parathyroid tissue, and that the optogenetic inhibition of PTH can be automatically controlled by serum $Ca^{2+}$ level.

**Optogenetic inhibition of PTH can enhance bone formation and decrease bone resorption through regulation of the bone remodeling process**. Fragile bones and decreased bone density are the most common symptoms of secondary hyperparathyroidism and chronic elevation of PTH in hyperparathyroidism can increase bone resorption and contribute to net bone loss over time[21]. We next investigated the chronic and long-term effects of optogenetic inhibition of human PTH release on bone formation

and resorption. Human ChETA-transfected parathyroid tissue from SHPT patients was transplanted into the backs of nude mice (Supplementary Fig. 15a, b) and blue light was used to illuminate the transplanted tissue for 30 min every other day during a period of 28 days (Fig. 7a). We found that serum levels of human PTH in nude mice were significantly lower in the ChETA group than in the PTG group (transplanted PTG without transfection) 5 and 10 min after blue-light stimulation, and had recovered at 15 min

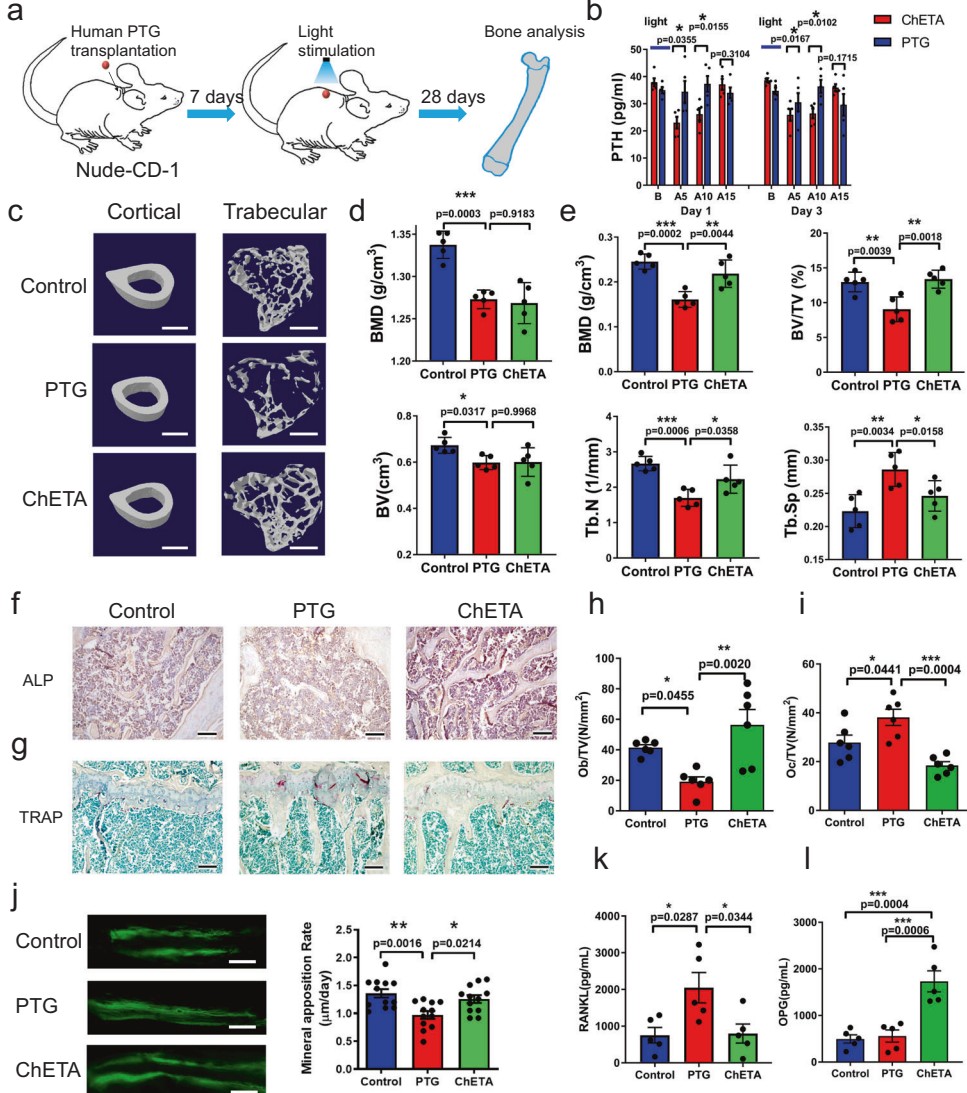

**Fig. 7 Optogenetic stimulation inhibits PTH secretion and promotes bone remodeling. a** Schematic diagram of the establishment of a nude mouse PTG transplantation model. **b** Quantification of PTH levels in the ChETA and PTG groups before (B) and at 5, 15, and 30 min after light stimulation ($n = 5$ per group). Two-tailed unpaired $t$-test, $p$ values as indicated. Values represent mean ± SEM. **c** MicroCT scanning was used to evaluate the structural changes of both cortical and trabecular bone in normal control, PTG and ChETA groups 28 days after blue light stimulation. Scale bar $= 1$ mm. **d** Quantification of bone mineral density (BMD) and bone volume (BV) of cortical bone from normal control, PTG, and ChETA groups ($n = 5$ per group). $p = 0.003$, $p = 0.9183$ for BMD; $p = 0.0317$, $p = 0.9968$ for BV. **e** Quantification of bone mineral density (BMD), bone volume/tissue volume (BV/TV), trabecular number (TbN), and trabecular separation (Tb.Sp) of trabecular bone from normal control, PTG, and ChETA groups ($n = 5$ per group). **f** Alkaline phosphatase (ALP) staining of trabecular bone in the control, PTG, and ChETA groups. Scale bar $= 150$ μm. **g** Tartrate-resistant acid phosphatase (TRAP) staining in the control, PTG and ChETA groups. Scale bar $= 150$ μm. **h** Quantification of ALP-positive osteoblast per tissue volume (Ob/TV) in trabecular bones from control, PTG and ChETA groups ($n = 6$ per group). $p = 0.0455$, $p = 0.002$, respectively. **i** Quantification of the number of TRAP-positive osteoclasts per tissue volume (Oc/TV) in trabecular bones from control, PTG, and ChETA groups ($n = 6$ per group). $p = 0.0441$, $p = 0.0004$, respectively. **j** In vivo calcein labeling and calculation of mineral apposition rate (MAR) in cortical bone in the control, PTG and ChETA groups ($n = 12$ sections per group). $p = 0.0016$, $p = 0.0214$, respectively. Scale bar $= 100$ μm. **k** Quantification of serum Rankl level in the control, PTG, and ChETA group ($n = 5$ mice per group). $p = 0.0287$, $p = 0.0344$, respectively. **l** Quantification of serum osteoprotegrin (OPG) levels in the control, PTG and ChETA groups ($n = 5$ per group). All statistical tests in **d–l** used: One-way analysis of variance (ANOVA) with Tukey's multiple comparisons test, $p$ values as indicated. Values represent mean ± SEM. Source data are provided as a Source Data file.

(Fig. 7b). There was no significant difference in PTH levels between groups at 3 and 9 h following stimulation (Supplementary Fig. 15c). Serum calcium decreased significantly in the ChETA group at 15 and 30 min after the blue-light stimulation, and then calcium levels recovered at 3 h and maintained these levels until at least 9 h (Supplementary Fig. 15d). Such a reversible inhibition of PTH was successfully reproduced at day 3 of the illumination schedule (Fig. 7b).

We then investigated whether this chronic and pulsatile inhibition of human PTH level could ameliorate loss of bone structure in nude mice. MicroCT scanning was used to evaluate structural changes of both cortical bone and trabecular bone in control (without PTG transplantation), PTG and ChETA groups 28 days after illumination (470 nm, 20 Hz, 9 mW; 30 min per session every other day over a 28-day period) (Fig. 7c). For cortical bone, we found that bone mineral density (BMD) and bone volume (BV) was significantly lower in the PTG group and light stimulation did not lead to the recovery of BMD or BV of cortical bone in the ChETA group (Fig. 7c, d). However, there was a significant enhancement of trabecular bone formation in the ChETA group compared to the PTG group (Fig. 7c, e): BMD was 20% higher in ChETA group compared to the PTG group and bone volume/tissue volume (BV/TV) was 25% higher. Similarly, the trabecular number (TbN) was 20% higher in ChETA group compared with the PTG group, accompanied by 15% lower trabecular separation (Tb.Sp) (Fig. 7e). Taken together, these data suggest that the pulsatile inhibition of PTH induced by optogenetic regulation can significantly ameliorate the loss of bone structure.

An obvious lower bone mass in the PTG (transplanted PTG without transfection) group and less trabecular bone structure were revealed by HE staining (Supplementary Fig. 16a). However, in the ChETA group, bone mass was greater and more trabecular bone structure was observed in the bone cavity compared to the PTG group (Supplementary Fig. 16a). Alkaline phosphatase (ALP) staining was conducted to evaluate bone formation in the different groups. We found that ALP expression level was attenuated in the PTG group. However, it was enhanced significantly in the ChETA group (Fig. 7f); the number of ALP-positive osteoblasts per tissue volume (Ob/TV) was significantly higher in ChETA group compared to the PTG group (Fig. 7h). Tartrate-resistant acid phosphatase (TRAP) staining was conducted to evaluate the bone resorption rate and osteoclast activity. We found much higher TRAP staining of trabecular bone in the PTG group; however, the staining intensity was significantly lower in the ChETA group (Fig. 7g). Quantification revealed that the number of TRAP-positive osteoclasts per tissue volume (Oc/TV) was significantly lower in ChETA group compared with the PTG group (Fig. 7i), suggesting decreased bone resorption after the inhibition of PTH. In vivo calcein labeling confirmed that new bone formation and mineral apposition rate (MAR) in cortical bone were both higher in the ChETA group than in the PTG group (Fig. 7j). Furthermore, the mineralizing surface versus the bone surface (MS/BS) level in the ChETA group was significantly higher than that of the PTG group, but similar to that of the control group (Supplementary Fig. 16d). Nevertheless, the bone formation rate in cortical bone was higher in the ChETA group than in the PTG group (Supplementary Fig. 16e).

Next, we investigated beta-catenin expression in trabecular bone after optogenetic inhibition of PTH. We found that the beta-catenin signals were sparsely distributed in the bone marrow in the PTG group. However, in the ChETA group, intense beta-catenin signals were observed both in the trabecular bone and bone marrow area (Supplementary Fig. 16b). We also performed gene expression analysis to investigate the effects of PTH on gene expression (beta-catenin, LRP6, FZD1, Wnt4, and Dkk-1) in the

Wnt signaling pathway and found that the expression of beta-catenin, LRP6, and FZD-1 in trabecular bone was higher in the ChETA group than in the PTG group. However, there was no significant difference in Wnt4 and Dkk-1 expression (Supplementary Fig. 16c).

Finally, we determined the level of Rank ligand, which regulates bone resorption and osteoclast activity[22] in serum, and we found that it was higher in the PTG group than control group, yet lower in the ChETA group compared to the PTG group on the last day of the 28-day illumination protocol (Fig. 7k). The level of the osteoprotegrin (OPG) was also determined and there was no significant difference between the PTG and control groups. However, OPG levels were higher in the ChETA group than in the control group at day 28 after the illumination (Fig. 7l).

## Discussion

In this study, we used an optogenetic approach to regulate the release of human PTH and investigated the underlying molecular mechanisms. We found that optical stimulation of SHPT-patient-derived parathyroid cells can induce inhibition of human PTH. Inhibition of human PTH secretion can be precisely controlled both ex vivo and in vivo by light based on ionized calcium concentration. Such an optogenetic approach could be used to regulate the systemic calcium balance and to markedly prevent hyperplastic parathyroid tissue-induced bone loss by influencing the bone remodeling process in mice.

PTH is one of the most important regulators of calcium and phosphorus homeostasis and is mainly secreted by parathyroid gland chief cells. Precise and rapid regulation of PTH secretion in response to variations in serum calcium is mainly mediated by CaSR on parathyroid cells[4]. Allosteric modulators of CaSR were the first drugs in their class to become available for clinical use to treat secondary hyperparathyroidism (SHPT)[23]. However, hyperplasia of the parathyroid in SHPT is always accompanied by a reduction of CaSR expression and decreased sensitivity to serum calcium on PTH secretion[6,24]. Mutations or lack of CaSR expression in the parathyroid of some patients leads to the progression of hyperparathyroidism and resistance to cinacalcet[9]. Autoantibodies against CaSR are also found in patient serum, which may inhibit the response of CaSR to the extracellular calcium[25]. In our study, we evaluated the calcium-induced response of primary parathyroid cells from SHPT patients and we found that human parathyroid cells isolated from SHPT indeed had a decreased response to extracellular $Ca^{2+}$; high extracellular calcium did not lead to intracellular $Ca^{2+}$ concentration changes and inhibition of PTH secretion in hyperplastic parathyroid cells. Our patch clamp data demonstrate that the low response of hyperplastic parathyroid cells to calcium and indicate that CaSRs in SHPT patients were not able to regulate PTH secretion. This blunted response to extracellular calcium by parathyroid cells isolated from patients with secondary hyperparathyroidism was consistently supported by our PTH assay, calcium fluorescence data and electrophysiological recordings.

To overcome the shortcomings of CaSR in regulating PTH, we developed an optogenetic approach to regulate the secretion of PTH using blue light. Our data suggest that optogenetic stimulation can effectively depolarize the membrane potential of parathyroid cells. Most importantly, our data demonstrate that light stimulation can effectively and reversibly inhibit PTH release from the patient-derived parathyroid cells. The inhibition of PTH release in eight patient-derived cells after light-induced depolarization adds weight to the notion that optogenetics may be a feasible and efficient method to regulate PTH secretion in human hyperplastic parathyroid cells.

Synthesis and secretion of PTH are stringently controlled by parathyroid cells through intracellular signaling pathways. We observed a robust propensity of the cells to reduce and then recover PTH levels at specific time intervals after the light stimulation, suggesting that the optogenetic treatment regulated downstream molecular pathways in a time-resolved pattern. A calcium fluorescence assay showed that blue light stimulation effectively regulated the intracellular calcium level in human parathyroid cells. The decrease of PKA and cAMP shortly after stimulation implies that the secretion of PTH may be reversibly inhibited by optogenetic treatment. The light-induced elevation and recovery of the levels of PLA2, arachidonic acid, 12-LO, and 15-LO suggest that PLA2-arachidonic acid (AA) signaling pathways might be, at least partially, involved in the light-stimulation-induced inhibition of PTH release. Our data thus demonstrate that an optogenetic approach can induce elevation of intracellular calcium and regulate downstream signaling molecules, and the time-resolved change of these intracellular signals were concordant with PTH levels after light stimulation.

The ex vivo organoid culture model has been used to recapitulate cell–cell interactions, dissect functional molecular networks and facilitate disease modeling[26,27]. In our study, we established both a rat and a human parathyroid organoid culture model to investigate the inhibitory effects of optical stimulation on PTH release. The expression of the optogene ChETA in the ex vivo rat parathyroid gland, and the robust reduction and recovery of the PTH after stimulation indicate the feasibility and efficacy of optogenetic regulation of ex vivo cultured rat parathyroid gland. A human parathyroid tissue organoid culture was also established, and we confirmed the inhibition of PTH in human hyperplastic parathyroid tissue. The light, which was controlled using the concentration of ionized calcium, induced immediate inhibition of PTH levels followed by a recovery. This optogenetic inhibition of the PTH secretion in our established organoid culture system thus provides a valuable approach to precisely regulate PTH release with space-revolved and time-revolved efficiency.

The SHPT rat model was used to evaluate the effect of optogenetic regulation in vivo. Compared with the monolayer cell culture and organoid culture of the parathyroid tissue, the release of PTH in vivo was influenced not only by the innervation/vascularization of the parathyroid gland, but also by the redundant capacity of the parathyroid gland. We observed a prompt decrease and recovery of PTH levels after illumination, and the quick PTH response and shorter recovery time in comparison with in vitro culture indicates the rapid regulatory function of blood vessels and nerve fibers, and the functional compensatory effects of the uninfected cells in the parathyroid gland. Interestingly, we also found that PTH fluctuations induced an immediate calcium response; serum calcium decreased significantly in the ChETA group 30 min after light stimulation. Since PTH is the most important humoral regulator of calcium and phosphorous homeostasis, our data suggests that the optogenetic regulation of the PTH release could also be used as an efficient approach to regulate systemic calcium homeostasis. Although parathyroid glands from both diet-induced SHPT rats and human SHPT patients were hyperplastic, this does not necessarily mean that these two models are the equivalent with respect to functional calcium sensing responsiveness.

Transplantation of human parathyroid tissue has been used clinically to restore the basal release of PTH after the parathyroidectomy. However, continuous elevated PTH levels after parathyroidectomy delays symptom improvement[11], and only PTH pulsatility (not calcium sensitivity) is restored after parathyroidectomy with transplantation[12]. To overcome these shortcomings, we established a nude-mouse model transplanted with human parathyroid tissue, and demonstrated that an optogenetic approach precisely controlled the release of human PTH from the transplanted parathyroid tissue. We found the viability and successful expression of the optogene in the transplanted human parathyroid tissue and the transplanted parathyroid tissue integrated well with the host tissue in nude mice as evidenced by the ingrowth of nerve fibers and vascular tissue. Quantification of human PTH in the serum of nude mice shows that illumination did indeed inhibit the secretion of human PTH, and the feasibility of combining light stimulation with calcium analysis facilitated the automatic control of PTH release based on levels of ionized calcium. Our study thus established an ideal physiological closed-loop regulatory method controlling the secretion of human PTH determined by serum $Ca^{2+}$ concentration.

A crucial role in regulating bone structure and in the remodeling process is played by PTH. Whilst intermittent PTH treatment significantly boosts bone formation and reduces fracture risks[28], chronic elevation of PTH in hyperparathyroidism nevertheless increases bone resorption and contributes to net bone loss over time[21]. Impaired mineralization of bone and significantly reduced number of osteoblasts are also recently observed after parathyroidectomy[29]. In our study, we investigated the chronic and long-term effects of optogenetic inhibition of PTH; our data demonstrates that, whilst human parathyroid transplantation decreased bone mineral density in nude mice, the optogenetic inhibition of PTH ameliorated impaired bone structure and arrested the continuous loss of the trabecular bone in hyperparathyroidism. Bone structure parameters, including BV/TV and TbN, together with trabecular bone morphology, reliably demonstrate that optogenetic inhibition of PTH partially attenuated bone loss induced by PTG transplantation. Regular inhibition of PTH secretion by optogenetic method can both enhance bone formation and inhibit bone resorption, which was demonstrated by ALP and TRAP staining and confirmed by the increased level of OPG and lower level of the rankl ligand. It is noteworthy that the contribution of pulsatile PTH secretion to the observed effects is unclear and other cell types infected by the lentivirus may contribute the observed effects.

In our study, both trabecular and cortical bone were affected during the development of secondary hyperparathyroidism, which is consistent with previous work[30]. However, after optogenetic inhibition of PTH secretion, we found an increase of BMD in trabecular bone, but not in cortical bone. This result is consistent with work by Ambrogini and colleagues, who found that parathyroidectomy treatment induces recovery of BMD in trabecular bone prior to cortical bone[31]. It is possible that a longer course of optogenetic treatment may induce an increase in cortical bone mass.

Both human and animal studies investigating PTH injection have found both trabecular and cortical bone to be affected[32–34]. It is likely that cortical and trabecular bone both respond to changes in PTH, even though there may be latency between observable changes in cortical and trabecular bone. The large number of bone cells conjugated in trabecular bone allows easier maintenance of BMD under the effect of PTH in the early stage. In addition, cortical bone serves as a mineral reservoir[35], which would be primarily be utilized by PTH to serve calcium requirements. Long-term effects of optogenetic treatment on the recovery of cortical bone mass require careful study.

The fate of bone marrow-derived mesenchymal stem cells can be directed by PTH to induce osteoblastic differentiation[36] and reduce adipogenetic differentiation[37]. The Wnt pathway is crucial in regulating the osteogenic differentiation of stem cells in the trabecular bone. In our study, we found that inhibition of PTH changed the distribution pattern of Wnt signals and that beta-catenin signals were concentrated in the active region of bone

formation after the optogenetic inhibition of PTH. These data suggest that optogenetic inhibition of PTH may modulate Wnt signaling pathways in mesenchymal stem cells to influence the bone remodeling process.

In summary, we have established an optogenetic approach bypassing CaSR to inhibit PTH secretion in both cell/organoid culture models and in situ/transplant animal models. Our data also demonstrates that light stimulation can be successfully controlled automatically using the level of ionized calcium to regulate human PTH secretion in transplanted hyperplastic parathyroid tissues from SHPT patients. The chronic inhibition of PTH induced by optogenetic treatment altered trabecular bone formation and influenced the bone remodeling process, which partially attenuated bone loss induced by PTG transplantation in mice. Our study thus provides a strategy to regulate the para-thyroid and restore human PTH release in secondary hyperpar-athyroidism through an optogenetic approach (Supplementary Fig. 17).

## Methods

**Ethics statement**. All experiments were approved by Experimental Animal Ethics Committee of Shenzhen Institutes of Advanced Technology, Chinese Academy of Sciences, and all experimental procedures involving animals were carried out in strict accordance with the animal use guidelines of the committee. The protocol number is SIAT-IRB-160909-NS-YANGF-A0237. All animals involved experi-ments were performed under full anesthesia, and every effort was made to mini-mize animal suffering. Studies with human participants were conducted in line with the Declaration of Helsinki. All tissue specimens were collected from August 2017 to February 2019, and the studies were approved by Ethics Committee of Shenzhen People's Hospital. The written consent obtained from patients was informed.

**Animals**. Adult male C57BL/6 mice (6–8 weeks old), male Sprague Dawley rats (8–10 weeks old) and male CD-1 nude mice (8–10 weeks old) were used in this study (Guangdong Medical Laboratory Animal Center, Guangzhou, China). Mice and rats were housed at 22–25 °C, 40–60% humidity, on a 12 h light/dark circadian cycle with ad libitum access to food and water. Animal care and experimental procedures were performed with approval by the Research Committee of the Shenzhen Institutes of Advanced Technology, Chinese Academy of Sciences.

**Primary culture of human parathyroid chief cells and organoid**. A total of ten glands from eight patients diagnosed with secondary hyperparathyroidism were used in the study (Table 1). The primary culture of human parathyroid chief cells and organoid were performed as previously described with minor modifications[38]. In brief, human parathyroid glands were obtained by parathyroidectomy from patients. For organoid culture, the glands were minced to 1 mm fragments in 37 °C RPMI 1640 medium and washed three times with the same medium, then suspended in culture medium (RPMI-1640, supplemented with 10% FBS, 1% glutamine, penicillin 100 U/ml, streptomycin 0.1 mg/ml). For primary cell culture, minced tissues were digested for 30 min at 37 °C in RPMI 1640 with 2 mg/ml collagenase I. The digest was transferred to 0.25% trypsin for 15 min at 37 °C for further digestion (Invitrogen, CA). After washing out the trypsin with medium, parathyroid cells were dispersed by gentle pipetting and filtered through a 100 mm mesh and suspended in culture medium. The cultures were incubated in a humidified atmosphere of 5% CO$_2$ at 37 °C for 5–7 days with fresh medium added every other day until 70% confluence was reached, at which point virus transfec-tion, immunohistochemistry or electrophysiology were performed. All glass cov-erslips were coated with poly-L-lysine for immunohistochemistry or electrophysiology.

**Fura-2-AM loading and Ca$^{2+}$ imaging**. For calcium imaging, Fura-2/AM was loaded to normal or ChETA-expressing chief cells as previous described with minor modifications[13]. Cells grown on glass coverslips were incubated with 2 mM Fura-2/AM at 37 °C for 30 min and then washed with medium and stored until experiments were performed. The coverslips were mounted into a perfusion chamber with imaging solution (138 mM NaCl, 5.6 mM, 1.2 mM MgCl$_2$, 5 mM HEPES-NaOH, and 0.5 mM CaCl$_2$) or PBS and was transferred onto the stage of a Carl Zeiss Axio Imager A2 fluorescence microscope. Calcium activity was recorded with a 20×/0.50 objective. For optogenetic stimulation, ChETA was excited by a manual controlled 470 nm LED device which fixed onto the microscope. For high extracellular calcium challenge, high calcium solution (138 mM NaCl, 5.6 mM, 1.2 mM MgCl$_2$, 5 mM HEPES-NaOH, and 2.6 mM CaCl$_2$) was applied to stimulate the cells. Fura-2 was excited using the 365/12 band pass filter (BP 365/12, Zeiss). Frames were collected using a cooled CCD camera (Axiocam506, Zeiss) at 1 Hz using ZEN 2.3 (Zeiss). Intracellular Ca$^{2+}$ changes were detected in single cells and

expressed as the fluorescence intensity ratios $\Delta F = F/F0$ with the same size back-ground fluorescence value $F0$, determined at every frame in each experiment by image J software.

**Immunostaining**. The immunostaining procedure was performed as previously described. Briefly, cells were fixed with 4% paraformaldehyde, blocked with 10% normal goat serum (Jackson ImmunoResearch, PA) in 0.3% PBS for 1 h and incubated with primary antibodies for 2 h at room temperature. Immunostaining in the current study was performed using the following primary antibodies: rabbit anti-Calcium sensory receptor (CaSR, 1:400; Abcam, MA), mouse anti-parathyroid hormone (PTH, 1:400; Abcam, MA), rat anti-vitamin D receptor (VDR, 1:400, Abcam, MA), Anti-beta III Tubulin antibody (Tuj-1, 1:200; Abcam, MA), Anti-alpha smooth muscle Actin antibody [1A4] (1:100; Abcam, MA). We employed Alexa Fluor@ 488 or 594-conjugated goat anti-rabbit, anti-mouse or anti-rat IgG antibodies (1:100; Jackson ImmunoResearch, PA) as the secondary antibodies. Nuclei were counterstained using Hoechst33342 or DAPI and then mounted for image acquisition.

The bone samples for immunochemistry staining were dehydrated by serial ethanol (70%, 2 h × 2; 95%, 2 h × 2; 100% 2 h × 2) and then cleared by xylene (2 h × 2) before being embedded into paraffin (Surgipath Paraplast, Leica). Samples were cut into 5–7 μm slices with a microtome (RM2235, Leica) and transferred to adhesion glass slices (Superfrost Plus Adhesion Slide, Thermo fisher). Slides were deparaffinized by washing the section slides in xylene (3 min ×2) and serial ethanol (100% 3 min ×2; 95 %, 3 min ×2; 70 %, 3 min ×2). Finally, the slides were rinsed and maintained in cold tap water until retrieval of antigens. The rehydrated bone slices were incubated with the following primary antibodies: Alkaline phosphatase (ALP, 1:500, abcam), Beta-catenin (1:500, Sigma) for 60 min. Sections were then washed and labeled with fluorescent secondary antibodies. The enzymatic detection process was performed with SP Rabbit & Mouse HRP Kit (005-000-121, CWBio). Slides were counterstained with hematoxylin (National diagnostics, Harris' Hematoxylin) for 20 min, washed and blued with Scott's tape water (1% MgSO$_4$, 0.067% NaHCO$_3$). Osteoclasts were located by TRAP kit (387A, Sigma) according to the kit instruction. The cell nuclei were labeled by counterstaining of Methyl Green Staining Solution (C0115, Beyotime). After the staining, bone slides were then washed in running water until no more dye washed out. Dehydration and clearance were performed with series ethanol (70%, 2 min ×2; 95%, 3 min ×2; 100%, 3 min ×2) and finally, xylene (2 min ×2). Slides were coverslipped with Eukitt® Quick-hardening (Sigma).

**Establishment of the rat model of secondary hyperparathyroidism**. The rat model of secondary hyperparathyroidism was established as previously described with minor modification[39]. Rats were divided into two groups based on the diet fed: (1) experimental group on low calcium and high phosphate diet (0.2% Ca, 1.2% P); (2) control on a normal phosphate and calcium diet (0.6% Ca, 0.6% P). All diets were customized and had the same constituents except for the calcium or phosphate contents: 20% protein and vitamin D 100 IU/100 g (Beijing Keao Xieli Feed Co. Ltd., Beijing, China). Rats were fed with the high phosphate/low calcium (HPLCa) diet for 7 days before lentivirus injection. This was followed by l5 days during which lentivirus was expressed whilst the HPLCa-diet was not changed. Light stimulation or parathyroid tissue isolation was performed after animals were treated with HPLCa-diet for 21 days. To test whether the diet-induced PTH increase is reversible, we removed the HPLCa-diet and replaced it with standard chow on the 21st day after the HPLCa-diet began. All rats were housed at 22–25 °C on a 12 h light/dark circadian cycle with ad-libitum access to food and water for 2 weeks. For serum biochemical determinations, blood samples were collected from anesthetized animals (intraperitoneal pentobarbital sodium) by aortic puncture. The immunohistochemisty and HE staining strategies were employed to confirm the parathyroid hyperplasia. Then, virus injection and light regulation were per-formed to modulate PTH secretion.

**Viral transfection of ChETA in primary cultured parathyroid cells and rat parathyroid glands**. ChETA or eYFP was expressed in chief cells and organoid by viral methods as previously described. The primary human chief cells were transfected with the Lentivirus (10$^9$ Tu/ml) carrying the ChETA-eYFP or eYFP gene in a serum-free medium for 24 h and replaced by fresh normal medium for another 48 h.

For rat parathyroid glands, injections were conducted with a 10 μl syringe connected to a 33-G needle (Neuros; Hamilton, Reno, USA), using a microsyringe pump (UMP3/Micro4, USA). Lentivius or AAV9-PTH-ChR2-eYFP-WPRE-pA (BrainVTA Co.Ltd, China) were delivered to surgically exposed rat parathyroid glands at 400 nl per site bilaterally. Expression of ChETA-eYFP or eYFP in the cells was monitored by observation of enhanced yellow fluorescent protein (eYFP) expression using fluorescence microscope.

**Light stimulation of human parathyroid cells and rat parathyroid gland**. The light stimulation of the cultured human parathyroid cells was performed after ChETA expression. Briefly, a homemade 470 nm blue light LED device was employed to deliver the light for 30 min at 20 Hz and 50 ms stimulus intervals. The arrangement of the lamps was perfectly matched with plate wells, and the average

output power of the LED was 1.65 mW/mm². Human chief cells, human parathyroid organoids, and rat parathyroid glands were plated in 6-well plates and were divided into three groups: control group; eYFP group, which expressed eYFP and was stimulated by blue light at 470 nm; and ChETA group, which expressed ChETA and was stimulated with blue light at 470 nm. At 0 h (before stimulus), and 1, 2, 4, 6, and 24 h after the light stimulation, the medium from each group was collected for PTH concentration determination using human or rat PTH ELISA kits (LS-F12, LS-F5548, LifeSpan BioSciences) according to the manufacturer's suggested protocol. To test the effect of optogenetic stimulation on PTH secretion of parathyroid cells in 1–3 patients with SHPT, PTH concentration was determined at 1, 2, 4, 6, and 24 h after light stimulation. To verify the inhibitory effects of optogenetic stimulation on PTH secretion at 1 h, PTH determination were conducted with cells from 2 to 8 patient to test whether this inhibition of PTH secretion within 1 h after light exposure can be replicated. Due to the insufficient collection of parathyroid tissue from patient 1, we used the data from patient 1 in both Fig. 2j and Supplementary Fig 3a. For a positive control, 100 and 1000 nM of Cinacalcet was employed to regulate PTH secretion of SHPT cells. Samples were collected at before, 1 and 4 h after the stimulation for biochemical determination.

For in situ rat parathyroid glands stimulation, the ChETA-expressing or eYFP-expressing parathyroid glands were surgically exposed and stimulated with blue light (470 nm, 20 Hz, 5 mW) using optical fiber for 30 min. Blood samples were collected at before, 5, 10, and 30 min after the light illumination for PTH and calcium tests in serum using rat PTH ELISA kits (LS-F5548, LifeSpan BioSciences) and Calcium Colormetric Assay (MAK022, Sigma-Aldrich) respectively.

**Human parathyroid organoid culture, transplantation, and optogenetic stimulation.** Human organoid culture was performed as previous described. Briefly, minced human parathyroid tissues (1 × 1 × 1 mm) were transferred into a 25 cm² flask containing 12 ml of culture medium and incubated at 37 °C in humidified 95% air and 5% $CO_2$. After culturing for an appropriate period (1 week), virus transfection was performed. After that, every 8th piece of human parathyroid tissue from different groups were transplanted to nude mice including control (without PTG transplantation), PTG (transplanted with no-transfected PTG) and ChETA (transplanted with ChETA-transfected PTG) groups. Seven days after transplantation, light stimulation (470 nm, 20 Hz, 9 mW) was conducted for 30 min per session. Light intensity was measured using a photometer and the light penetrance rate was calculated (Table 2). For short-term stimulation evaluation, blood samples were collected at different time points, including pre-transplantation, before, 5 and 15 min, 3 h and 9 h after stimulation. PTH concentration and serum calcium were assessed through human, rat or mouse PTH ELISA (LS-F12, LS-F5548 or LS-F5549, LifeSpan BioSciences) and Calcium Colormetric Assay (MAK022, Sigma-Aldrich) according to manufacturer instructions. For long-term stimulation evaluation, light stimulation was conducted three times per week (30 min per session) and bone samples were collected for bone metabolism tests at 28 days after the first stimulation. Serum OPG and RANKL were assessed at day 28 after the start of illumination using mouse ELISA kits: 1) OPG, abcam, ab203365; 2) RANKL, abcam, ab100749. The results were read by Nano Quant (Tecan, Infinite 200Pro).

**ELISA tests for signaling pathway.** Normal, ChETA-expressing and eYFP-expressing human chief cells were analyzed at different time points before or after light stimulation using commercial cell lysis solution (Beyotime Biotechnology, China) according to the manufacturer instructions. For positive controls to activate the signaling pathway, 10 μM Forskolin (F6886, Sigma-Aldrich) or 5 mg/ml PLA2 activator (sc-3034, Santa Cruz Biotechnology) was employed to activate the cAMP pathway or PLA2 pathway in SHPT cells. ELISA tests were performed using the following ELISA kits: Human 15-lipoxygenase ELISA Kit (MBS9301466, MyBio-Source, CA; EL001622HU, CUSABIO, China), Human 12-lipoxygenase ELISA Kit

(MBS7254285, MyBioSource, CA), Human Arachidonic Acid ELISA Kit (MBS281426, MyBioSource, CA), Human Anti Phospholipase A2 Antibody ELISA Kit (MBS752742, MyBioSource, CA), Human Protein Kinase A ELISA Kit (MBS034086, MyBioSource, CA) and Human Cyclic AMP Complete ELISA (ab133051, Abcam, MA). All experiments are performed according manufacturer instructions.

**Patch-clamp electrophysiology.** Whole-cell patch-clamp recordings were performed on cultured human or rat parathyroid cells. Recordings were obtained with Multiclamp 700B amplifiers (Molecular Devices) and was under visual guidance using a Nikon FN1 microscope. Whole-cell recordings were performed with borosilicate glass electrodes (0.69 mm OD, 5–7 MΩ) with internal solution containing 125 mM KCl, 1 mM $MgCl_2$, 10 mM EGTA, 25 mM KOH, and 5 mM HEPES, with pH adjusted to 7.4 with KOH. The normal bath solution contained 138 mM NaCl, 5.6 mM KCl, 1.2 mM $MgCl_2$, 5 mM HEPES, and 2.6 mM $CaCl_2$, with pH adjusted to 7.4 with NaOH, and perfused at 1 ml/min. To acquire low $[Ca^{2+}]$ bath solution, calcium concentration in normal solution was adjusted to 2.5 mM and was perfused at the same rate. All chemicals were obtained from Sigma-Aldrich. After forming a high-resistance seal (GΩ), the cell was held in current-clamp mode for 7–10 min until access resistance stabilized.

To illuminate ChETA-expressed human PTG cells, a DG-4 optical switch was used to deliver 20 Hz blue light (10 s) through the light path in microscope. A Digidata 1440A (Molecular Devices) data acquisition device was used to drive the DG-4 and digitize analog current signal at 10 kHz. Electrophysiological data were analyzed with pCLAMP 10.0 and Origin 7.5 (Origin Lab) software.

**Histological staining.** Parathyroid tissue and proximal tibia from different groups were isolated and tissue processing and sectioning was carried out respectively. In brief, the tissue samples were fixed in 4% PFA for 48 h at 4 °C and bone tissue was decalcified in 4% ethylenediaminetetraacetic acid (EDTA) for 30 days. The tissue was then embedded in paraffin and sectioned at 5 μm thickness. HE staining were performed separately on consecutive tissue sections and images were taken using a microscope (ECLIPSE 50i, Nikon, Japan).

**Micro-computed tomography (Micro-CT) analysis.** Mice femora and tibiae were collected and immersed into 4% paraformaldehyde before micro-CT scanning (model 1076, SkyScan, Kontich, Belgium, software Version 2.6). The scanning was performed according the following settings: isotropic voxel 11.53 μm, voltage 48 kV, current 179 μA, and exposure time 1800 ms. Three-dimensional (3D) reconstructions was conducted using the SkyScan NRecon software (version 1.6.8.0, SkyScan) with a voxel size of 8.66 μm. The datasets were reoriented using DataViewer (version 1.4.4.0, SkyScan), while the calculation of morphological parameters was carried out with the CTAn software (version 1.13.2.1, SkyScan). The 3-D reconstructed models were displayed by CTVol software (version 2.2.3.0, SkyScan).

The trabecular bone was selected 0.1–0.8 mm distal from the proximal tibia growth plate. The region of interested (ROI) was selected from 2D images slice-by-slice by hand to exclude the cortical area and then binarization of the images with a global thresholding of gray level (85–255) as mineralized tissue, according to the tuning 3D reconstruction of the mineralized tissue. A Gaussian filter (radius = 1) was used for 3D reconstruction. Quantitative analysis involved all the bone areas within the ROI of the 3D images. Morphometric parameters included total volume (TV, m³), bone volume (BV, m³), and bone fraction (BV/TV, m³). In addition, the bone mineral density (BMD, g/cm³) of the whole trabecular bone was calibrated using the attenuation coefficient of two hydroxyapatite phantoms with defined mineral densities of 0.25 and 0.75 g/cm³.

The cortical bone was selected 6.5–7.2 mm proximal from the distal femur growth plat. The ROI was selected from 2D images with a threshold (80–255) as mineralized tissue. A Gaussian filter (radius = 1) was used for 3D reconstruction. Quantitative analysis involved all the bone areas within the ROI of the 3D images. Morphometric parameters included BV and BMD (within selected bone area).

**RNA extraction, cDNA synthesis, and qPCR.** Bone marrow mesenchymal stem cells from nude mice were collected from control, PTG, and PTG ETA groups 4 weeks after light stimulation. Total RNA was isolated from cells using TRIzol Reagent (Invitrogen) according to manufacturer instructions. Synthesis of cDNA was performed using a ReverTra Ace qPCR RT kit according to the manufacturer protocol (FSQ-101; Toyobo Co.) and qRT-PCR was carried out using SYBRGreen (QPK-201; Toyobo Co.) with LightCyclerVR 480 (Roche, Switzerland). For Wnt pathway analysis, primers used for β-catenin were: 5′CCCTGAGACGCTAGA TGAGG3′, 5′TGTCAGCTCAGGAATTGCAC3′ (Final product 175 bp); WNT4: 5′ GAGAAGTGTGGCTGTGACCGG3′, 5′ATGTTGTCCGAGCATCCTGACC3′ (Final product 77 bp); Dkk1: 5′CTGAAGATGAGGAGTGCGGCTC3′, 5′ GGCTGTGGTCAGAGGGCATG3′ (Final product 183 bp); LRP6: 5′GGTGAACA GGGTGTTCCTGG3′, 5′TTCGCACCAGGTTGGCCATC3′ (Final product 197 bp); Fzd1: 5′CAGCAGTACAACGGCGAAC3′, 5′GTCCTCCTGATTCGTGT GGC3′ (Final product 138 bp); β-actin was used as internal control to normalize RNA content: 5′ GCTGTATTCCCCTCCATCGTG3′, 5′CACGGTTGGCCTTA GGGTTCAG3′ (Final product 265 bp). For PTH production and vesicle trafficking

---

## Table 2 Tissue penetrance of the blue-light emitting LED device.

| | Pre-skin light intensity (mW) | Post-skin light intensity (mW) | Penetrance rate (%) |
|---|---|---|---|
| 1 | 35.75 | 7.87 | 22 |
| 2 | 38.58 | 12.57 | 32.60 |
| 3 | 36.15 | 6.85 | 18.90 |
| 4 | 36.6 | 10 | 27.30 |
| 5 | 36.9 | 9.15 | 24.80 |
| 6 | 34 | 8.12 | 23.90 |
| 7 | 34.82 | 8.67 | 24.90 |
| 8 | 33 | 8.81 | 26.70 |
| 9 | 33.25 | 9.45 | 28.40 |

Light intensity was measured using the photometer (SANWA-LP1, Japan) above and below the skin, and the light penetrance rate was calculated as the percentage of light intensity.

analysis, primers used for *GCM2* were: 5′CGCTTCTTCTAGCTTCTGTCTC3′, 5′TGATGGCAACGCGATCTT3′ (Final product 89 bp); *MAFB*: 5′GGGTTC ATCTGCTGGTAGTT3′, 5′GCTCAGCACTCCGTGTAG3′ (Final product 114 bp); *PTH*: 5′CGTAAGAAGCTGCAG3′, 5′CACATCAGCTTTGTC3′ (Final product 153 bp); *Rab8a*: 5′TTGGATTCGGAACATTGAGG3′, 5′GCCTCTT GTCGTTCACATCAC3′ (Final product 86 bp); *Rab11a*: 5′GCAACAAGAAG CATCCAGG3′, 5′GCACCTACTGCTCCACGAT3′ (Final product 116 bp); *GAPDH* was used as internal control to normalize RNA content: 5′GCATGG CCTTCCGTGTTC3′, 5′CCTGCTTCACCACCTTCTTGAT′ (Final product 105 bp). Experiments were carried out according to the manufacturer instructions. The thermal conditions were: 1. 95 °C (3 min); 2. 95 °C (35 s), 60 °C (30 s), and 72 °C (30 s) for 40 cycles; 3. 4 °C (10 min). Expression levels for all the genes analyzed were normalized to β-actin, and the data were analyzed using $2^{-\Delta\Delta Ct}$ method.

**Calcium responding, PTH secretion control set up**. A calcium-responding, light-controlling system was set up to detect calcium ion changes and to control the on/off state of blue light. In detail, an electrode was immersed in the dish, which determined the calcium concentration in a real-time manner. At the same time, a homemade electronic microchip (and associated software) attached to the electrode compared the calcium concentration with the setting threshold. If the concentration was lower than the threshold, the light above the dish remained off and when the concentration exceeded the threshold, the software then controlled a connected LED device, which turned on to illuminate with blue light (470 nm).

For in vitro tests, the program-controlled calcium ion detector was immersed in RPMI-1640 medium in one well of a 6-well plate containing four pieces of ChETA/eYFP-expressing PTG organoid. To increase the calcium concentration, 1 ml high concentration calcium chloride (CaCl$_2$, 100 mg/ml, Sigma) solution was added to the well. Once the concentration reached the threshold (50 mg/l), the monitoring program turned the light on. PTH levels in the ChETA and eYFP groups were determined after light stimulation.

For in vivo tests, the established system was upgraded by combining with a microdialysis system. The mice were anesthetized by an intraperitoneal (i.p.) injection of pentobarbital (80 mg/kg) before probe implantation surgery and remained anesthetized throughout the experimental period with a rodent anesthesia machine (R520IP, RWD, China). Then, the neck hairs were cut and the blood microdialysis probes (MD-2310, IV-10, BASi, USA) were positioned within the jugular vein. The probe was perfused with 30% ethanol calcium-free physiological saline at a rate of 2.5 µl/min using a microinjection pump (MD-1000-B, BASi) and micro syringe (MDN-0100, BASi) for 1 h to reach baseline values. Blood dialysates were collected in one well of a 12-well plate, which was filled with 1.5 ml saline and the calcium detector electrode. The calcium ion concentration was read and replaced with fresh saline every 10 min. The resting value of calcium concentration was set as the threshold, and light stimulation was triggered by the detection of the elevated calcium ion. For calcium challenge, CaCl$_2$ (30 mM, 0.5 ml) was delivered by i.p. injection. PTH levels and serum calcium levels were determined after light stimulation in ChETA and eYFP groups.

**Bone histomorphometry**. Calcein (25 mg/kg; Sigma) was injected (i.p.) at 1 and 7 days before mice were sacrificed, to provide a double fluorochrome label in three groups: control (sham), eYFP-expressing parathyroid organoids transplanted + light and ETA-expressing parathyroid organoids-transplanted + light. After mice were sacrificed, tibiae were removed of soft tissue, fixed in 70% ethanol for 48 h, then dehydrated through a graded series of alcohols: 80% ethanol, 90% ethanol, then three changes of 100% ethanol for 24 h each. Tibiae were then cleared in chloroform for 24 h, placed for a further 24 h in 100% ethanol, and embedded without decalcification in optimal cutting temperature compound (4583, Sakura Finetek USA, Inc., CA) for frozen sections. For each tibia, four 40 µm sections were cut and mounted in aqueous Fluoromount-G (P36930, Fisher Scientific, PA). Calcein levels on the surface of bone were analyzed using a Zeiss Axio Imager A2 fluorescent microscopy (Carl Zeiss, Germany) with an excitation wavelength of 485 nm and emission wavelength of 510 nm. Mineral apposition rate (MAR, MAR = interlabel width/labeling period) in cortical bone was determined using Zen softwares 2.3 (Carl Zeiss, Germany). The bone formation rate was calculated according to the following formula: BFR = MAR × (MS/BS), where MS is the mineralizing surface and BS is the bone surface[40]. Adobe Photoshop cc2015 (Adobe Systems, San Jose, CA) is used to organize figures.

**Statistical analyses**. All the experiments were repeated at least three times, and similar results were obtained. All statistics were performed in Graph Pad Prism 7.0 (GraphPad Software, Inc.), unless otherwise indicated. Paired student tests, unpaired student tests, one-way ANOVAs and two-way ANOVAs were used where appropriate. Bonferroni post hoc comparisons and Tukey's tests were conducted to detect significant main effects or interactions. In all statistical measures, a *p* value <0.05 was considered statistically significant. Post hoc significance values were set as *$p < 0.05$, **$p < 0.01$, ***$p < 0.001$.

**Reporting summary**. Further information on research design is available in the Nature Research Reporting Summary linked to this article.

## Data availability
All data generated in this study that support our findings are presented within this paper, its Supplementary Information, or in the source data. All additional information will be made available upon reasonable request to the corresponding author. Source data are provided with this paper.

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

## Acknowledgements

The authors thank Chenghua Mao, Zhongkai Xiao, Bingfeng Liu and Ningning Li for their expert technical assistance. This project was partly supported by the National Natural Science Foundation of China (31800881 to Y.L., 82072489 to F.Y.); Key Research Program of Frontier Sciences of Chinese Academy of Sciences (QYZDB-SSW-SMC056 to F.Y.); Shenzhen Governmental Basic Research Grant (JCYJ20180507182301299 to F.Y.); Shenzhen Key Laboratory of Kidney Diseases (ZDSYS201504301616234 to X.Z.).

## Author contributions

F.Y. and Y.L. designed the experiments, analyzed data and wrote the manuscript. Y.L., L.Z., N.H., J.S., and D.Y. performed the experiments and analyzed the data. S.H., C.R., L.W., W.L., and X.Z. contributed to the design, analysis, revision, and interpretation of data. F.Y. and X.Z. conceived and supervised the project. All authors contributed to the final version of the manuscript.

## Competing interests

The authors declare no competing interests.

## Additional information

**Peer review information** *Nature Communications* thanks James Koh and the other anonymous reviewer(s) for their contribution to the peer review this work. Peer reviewer reports are available.

