## [Peer Review File · Nature Communications]

Reviewers' Comments:

Reviewer #1:

Remarks to the Author:

The authors show that cells from the parathyroid glands of patients with what they call secondary hyperparathyroidism do not respond in vitro to calcium with changes in intracellular calcium or suppression of PTH secretion. Then, activation of a mutant rhodopsin sensitive to blue light for 30 minutes, after transfection into parathyroid cells, can depolarize cells from patients with secondary hyperparathyroidism and lead to an increase in PTH secretion for minutes to hours in vitro. Repeated light exposures led to suppression of intracellular protein kinase A activity and cAMP levels, as well as increase in PLA2 levels, arachidonic acid levels and 12-LO and 15-LO levels transiently. When rat parathyroids were infected with lentivirus expressing the mutant rhodopsin, and the infected glands were cultured in vitro, light exposure led to inhibition of PTH secretion for several hours in vitro. Rats with secondary parathyroid hyperplasia had the rhodopsin virus injected into parathyroid cells and, after light exposure, PTH level fell for 15 minutes and calcium levels fell after 30 minutes for an unspecified time, but presumably short. Then, using a system that senses ambient calcium to trigger light administration, human parathyroid cells expressing mutant rhodopsin led to a transient fall in PTH levels in response to light. After transplantation of human cells into nude mice, calcium administration led to turning on of blue light and transient suppression of PTH and calcium levels. When light was shown on these mice every other day for 28 days, mice without mutant rhodopsin lost bone mass. With light-stimulated suppression of PTH for 15 minutes every other day, cortical bone mass fell in all mice, but in those with the light sensor in the parathyroid, trabecular bone loss was more modest, mineral apposition rate was partly restored, RANKL in blood was lowered to that in controls, and OPG levels were increased.

The use of a light sensor to depolarize parathyroid cells is a novel idea and the evidence that, after shining light, PTH secretion is suppressed in association with a number of intracellular changes is of interest. I have a number of concerns in data description and interpretation, however.

1. Introduction: The authors misquote reference 1; there is no evidence that Tip 39 or the PTH2 receptor have anything to do with calcium or bone metabolism.
2. The sentence, "The current situation is that a method of precise regulation of rhythmic PTH secretion in both in situ and transplanted human PTG tissues, especially secretion based on serum calcium concentration, is lacking." is somewhat misleading. The trouble with transplanted human tissue has nothing to do with it being non-rhythmic; the only problem is if the levels of PTH are sustained at too high a level. In normal humans, there is no evidence that the mild rhythmicity of PTH levels is important...only that the levels be normal.
3. The description of the patients in Table 1 needs expansion. I am guessing that these patients with "secondary" hyperparathyroidism had glands removed because of hypercalcemia in the setting of renal failure. Is that right? If so, it would be useful to specify the blood levels of calcium. These glands probably are autonomous and have been described as "tertiary" hyperparathyroidism. That should be clarified.
4. Figure 1c. Controls for antibody specificity should be shown.
5. The authors need to show that their protocol of increasing extracellular calcium is capable of lowering PTH levels in normal parathyroid cells. Otherwise the failure to do so here cannot be interpreted (need for a positive control). They can do that with normal rat parathyroid cells, such as those used in later experiments. But they need a positive control for the experiment that shows that calcium elevation fails to lower PTH levels.
6. The authors don't quote reference 5 sufficiently: the Calcium sensing receptor increases intracellular calcium levels both through release from intracellular stores AND from opening plasma membrane calcium channels.
7. The data in Figure 3 f-k all appear to be "normalized with 1.0 being some central number. I think that NONE OF THE DATA IN THE PAPER SHOULD INVOLVE NORMALIZATION. Please give real numbers with appropriate units and no normalization to anything.
8. The authors have much more faith in the idea that we understand the regulation of PTH secretion than I do. For example, while there is a case that PLA2-arachidonic acid pathways may contribute to regulation of PTH secretion, the claim, "We know that the PLA2-arachidonic acid (AA) intracellular signaling pathway mediates the inhibition of PTH secretion¹⁹," quotes a very old paper that suggests that this pathway may contribute, but tremendously overstates what we

"know". The authors should simply state that this pathway may contribute to regulation of PTH secretion.

9. Figure 4. Since this study involves normal parathyroids, the authors should compare the effects of light to the effects of raising calcium in the medium.

10. Neither the text, the figure legends or the Methods section explains how the rats were made to develop secondary hyperparathyroidism and how that was assessed in the rats.

11. Figure 5 f, g, h: Please no ratios. Give real data with units.

12. I found the description of the experiment in Figure 6 confusing. Does the method in 6e involve measurement of blood calcium? If so, how? And how is this information conveyed to the mysterious light machine? What is the box in the lower right of Figure 6e doing.

13. Again, please no normalization for 6 h and 6i. And I'm confused about what the $CaCl_2$ arrow signifies. And again, no normalization in 6h and i...give real data.

14. Figure 7 is impressive and very surprising. I want to be sure I understand the data. I think the authors show that a slight decrease in PTH levels for 15 minutes a day is enough to change trabecular bone mass over 30 days. Is that right? Is the light stimulation given for a full 24 hours every other day? Were PTH levels checked at various times during the day to see if PTH levels ever fall again? In 7b, I'm not sure what the blue line under "light" is meant to signify. Does it mean that light was on for 24 hours?

15. The experiment in Figure 7 doesn't show calcium measurements, only PTH for 15 minutes. I'd like to see more PTH levels and definitely some evidence that blood calcium is affected (unless it isn't).

16. Please no normalization in Figure 7b.

17. Figure 7 k and L. Please make clear when blood was drawn with respect to when the light was turned on and PTH levels transiently slightly fall.

18. Discussion. The authors state, "The immediate decrease of PKA and cAMP after stimulation indicates that the synthesis of PTH was reversibly inhibited by optogenetic treatment; the light-induced elevation and recovery of the level of PLA2, arachidonic acid, 12-LO and 15-LO all suggest that light stimulation can effectively inhibit the release of PTH." Actually, I don't know of any data showing effects of PKA and cAMP on synthesis of PTH. Lowering of PKA activity by lowering cAMP probably decreases secretion of PTH. Calcium, by mechanisms not completely understood, decreases the destruction of PTH mRNA, but exactly how that happens and whether that involves cAMP isn't settled, I believe (see studies of Many and Silver).

19. The authors exaggerate the effect of light in the experiment in Figure 7. In the discussion, they say, "our data demonstrated that, while human parathyroid transplantation decreased bone mineral density of nude mice, the rhythmic inhibition of PTH increased bone mineral density and improved the structure of trabecular bone." That's not shown. What the authors show is that bone loss still occurs after light administration, just not as much as in the absence of light. Light makes the bone better than without light, but does not completely reverse the effects of hyperparathyroidism. To make that claim, the authors would need to compare, for example, the control and ChETA bone density measurements. This experiment appears to be very underpowered to test that question.

Reviewer #2:

Remarks to the Author:

Liu and colleagues describe a new approach to regulate PTH secretion from parathyroid cells using an optogenetic approach. Using the light-activated channelrhodopsin variant ChETA, they infect dispersed human primary parathyroid cells, rat glands *in vivo*, and human parathyroid tissue pieces using lentiviral constructs. They show that membrane depolarization inhibits PTH synthesis and secretion with their system. Using a rat model of secondary hyperparathyroidism, they show a decrease in circulating PTH and calcium. And using transduced human parathyroid pieces, transplanted under the skin of nude mice, they expose the mice with intermittent blue light and show an increase in bone density compared to nude mice carrying pieces of human parathyroid glands without light stimulation.

The authors also engineer a system *in vitro* and *in vivo*, which allows light stimulation automatically if culture medium or serum calcium, respectively, increases to a certain concentration.

Comments:

Lentivirus is known to have low efficacy in infecting organs in vivo. How do the authors reconcile their findings with this observation?

Even if a large portion of a parathyroid gland can be infected and their PTH secretion lowered, the remaining uninfected cells will compensate for this. How do the authors explain the in vivo effects they observe?

Introduction: PTH2 does not have a role in calcium-phosphate and bone metabolism.

Low calcium levels do not activate the calcium-sensing receptor.

How was the rat model of secondary hyperparathyroidism established?

Fig 3: how was "PKA" measured?

Figures 5, 6, and 7: what is "PTH ratio"? Please show the absolute PTH values. Same with calcium values.

Reviewer #3:

Remarks to the Author:

The manuscript 'An Optogenetic Approach for Regulating Human Parathyroid Hormone Secretion' describes an optogenetic approach to regulating PTH in models of secondary hyperparathyroidism. This approach bypasses CaSR signaling to activate similar downstream effectors resulting in a reduction in PTH production. While the approach is novel in regards to the application of optogenetics, I have some concerns regarding the methods and controls.

1) The methods surrounding in vivo optical stimulation are lacking. There is no mention of the light source or intensity in in vivo experiments. Blue light has very poor tissue penetrance with the in vivo stimulation its important to show sufficient light is actually reaching the implanted organoid and parathyroid gland. Again, here more experimental details would be helpful in the interpretation of the results.

2) There are a number of experiments throughout the manuscript that lack important positive controls that would strengthen the author's findings.

- In the first figure, the authors demonstrate that parathyroid cells have an attenuated/ limited response changes in extracellular calcium. Two positive controls would be useful/necessary here.

1) Normal (non-SHPT) parathyroid cells that respond to changes in extracellular calcium demonstrating what a normal response looks like. 2) Perfusion of a drug or compound that elicits a 'normal' response from the SHPT cells, to demonstrate these cells are healthy and otherwise capable of responding to other types of stimuli

-Extended Figure 2 - Normal non SHPT cell response

-Figure 3 F-K - + control for the Elissa assays - example - F and G Forskolin -> increase cAMP.

3) How does optogenetic stimulation of these cells compare to a normal response to changes in extracellular calcium? Can the optogenetic stimulation be tailored to replicate the response to specific concentrations of extracellular calcium in normal non-SHPT cells?

Minor

1) There are too many abbreviations throughout the paper, which makes it difficult to read.

Removing some of the less used abbreviations would improve the readability of the manuscript. Often abbreviations are redefined multiple times.

2) Figure 3a and b doesn't really show much.

3) Figure 3- How are the cells being ontogenetically stimulated along with Fura2. 450-495nm is within the excitation range of Fura2.

4) It is unclear from the methods how Fura2 emissions are being represented. Fura2 is ratio metric and only 365 is being measured?

5) Figure 3C and D X axis units?

6) Figure 4J and K what is S?

7) Methods lacking on how the rat SHPT model was developed, rat strain, mouse strain and vendor information.

- 8) how long were cells stored after incubation with fura 2?
- 9) Methods for calcium imaging – what is the LED device?
- 10) What was the source and sterotype of the AAVs?
- 11) Why 20Hz stimulation?
- 12) Calcium Monitoring using electrode and blue light stimulation – Methods need more details on how the system was programed to work automatically? What hardware was used to monitor calcium levels and deliver light?

Response to Reviewer 1:

The authors show that cells from the parathyroid glands of patients with what they call secondary hyperparathyroidism do not respond in vitro to calcium with changes in intracellular calcium or suppression of PTH secretion. Then, activation of a mutant rhodopsin sensitive to blue light for 30 minutes, after transfection into parathyroid cells, can depolarize cells from patients with secondary hyperparathyroidism and lead to an increase in PTH secretion for minutes to hours in vitro. Repeated light exposures led to suppression of intracellular protein kinase A activity and cAMP levels, as well as increase in PLA2 levels, arachidonic acid levels and 12-LO and 15-LO levels transiently. When rat parathyroids were infected with lentivirus expressing the mutant rhodopsin, and the infected glands were cultured in vitro, light exposure led to inhibition of PTH secretion for several hours in vitro. Rats with secondary parathyroid hyperplasia had the rhodopsin virus injected into parathyroid cells and, after light exposure, PTH level fell for 15 minutes and calcium levels fell after 30 minutes for an unspecified time, but presumably short. Then, using a system that senses ambient calcium to trigger light administration, human parathyroid cells expressing mutant rhodopsin led to a transient fall in PTH levels in response to light. After transplantation of human cells into nude mice, calcium administration led to turning on of blue light and transient suppression of PTH and calcium levels. When light was shown on these mice every other day for 28 days, mice without mutant rhodopsin lost bone mass. With light-stimulated suppression of PTH for 15 minutes every other day, cortical bone mass fell in all mice, but in those with the light sensor in the parathyroid, trabecular bone loss was more modest, mineral apposition rate was partly restored, RANKL in blood was lowered to that in controls, and OPG levels were increased.

The use of a light sensor to depolarize parathyroid cells is a novel idea and the evidence that, after shining light, PTH secretion is suppressed in association with a number of intracellular changes is of interest. I have a number of concerns in data description and interpretation, however.

Response: We thank the reviewer for the summary of our findings and also very much appreciate the reviewer's comments on the manuscript. Based on these suggestions, we have performed new experiments and added new data to the revised manuscript. We address the reviewer's concerns point-by-point in the following text. We hope that these responses fully satisfy the reviewer's concerns. The revised manuscript has been improved based on the reviewer's suggestions.

1. Introduction: *The authors misquote reference 1; there is no evidence that Tip 39 or the PTH2 receptor have anything to do with calcium or bone metabolism.*

Response: Thanks for the comments, we agree that PTH2 does not play a significant role in calcium-phosphate and bone metabolism. We have deleted the reference according to your suggestion.

- On page 4, lines 3- 5 (Introduction):

The PTH peptide family include the PTH and the PTH-related peptide (PTHrP), both which play important roles in maintaining homeostasis of calcium-phosphate and bone metabolism.

2. The sentence, “The current situation is that a method of precise regulation of rhythmic PTH secretion in both in situ and transplanted human PTG tissues, especially secretion based on serum calcium concentration, is lacking.” is somewhat misleading. The trouble with transplanted human tissue has nothing to do with it being non-rhythmic; the only problem is if the levels of PTH are sustained at too high a level. In normal humans, there is no evidence that the mild rhythmicity of PTH levels is important…only that the levels be normal.

Response: Thanks for the comments and we agree. The most important problem of transplanted human tissue is that PTH levels are sustained too high. Recent studies demonstrate that normal human PTH secretion have rhythms with a surge in secretion at 4:00am and also show small fluctuation during the whole day. However, up to now, there is still a lack of direct evidence demonstrating that rhythmic PTH secretion correlates with the bone metabolism.

According to your suggestion, we have rewritten the sentence to make this clear.

- On page 5, lines 7- 9 (Introduction):

The current situation is that a method of precise inhibition of PTH secretion in both in situ and transplanted human PTG tissues, especially secretion based on serum calcium concentration, is lacking.

3. The description of the patients in Table 1 needs expansion. I am guessing that these patients with “secondary” hyperparathyroidism had glands removed because of hypercalcemia in the setting of renal failure. Is that right? If so, it would be useful to specify the blood levels of calcium. These glands probably are autonomous and have been described as “tertiary” hyperparathyroidism. That should be clarified.

Response: Thanks for the suggestion. We have expanded the Table 1 with serum calcium, creatinine, and glomerular filtration rates for all the patients. The PTH, creatinine and glomerular filtration rate data demonstrate that all patients suffered from chronic kidney disease and developed secondary hyperparathyroidism. Serum calcium levels were slightly increased and most of the patients had received hemodialysis for a long time period.

- On page 46, lines 4- 6 (Figure legends):

Table 1: Sex, age, serum PTH level, serum calcium level and serum creatinine level in eight patients diagnosed with secondary hyperparathyroidism (SHPT).

4. Figure 1c. Controls for antibody specificity should be shown.

Response: Thanks for the comments. As the reviewer suggested, we performed the antibody specificity tests. By reproducing the immunostaining procedure as before, we replaced the first antibody with PBS and found that no non-specific fluorescence could be detected, which demonstrated that these antibodies worked well (Extended Data, Figure 1c). We further tested antibody specificity on rat parathyroid and adjacent thyroid tissue. We found positive signals on parathyroid tissue, however there are few signals in adjacent thyroid tissue (Extended Data, Figure 1d), suggesting that the used antibody were specific for the parathyroid cells.

Besides our experimental data, all the antibodies were also validated for use in mouse/rat/human tissues based on previous publications or preliminary tests: anti-PTH (Abcam Cat# ab14493, RRID:AB_301271), anti-CaSR (Abcam Cat# ab18200, RRID:AB_725798), anti-VDR (Abcam Cat# ab115495, RRID:AB_10903196).

- On page 7, lines 8-11 (results):

Immunofluorescent staining of the cultured cells revealed that most of the isolated cells expressed parathyroid hormone (PTH), calcium-sensing receptors (CaSR) and Vitamin D receptors (VDR) (Fig. 1c), and there were few signals in the negative control group and the adjacent thyroid gland tissue (Extended Data Fig. 1c, d).

- On page 48, lines 8-12 (Figure legends):

c, Negative control of immunostaining of parathyroid hormone (PTH), calcium sensing receptor (CaSR) and Vitamin D receptor (VDR) on representative cultured human parathyroid cells; scale bar=50 μ m. d, Immunostaining of parathyroid hormone (PTH) on rat parathyroid gland and thyroid gland tissue.

5. The authors need to show that their protocol of increasing extracellular calcium is capable of lowering PTH levels in normal parathyroid cells. Otherwise the failure to do so here cannot be interpreted (need for a positive control). They can do that with normal rat parathyroid cells, such as those used in later experiments. But they need a positive control for the experiment that shows that calcium elevation fails to lower PTH levels.

Response: Thanks for the comments and we fully agree. As suggested, we have now isolated and cultured normal rat parathyroid cells, then performed new experiments to examine the effects of extracellular calcium challenge on

membrane potential, intracellular calcium and PTH secretion (Extended Data, Figure 2). Our results showed that an increasing extracellular calcium challenge could indeed induce a significant decrease of PTH secretion from normal rat PTG tissues (Extended Data, Figure 2g). Membrane potential of normal rat parathyroid chief cells also fluctuated during an increase of extracellular Ca^{2+} from 0.5 mM to 2.6 mM (Extended Data, Figure 2b); normal rat parathyroid cells were sensitive to extracellular Ca^{2+} and showed fluctuations of fluorescence during extracellular Ca^{2+} changes (Extended Data, Figure 2e). Taken together, these new data demonstrate that increasing extracellular calcium led to lower PTH levels in normal parathyroid cells compared with the SHPT cells.

- On page 8, lines 17 -page 9, line 7 (results):

In order to demonstrate the normal response to extracellular calcium, we also isolated and cultured normal rat parathyroid cells (Extended Data Fig. 2a). Patch-clamp recordings showed that the membrane potential of normal rat parathyroid chief cells fluctuated during an increase of extracellular Ca^{2+} from 0.5 mM to 2.6 mM (Extended Data Fig. 2b). Of 10 recorded cells, eight cells were responsive to high extracellular calcium and showed fluctuations of the membrane potential (Extended Data Fig. 2c). Calcium imaging and quantification showed that normal rat parathyroid cells were sensitive to extracellular Ca^{2+} and showed fluctuations of fluorescence during extracellular Ca^{2+} changes (Extended Data Fig. 2e, f). Statistics revealed that 12 of 12 cells were responsive to extracellular Ca^{2+} , showing fluctuations in fluorescence (Extended Data Fig. 2d). Our data also demonstrate that increasing extracellular calcium is capable of lowering PTH levels in normal rat parathyroid cells (Extended Data Fig. 2g)

6. The authors don't quote reference 5 sufficiently: the Calcium sensing receptor increases intracellular calcium levels both through release from intracellular stores AND from opening plasma membrane calcium channels.

Response: Thanks for the comments. We have rewritten this sentence as suggested.

- On page 8, lines 2-4 (results):

Intracellular calcium levels can be increased by CaSR through the release of Ca^{2+} from the endoplasmic reticulum and the opening of plasma membrane calcium channels to inhibit PTH secretion.

7. The data in Figure 3 f-k all appear to be “normalized with 1.0 being some central number. I think that NONE OF THE DATA IN THE PAPER SHOULD INVOLVE NORMALIZATION. Please give real numbers with appropriate units and no normalization to anything.

Response: Thanks for the comments and we fully agree. We have provided the non-normalized data with real numbers and appropriate units. Please refer to the new Figure 3f-k. (page 34-page 35)

8. The authors have much more faith in the idea that we understand the regulation of PTH secretion than I do. For example, while there is a case that PLA2-arachidonic acid pathways may contribute to regulation of PTH secretion, the claim, “We know that the PLA2-arachidonic acid (AA) intracellular signaling pathway mediates the inhibition of PTH secretion¹⁹,” quotes a very old paper that suggests that this pathway may contribute, but tremendously overstates what we “know”. The authors should simply state that this pathway may contribute to regulation of PTH secretion.

Response: Thanks for the suggestion and we totally agree. We have rewritten this sentence more precisely according to the reviewer's suggestion

- On page 12, lines 17-19 (results):

Next, we investigated the molecular pathway regulating the PTH secretion process. Because the PLA2-arachidonic acid (AA) intracellular signaling pathway may contribute to regulation of PTH secretion,...

9. Figure 4. Since this study involves normal parathyroids, the authors should compare the effects of light to the effects of raising calcium in the medium.

Response: Thanks for the suggestion. We have performed new experiments to compare the effects of lights to the effects of raising calcium using normal rat parathyroid cells (Extended Data, Figure 5).

Our data shows that gradually increased extracellular Ca^{2+} levels significantly inhibited PTH secretion from parathyroid cells (Extended Data, Figure 5a). PTH level ($2.63 \pm 0.14 \text{ pg/ml}$) at 2.5 mM extracellular Ca^{2+} was significantly lower than the PTH level ($3.20 \pm 0.11 \text{ pg/ml}$) at 0.5 mM. Thirty minutes of optical stimulation also significantly inhibited PTH secretion from $3.30 \pm 0.22 \text{ pg/ml}$ to $2.56 \pm 0.14 \text{ pg/ml}$ (Extended Data, Figure 5b), suggesting a comparable effect between optical stimulation and changing extracellular calcium level. Taken together, these data support that both changing extracellular calcium and optogenetic stimulation can induce a decrease in PTH secretion from normal parathyroid cells.

- On page 14, lines 20 - page 15, line 13 (Results):

Since extracellular calcium and optogenetic stimulation both successfully induced decreased PTH levels in normal parathyroid cells, we compared the inhibitory effects of light stimulation and raising extracellular calcium on PTH level. Quantification of the level of PTH secreted from rat parathyroid cells was performed at 0.5 mM, 1.5mM and 2.5 mM extracellular Ca^{2+} respectively, and we found that the gradually increased extracellular Ca^{2+} level significantly inhibited PTH secretion from parathyroid cells (Extended Data Fig. 5a). The level of PTH at 2.5 mM extracellular Ca^{2+} was significantly lower than the PTH at 0.5 mM and 1.5 mM Ca^{2+} . Next, we investigated the effects of light stimulation duration and frequency on inhibition of secreted PTH and found that increased light stimulation duration significantly enhanced the

inhibitory effects on PTH levels. Blue light stimulation (30 min at 20 Hz) induced a significantly lower PTH level than 15 min of stimulation did, however there was no significant differences in PTH levels between 30 min and 1 h of light stimulation (Extended Data Fig. 5b). It is interesting to note that PTH levels were comparable between 10 Hz, 20 Hz and 30 Hz of light stimulation, suggesting that changing light frequency may not enhance the inhibitory effects on PTH (Extended Data Fig. 5c).

10. Neither the text, the figure legends nor the Methods section explains how the rats were made to develop secondary hyperparathyroidism and how that was assessed in the rats.

Response: Thanks for pointing this out. We have added our detailed protocol for establishing the rat model of secondary hyperparathyroidism in the Methods section.

- On page 66, lines 6- 17 (Methods):

The rat model of secondary hyperparathyroidism was established as previously described with minor modification³². Rats were divided into two groups based on the diet fed: (1) experimental group on low calcium and high phosphate diet (0.2% Ca, 1.2 % P); (2) control on a normal phosphate and calcium diet (0.6% Ca, 0.6% P). All diets were customized and had the same constituents except for the calcium or phosphate contents: 20% protein and vitamin D 100 IU/100 g (Beijing Keao Xieli Feed Co. Ltd., Beijing, China). All rats were housed at 22-25 °C on a 12-hour light/dark circadian cycle with ad-libitum access to food and water for two weeks. For serum biochemical determinations, blood samples were collected from anesthetized animals (intraperitoneal pentobarbital sodium) by aortic puncture. The immunohistochemistry and HE staining strategies were employed to confirm the parathyroid hyperplasia. Then, virus injection and light regulation were performed to modulate PTH secretion.

11. Figure 5 f, g, h: Please no ratios. Give real data with units.

Response: Thanks for the comments. We have provided the non-normalized data with real numbers and appropriate units. Please refer to the new Figure 5f-h. (Page 39)

12. I found the description of the experiment in Figure 6 confusing. Does the method in 6e involve measurement of blood calcium? If so, how? And how is this information conveyed to the mysterious light machine? What is the box in the lower right of Figure 6e doing.

Response: Thanks you these comments. In Figure 6, we established a human parathyroid organoid culture model and transplantation model, in order to regulate *ex vivo* and *in vivo* PTH secretion using optogenetics based on ionized calcium concentration.

In *ex vivo* human parathyroid organoid culture model (Figure 6c), we immersed an electrode in the dish, which measured calcium concentration in real time. At the same time, a microchip in the box also compared the calcium

concentration we have read with the setting threshold. If the read mean was lower than the threshold, the light above the dish was 'off'. When calcium concentration exceeded the setting threshold, the chip's operating program sent a command to the light source, and the light was turned on.

For our *in vivo* parathyroid tissue transplantation model, automatic control of PTH secretion using optogenetics was achieved by incorporating the microdialysis system (lower right of Figure 6e). We combined the calcium controlled light stimulation system, which we used in 6c, with a microdialysis system. We embedded a microdialysis probe within the jugular vein in mice. The pump (right, bottom) pushed artificial cerebrospinal fluid (aCSF) into the probe and the dialysate came out of the probe into the dish. The calcium concentration in the dish was determined by calcium monitoring system. Once the calcium concentration in the dialysate reached the threshold, the blue light was turned on.

13. Again, please no normalization for 6 h and 6i. And I'm confused about what the CaCl_2 arrow signifies. And again, no normalization in 6h and i... give real data.

Response: Thanks for the comments and we apologize. We have provided the non-normalized data with real numbers and appropriate units. Please refer to the new Figure 6h-I (page 39).

The CaCl_2 arrow signifies that CaCl_2 injection was performed from the mouse tail vein to elevate serum calcium. When the elevated serum calcium concentration reached the threshold, the blue light was automatically turned on. PTH levels were inhibited in the ChETA group compared to the control group at 5 and 15 min after the 30 min of light stimulation.

14. Figure 7 is impressive and very surprising. I want to be sure I understand the data. I think the authors show that a slight decrease in PTH levels for 15 minutes a day is enough to change trabecular bone mass over 30 days. Is that right? Is the light stimulation given for a full 24 hours every other day? Were PTH levels checked at various times during the day to see if PTH levels ever fall again? In 7b, I'm not sure what the blue line under "light" is meant to signify. Does it mean that light was on for 24 hours?

Response: Thanks for the comments. To clarify, we stimulated these transplanted nude mice with blue light for 30 min every other day for 4 weeks - light stimulation was *not* given for a full 24 hours every other day.

We found that human PTH levels in the serum of nude mice were significantly lower in the ChETA group at 5 and 10 min after blue light stimulation than in the control PTG group. According to your suggestion/thoughts, we also checked PTH levels at 3 and 9 h after stimulation and found that there was no significant difference in PTH levels between ChETA group and PTH group at either of these time periods (Extended Data Fig. 8c).

In Figure 7b, the blue line under “light” was intended to signify light stimulation for 30 min in each group every other day, not for 24 hours.

15. The experiment in Figure 7 doesn't show calcium measurements, only PTH for 15 minutes. I'd like to see more PTH levels and definitely some evidence that blood calcium is affected (unless it isn't).

Response: We appreciate the reviewer's suggestion. We performed new experiments and test PTH levels and calcium levels at different time points according to this suggestion. Our data showed that there was no significant difference in PTH levels between groups at 3 h and 9 h after stimulation. Serum calcium was significantly lower in the ChETA group at 15 and 30 min after blue-light stimulation, and then recovered at 3 h and 9 h post stimulation (Extended Data Fig. 8d).

- On page 19, lines 13- line 21 (Results):

We found that human PTH levels in the serum of nude mice were significantly lower in the ChETA group compared to the control PTG group 5 and 10 min after blue-light stimulation, and had recovered at 15 min (Fig. 7b). There was no significant difference in PTH levels between groups at 3 h and 9 h following stimulation (Extended Data Fig. 8c). Serum calcium decreased significantly in the ChETA group at 15 and 30 min after the blue-light stimulation, and then calcium levels recovered at 3 h and still maintained at 9 h (Extended Data Fig. 8d). Such a reversible inhibition of PTH was successfully reproduced at day 3 of the illumination schedule (Fig. 7b).

16. Please no normalization in Figure 7b.

Response: Thanks for the comments and we fully agree. We have provided the non-normalized data with real numbers and appropriate units. Please refer to the new Figure 7b. (page 43)

17. Figure 7 k and L. Please make clear when blood was drawn with respect to when the light was turned on and PTH levels transiently slightly fall.

Response: Thanks for the comments. We investigated the chronic and long-term effects of optogenetic inhibition of human PTH release on bone. Blue light was used to illuminate the transplanted parathyroid tissue every other day during a period of 28 days. Histological and biochemical analysis were all performed at day 28 after illumination to reflect the long-term effect of the optogenetic regulation.

For the measurement of Rankl and OPG in figure 7k and 7l, we drew the blood and performed the analysis at day 28 after the illumination.

- On page 22, lines 1 - line 10 (Results):

Finally, we determined the level of Rank ligand, which regulates bone resorption and osteoclast activity²¹ in serum, and we found that it was higher in the PTG group than control group, yet lower in the ChETA group compared to the PTG group on the last day of the 28-day illumination protocol (Fig. 7k). The level of the osteoprotegrin

(OPG) was also determined and there was no significant difference between the PTG and control groups, however, OPG levels were higher in the ChETA group than in the control group at day 28 after the illumination (Fig. 7I). Taken together, these data consistently indicate that chronic and rhythmic inhibition of the elevated PTH using optogenetics can enhance bone formation, decrease bone resorption and influence the bone remodeling process.

18. Discussion. The authors state, “The immediate decrease of PKA and cAMP after stimulation indicates that the synthesis of PTH was reversibly inhibited by optogenetic treatment; the light-induced elevation and recovery of the level of PLA2, arachidonic acid, 12-LO and 15-LO all suggest that light stimulation can effectively inhibit the release of PTH.” Actually, I don’t know of any data showing effects of PKA and cAMP on synthesis of PTH. Lowering of PKA activity by lowering cAMP probably decreases secretion of PTH. Calcium, by mechanisms not completely understood, decreases the destruction of PTH mRNA, but exactly how that happens and whether that involves cAMP isn’t settled, I believe (see studies of Many and Silver).

Response: Thanks for the comments. We agree that there is not much evidence supporting the effects of PKA and cAMP on synthesis of PTH, so sorry for this unnecessary confusion. In our study we found that PKA and cAMP levels were significantly lower in the ChETA group compared with both control and eYFP groups immediately after the light stimulation; however, this had recovered 1 h after light stimulation, suggesting that transient lowering of PKA activity and cAMP level by optogenetic stimulation likely decreases secretion of PTH.

The PLA2-arachidonic acid (AA) intracellular signaling pathway may also contribute to regulation of PTH secretion. Our analysis showed that PLA2 and AA levels were significantly higher in ChETA group compared with eYFP group immediately after light stimulation, then had recovered by 1 h after light stimulation. These data suggested that optogenetic regulation of parathyroid cells may inhibit the secretion of human PTH through PLA2-arachidonic acid (AA) signaling pathways.

We agree that we still need to investigate how the delicate intracellular pathways may be involved in optogenetic regulation of PTH. More research is needed to explore the role of crosstalk between calcium and cAMP in synthesis and secretion of PTH.

According to the suggestion, we have rewritten this part of the manuscript to make it clear.

- On page 24, lines 17 - 21 (Discussion):

The immediate decrease of PKA and cAMP after stimulation indicates that the secretion of PTH was reversibly inhibited by optogenetic treatment. The light-induced elevation and recovery of the level of PLA2, arachidonic acid, 12-LO and 15-LO all suggest that light stimulation might effectively inhibit the release of PTH through

PLA2-arachidonic acid (AA) signaling pathways.

19. The authors exaggerate the effect of light in the experiment in Figure 7. In the discussion, they say, “our data demonstrated that, while human parathyroid transplantation decreased bone mineral density of nude mice, the rhythmic inhibition of PTH increased bone mineral density and improved the structure of trabecular bone.” That’s not shown. What the authors show is that bone loss still occurs after light administration, just not as much as in the absence of light. Light makes the bone better than without light, but does not completely reverse the effects of hyperparathyroidism. To make that claim, the authors would need to compare, for example, the control and ChETA bone density measurements. This experiment appears to be very underpowered to test that question.

Response: Thanks for the comments and we agree. Our data supported that optogenetic stimulation improves bone more so than without light, but does not completely reverse the effects of hyperparathyroidism, especially for the cortical bone. Chronic and pulsatile inhibition of human PTH level did ameliorate loss of bone structure in hyperparathyroidism - there was a significant enhancement of trabecular bone formation in the ChETA group compared with the PTG group. HE staining also showed that bone mass was greater and more trabecular bone structures in the ChETA group compared with the PTG group. Taken together, our data supported that the pulsatile inhibition of PTH induced by optogenetic regulation can significantly ameliorate the loss of bone structure in hyperparathyroidism.

According to the suggestion, we have reworded this part for accuracy.

- On page 27, lines 12-16 (Discussion):

In our study, we investigated the chronic and long-term effects of optogenetic inhibition of PTH; our data demonstrated that, while human parathyroid transplantation decreased bone mineral density of nude mice, the optogenetic inhibition of PTH ameliorated the bone structure and arrested the continuous loss of the trabecular bone in hyperparathyroidism.

Response to Reviewer 2 :

Liu and colleagues describe a new approach to regulate PTH secretion from parathyroid cells using an optogenetic approach. Using the light-activated channelrhodopsin variant ChETA, they infect dispersed human primary parathyroid cells, rat glands in vivo, and human parathyroid tissue pieces using lentiviral constructs. They show that membrane depolarization inhibits PTH synthesis and secretion with their system. Using a rat model of secondary hyperparathyroidism, they show a decrease in circulating PTH and calcium. And using transduced human parathyroid pieces, transplanted under the skin of nude mice, they expose the mice with intermittent blue light and show an increase in bone density compared to nude mice carrying pieces of human parathyroid glands without light stimulation.

The authors also engineer a system in vitro and in vivo, which allows light stimulation automatically if culture medium or serum calcium, respectively, increases to a certain concentration.

Response: We thank the reviewer for the summary of our findings. We also very much appreciate the reviewer's comments on the manuscript. Based on the suggestions, we have performed new experiments and added new data in the revised manuscript. We also addressed reviewer's concerns point-by-point in the following text. We hope that this adequately addresses the concerns of the reviewer. The revised manuscript has been improved based on these suggestions, and we are thankful for this.

Comments:

1. Lentivirus is known to have low efficacy in infecting organs in vivo. How do the authors reconcile their findings with this observation?

Response: Thanks for the comments. We agree that many studies have shown that lentivirus has low efficacy in infecting organs *in vivo*. Transfection efficacy depends on the injection position and duration, and our data also showed that we could only transfect a limited proportion of parathyroid chief cells (around 50%, Figure 5d). However, besides transfection rate, specificity of transfection can also determine optogenetic regulatory effects. In a previous study of ours, we demonstrated that we could regulate FGF-2 synthesis and its secretion from astrocytes efficiently with 85.15% specific infection of the cells

(Yang et al. *Nature Communications*, 2014).

In current study, our data shows that the specificity of the viral transfection is around 87% (865 PTH/ChETA double positive cells per 1000 ChETA positive cells), which means that most of the transfected cells were PTH-expressing chief cells and these transfected cells may efficiently inhibit PTH secretion. Our *in vivo* data also confirmed the inhibitory effects of these transfected cells - optical stimulation of the ChETA-transfected parathyroid cells was sufficient to regulate PTH secretion after the light stimulation, PTH levels were significantly lower in the ChETA group compared to the eYFP group 5 and 15 min after the light stimulation, (Figure 5g-h).

Yang F et al., Activated astrocytes enhance the dopaminergic differentiation of stem cells and promote brain repair through bFGF. Nat Commun. 2014 Dec 17;5:5627.

2. Even if a large portion of a parathyroid gland can be infected and their PTH secretion lowered, the remaining uninfected cells will compensate for this. How do the authors explain the *in vivo* effects they observe?

Response: Thanks for the comments. We fully agree that the remaining uninfected cells could compensate for the effects of lowering PTH secretion.

In our experiments, our data showed that serum PTH levels were significantly lower in the ChETA group at 5 and 15 min after the light stimulation, and had recovered at 30 min after stimulation (Fig. 5g). It was interesting for us to observe a prompt decrease and recovery of PTH levels after the *in vivo* light stimulation - the quick PTH response and shorter recovery time compared with *in vitro* stimulation supports the functional compensatory effects of the remaining parathyroid gland that you mention. The capacity of secreting PTH in the remaining uninfected cells might contribute to the observed compensatory effects. We agree that it would be interesting to investigate these effects and underlying mechanism of this compensation in more detail in the future.

According to your suggestion, we have discussed these compensatory effects and underlying mechanism in the discussion part.

- On page 25, lines 20- page 26, line 5 (Discussion):

Compared with the monolayer cell culture and organoid culture of the parathyroid tissue, the release of PTH *in vivo* was influenced not only by the innervation/vascularization of the parathyroid gland, but also by the redundant capacity of the parathyroid gland. We observed a prompt decrease and recovery of PTH levels after illumination, and the quick PTH response and shorter recovery time in comparison with *in vitro* culture, indicates the rapid regulatory function of blood vessels and nerve fibers, and the functional compensatory effects of the uninfected cells in parathyroid gland.

3. Introduction: PTH2 does not have a role in calcium-phosphate and bone metabolism.

Response: Thanks for the comment. We agree that PTH2 does not play a

significant role in calcium-phosphate and bone metabolism. We have deleted the reference according to your suggestion.

- On page 4, lines 3- 5 (Introduction):

The PTH peptide family include the PTH and the PTH-related peptide (PTHrP), both which play important roles in maintaining homeostasis of calcium-phosphate and bone metabolism¹.

4. Low calcium levels do not activate the calcium-sensing receptor.

Response: Thanks for the suggestion. We apologize for the previous inaccuracy. We have rewritten this sentence.

- On page 4, lines 15- 17 (Introduction):

High concentration of calcium ions may activate CaSRs in the parathyroid gland, induce intracellular calcium changes and inhibit the secretion of PTH.

5. How was the rat model of secondary hyperparathyroidism established?

Response: Thanks for the comments. We have added our detailed protocol of establishing rat model of secondary hyperparathyroidism in Methods section.

- On page 66, lines 6- line 17 (Methods):

The rat model of secondary hyperparathyroidism was established as previously described with minor modification³². Rats were divided into two groups based on the diet fed: (1) experimental group on low calcium and high phosphate diet (0.2% Ca, 1.2 % P); (2) control on a normal phosphate and calcium diet (0.6% Ca, 0.6% P). All diets were customized and had the same constituents except for the calcium or phosphate contents: 20% protein and vitamin D 100 IU/100 g (Beijing Keao Xieli Feed Co. Ltd., Beijing, China). All rats were housed at 22-25 °C on a 12-hour light/dark circadian cycle with ad-libitum access to food and water for two weeks. For serum biochemical determinations, blood samples were collected from anesthetized animals (intraperitoneal pentobarbital sodium) by aortic puncture. The immunohistochemistry and HE staining strategies were employed to confirm the parathyroid hyperplasia. Then, virus injection and light regulation were performed to modulate PTH secretion.

6.Fig 3: how was “PKA” measured?

Response: the PKA was tested by ELISA kit (Human Protein Kinase A ELISA Kit, MBS034086, MyBioSource, CA). We measured and compared PKA levels in control, eYFP and ChETA groups. This is now written in the appropriate Method section.

7.Figures 5, 6, and 7: what is “PTH ratio”? Please show the absolute PTH values. Same with calcium values.

Response: Thanks for the comments. We have now provided the non-normalized data with real numbers and appropriate units. Please refer to the new Figure 5, 6 and 7 for the absolute PTH and calcium values.

Response to Reviewer 3

The manuscript ‘An Optogenetic Approach for Regulating Human Parathyroid Hormone Secretion’ describes an optogenetic approach to regulating PTH in models of secondary hyperparathyroidism. This approach bypasses CaSR signaling to activate similar downstream effectors resulting in a reduction in PTH production. While the approach is novel in regards to the application of optogenetics, I have some concerns regarding the methods and controls.

Response: We thank the reviewer for the summary of our findings. We also very much appreciate the reviewer's comments on the manuscript. Based on these comments, we have performed new experiments and added new data in the revised manuscript. We address the reviewer's concerns point-by-point in detail in the following text. We hope that these responses fully address the reviewer's concerns. The revised manuscript is much improved based on the suggestions, and we are grateful for this.

1) The methods surrounding in vivo optical stimulation are lacking. There is no mention of the light source or intensity in in vivo experiments. Blue light has very poor tissue penetrance with the in vivo stimulation its important to show sufficient light is actually reaching the implanted organoid and parathyroid gland. Again, here more experimental details would be helpful in the interpretation of the results.

Response: Thanks for the comments. We fully agree that more experimental details would be helpful in the interpretation of our results. According to your suggestion, we have now provided more experimental details of the optical stimulation protocol in the Methods section.

Blue-light stimulation is widely used for the *ex vivo* and *in vivo* optogenetic experiments. Ye et al. used this method to stimulate mice to control glucagon-like peptide 1 expression to attenuate glycemic excursions in type II diabetic mice (*Science*. 2011 Jun 24;332(6037):1565-8). Recently, Zhang et al. used the same strategy to control insulin secretion (*ACS Synth. Biol.* 2019, 8, 2248-2255) from pancreatic tissue. Optogenetic regulatory effects can be successfully obtained *in vivo* if the light intensity reaches 1 mW/mm² in the

tissue (*European Journal of Neuroscience*, 43 (10), 1298-306) (<http://www.optoneuro.eu/>).

According to your suggestions, we performed new experiments to test the tissue penetrance of our LED device. Our data showed that the penetration rate of the blue light is about 26%, which suggests that the transplanted organoid tissues underneath the skin receive about 9 mW power. Since 1 mW is sufficient to activate the optogene *in vivo*, the device and the strategy we employed are capable of triggering optogenetic effects. For the stimulation of the parathyroid gland *in situ*, we used the optical fiber to deliver light (470 nm, 20 Hz, 5 mW). The light power was 5 mW and was sufficient to activate the expressed optogene in parathyroid cells. PTH quantification after the stimulation also supported that light stimulation effectively inhibited the release of PTH in an established rat model of secondary hyperparathyroidism.

- On page 47, (Table 2):

Table 2

	Pre-skin Light intensity (mW)	Post-skin Light intensity (mW)	Penetrance rate
1	35.75	7.87	22%
2	38.58	12.57	32.60%
3	36.15	6.85	18.90%
4	36.6	10	27.30%
5	36.9	9.15	24.80%
6	34	8.12	23.90%
7	34.82	8.67	24.90%
8	33	8.81	26.70%
9	33.25	9.45	28.40%

Tissue penetrance of the blue light emitting LED device

Light intensity was measured using the photometer above and below the skin, and the light penetrance rate was calculated as the percentage of light intensity

Dheeraj Pelluru, et al. Optogenetic Stimulation of Astrocytes in the Posterior Hypothalamus Increases Sleep at Night in C57BL/6J Mice. *European J Neurosci*, 43 (10), 1298-306 May 2016

2) There are a number of experiments throughout the manuscript that lack important positive controls that would strengthen the author's findings.

- In the first figure, the authors demonstrate that parathyroid cells have an attenuated/ limited response changes in extracellular calcium. Two positive controls would be useful/necessary here.

1) Normal (non-SHPT) parathyroid cells that respond to changes in extracellular calcium demonstrating what a normal response looks like.

Response: Thanks for the suggestions and we fully agree. As suggested, we

have added the positive controls in the respective experiments. We used normal rat parathyroid cells (non-SHPT) as the normal positive control to examine the effects of extracellular calcium challenge on membrane potential, intracellular calcium and PTH secretion level respectively. These new data were summarized in Extended Figure 2 (a-g).

Our results show that an increasing extracellular calcium challenge induced a significant decrease of PTH secretion from normal rat parathyroid cells (Extended Data, Figure 2g). Membrane potential of normal rat parathyroid cells fluctuated during an increase of extracellular Ca^{2+} from 0.5 mM to 2.6 mM; (Extended Data, Figure 2b-c); normal rat parathyroid cells were also sensitive to extracellular Ca^{2+} and showed fluctuations of fluorescence during extracellular Ca^{2+} change (Extended Data, Figure 2e-f). Taken together, these data consistently demonstrate that changing extracellular calcium can induce the normal response in non-SHPT cells.

- On page 8, lines 17-page 9, line 7 (results):

In order to demonstrate the normal response to extracellular calcium, we also isolated and cultured normal rat parathyroid cells (Extended Data Fig. 2a). Patch-clamp recordings showed that the membrane potential of normal rat parathyroid chief cells fluctuated during an increase of extracellular Ca^{2+} from 0.5 mM to 2.6 mM (Extended Data Fig. 2b). Of 10 recorded cells, eight cells were responsive to high extracellular calcium and showed fluctuations of the membrane potential (Extended Data Fig. 2c). Calcium imaging and quantification showed that normal rat parathyroid cells were sensitive to extracellular Ca^{2+} and showed fluctuations of fluorescence during extracellular Ca^{2+} changes (Extended Data Fig. 2e, f). Statistics revealed that 12 of 12 cells were responsive to extracellular Ca^{2+} , showing fluctuations in fluorescence (Extended Data Fig. 2d). Our data also demonstrate that increasing extracellular calcium is capable of lowering PTH levels in normal rat parathyroid cells (Extended Data Fig. 2g).

2) Perfusion of a drug or compound that elicits a ‘normal’ response from the SHPT cells, to demonstrate these cells are healthy and otherwise capable of responding to other types of stimuli

Response: Thanks for the suggestions and we agree. To demonstrate the cells we used are capable of responding to other types of stimuli. We have now tested the response of SHPT cells to the drug Cinacalcet, which is a calcimimetic and indicated for treating hyperparathyroidism. Our data showed that PTH secretion of the SHPT cells decreased significantly 1 hour after cinacalcet treatment compared with the control, and the response of SHPT cells to cinacalcet was also in a dose-dependent manner (Extended Data Fig. 3c). These data suggested that the SHPT cells were still sensitive to the drug and might respond to the other type of stimuli.

- On page 10, lines 21 – page 11, line 2 (results):

Perfusion of cinacalcet, which is indicated for the treatment of hyperparathyroidism, also induced a decrease in PTH level from SHPT cells 1 h following treatment,

evidence that these cells were healthy and responded to the stimuli (Extended Data Fig. 3c).

-Extended Figure 2 - Normal non SHPT cell response

Response: Thanks for the suggestions - we have added the data about normal non SHPT response into the data (Extended Data Fig. 3b).

- On page 11, line 2-line 4 (results):

We also determined the level of PTH secreted from normal rat parathyroid cells after optogenetic stimulation and quantification showed that PTH levels also decreased after the light stimulation (Extended Data Fig. 3b).

-Figure 3 F-K - + control for the Elissa assays - example - F and G Forskolin -> increase cAMP.

Response: Thanks for these suggestions. We have added positive controls as suggested. Forskolin (10uM) was used as the positive control for the quantification of the cAMP and PKA; we found that the levels of PKA and cAMP in parathyroid cells were significantly increased at 5 min, and had recovered at 30 min after Forskolin treatment (Extended Data Fig. 4a-b).

PLA2 activator (5mg/ml) was used as the positive control for the quantification of the PLA2, AA, 12-LO and 15-LO. We found that the levels of A2 (PLA2), arachidonic acid (AA), 12-lipoxygenase (12-LO) and 15-lipoxygenase (15-LO) in parathyroid cells were significantly increased at both 5 min and 30 min after the treatment (Extended Data Fig. 4c-f).

- On page 12, lines 5- 8 (Results):

...we first treated human parathyroid cells with forskolin, which activates the enzyme adenylyl cyclase, and we found that the levels of PKA and cAMP in parathyroid cells were significantly increased at 5 min, and then recovered 30 min after the treatment (Extended Data Fig. 4a, b)

- On page 12, lines 18- page 13, line 1 (Results):

Because the PLA2-arachidonic acid (AA) intracellular signaling pathway may contribute to regulation of PTH secretion¹⁸, we first treated human parathyroid cells with PLA2 activator ,which activated the PLA2-arachidonic acid pathway, and we found that the levels of A2 (PLA2), arachidonic acid (AA), 12-lipoxygenase (12-LO) and 15-lipoxygenase (15-LO) in parathyroid cells were significantly increased at both 5 min and 30 min after the treatment (Extended Data Fig. 4c-f).

3) How does optogenetic stimulation of these cells compare to a normal response to changes in extracellular calcium? Can the optogenetic stimulation be tailored to replicate the response to specific concentrations of extracellular calcium in normal non-SHPT cells?

Response: Thanks for the comments. Normal response to changes in extracellular calcium include decreased PTH secretion, fluctuation of the membrane potential and elevation of intracellular calcium fluorescence (Extended Data Figure 2. b-g). In our study, we found that SHPT cells had

impaired responses to extracellular calcium compared with normal parathyroid cells (Figure 1, d-i). On the other hand, optogenetic treatment of these SHPT cells inhibited the human PTH secretion, depolarized the membrane potential and induced elevation of the intracellular calcium fluorescence (Figure 2 and Figure 3). These data indicate that the effects of optogenetic stimulation of SHPT cells were comparable to the effects of changing extracellular calcium in normal non-SHPT cells.

In our study, we found that optogenetic treatment could also inhibit PTH secretion from normal rat parathyroid cells (Figure 4), suggesting that extracellular calcium and optogenetic stimulation could both induce the decreased PTH secretion from normal parathyroid cells. In order to further compare the inhibitory effects of light stimulation and raising extracellular calcium on the PTH secretion from normal parathyroid cells, we added new experiments to compare the level of PTH after changing extracellular calcium and light stimulation, respectively. Our data show that the increased extracellular Ca^{2+} level significantly inhibited PTH secretion from normal parathyroid cells (Extended Data Fig. 5a). PTH level ($2.63 \pm 0.14 \text{ pg/ml}$) at 2.5 mM extracellular Ca^{2+} was significantly lower than the PTH level ($3.20 \pm 0.11 \text{ pg/ml}$) at 0.5 mM Ca^{2+} . Likewise, 30 min of optical stimulation also significantly inhibited PTH secretion from $3.30 \pm 0.22 \text{ pg/ml}$ to $2.56 \pm 0.14 \text{ pg/ml}$, our data thus suggested a comparable effect between optical stimulation and changing extracellular calcium level.

To further investigate whether optogenetic stimulation could be tailored to replicate these inhibitory effects of extracellular Ca^{2+} , we studied the effects of changing light stimulation time or frequency on inhibition of secreted PTH. We found that increased light stimulation exposure significantly enhanced the inhibitory effects on PTH level. 30 min of 20 Hz blue light stimulation induced a significantly lower PTH level than 15 min of stimulation did, however, there was no significant difference between 30 and 1 h of light stimulation (Extended Data Fig. 5b). As for the light stimulation frequency, PTH level was comparable with light stimulation at 10 Hz, 20 Hz and 30 Hz. Taken together, our data thus suggest that changing stimulation time does affect the inhibition of the PTH level and optical stimulation exposure length can be tailored to mimic the response to specific concentrations of extracellular calcium in normal rat parathyroid cells; however, changing the frequency of light stimulation did not influence the inhibitory effects on PTH (Extended Data Fig. 5c).

- On page 14, lines 20-page 15, line 13 (results):

Since extracellular calcium and optogenetic stimulation both successfully induced decreased PTH levels in normal parathyroid cells, we compared the inhibitory effects of light stimulation and raising extracellular calcium on PTH level. Quantification of the level of PTH secreted from rat parathyroid cells was performed at 0.5 mM, 1.5mM

and 2.5 mM extracellular Ca²⁺ respectively, and we found that the gradually increased extracellular Ca²⁺ level significantly inhibited PTH secretion from parathyroid cells (Extended Data Fig. 5a). The level of PTH at 2.5 mM extracellular Ca²⁺ was significantly lower than the PTH at 0.5 mM and 1.5 mM Ca²⁺. Next, we investigated the effects of light stimulation duration and frequency on inhibition of secreted PTH and found that increased light stimulation duration significantly enhanced the inhibitory effects on PTH levels. Blue light stimulation (30 min at 20 Hz) induced a significantly lower PTH level than 15 min of stimulation did, however there was no significant differences in PTH levels between 30 min and 1 h of light stimulation (Extended Data Fig. 5b). It is interesting to note that PTH levels were comparable between 10 Hz, 20 Hz and 30 Hz of light stimulation, suggesting that changing light frequency may not enhance the inhibitory effects on PTH (Extended Data Fig. 5c).

Minor

1) there are too many abbreviations throughout the paper, which makes it difficult to read. Removing some of the less used abbreviations would improve the readability of the manuscript. Often abbreviations are redefined multiple times.

Response: Thanks for the comments. We have removed some redundant abbreviations, such as conditional medium (CM) and et al.

2) Figure 3a and b doesn't really show much.

Response: Thanks for the comments. In Figure 3a and b, we used a calcium fluorescence assay to reveal light-induced changes of intracellular Ca²⁺ in both ChETA group and eYFP group. In ChETA-expressing cells, blue-light stimulation induced a significant increase in fluorescence signal, which then decreased when the light was turned off. In control eYFP cells, the fluorescence signal remained stable when the blue light was turned on and off. Combined with the quantification of the fluorescence signals (Figure 3c-d), our data demonstrated that blue light stimulation indeed could increase the concentration of the intracellular Ca²⁺.

In order to add to the visibility qualities of these graphs showing ChETA and eYFP groups under light stimulation, we have also uploaded videos of two respective groups of cells under light illumination in supplementary video 1 and 2.

3) Figure 3- How are the cells being optogenetically stimulated along with Fura2. 450-495nm is within the excitation range of Fura2.

Response: Thanks for the comments. Indeed, 450-495nm is within the excitation range of Fura2 and this is why slight elevation of calcium fluorescence was also detected in eYFP-expressing cells when we illuminated these cells with 470 nm blue light (Figure 3d). To measure calcium fluorescence during the optogenetic activation, we referred to Feng Zhang's approach to activating ChR2 with 473 nm and measured Fura2 under 340 nm

simultaneously (*Nature methods*, 2006; *Nature*, 2007). This strategy allowed us to evaluate the changes of calcium fluorescence under optogenetic activation.

Zhang F et al, Channelrhodopsin-2 and optical control of excitable cells. Nature Methods, 2006 Oct;3(10):785-92 (Figure 2)

Zhang F et al., Multimodal fast optical interrogation of neural circuitry. Nature. 2007 Apr 5;446(7136):633-9 (Figure 5c)

4) It is unclear from the methods how Fura2 emissions are being represented. Fura2 is ratio metric and only 365 is being measured?

Response: We used the fluorescence measured at 365 nm to represent the Fura2 emissions. We agree that Fura 2 is ratio metric as previously reported, however, in order to determine the changes of calcium fluorescence in the cells under optogenetic activation, we only measured the fluorescence intensity at 365 nm as described in Zhang's paper.

Zhang F et al., Multimodal fast optical interrogation of neural circuitry. Nature. 2007 Apr 5;446(7136):633-9 (Figure 5c)

5) Figure 3C and D X axis units?

Response: Thanks for the comments. We transferred the images from the 1 Hz acquired video, so the interval of each frame is 0.3 seconds.

6) Figure 4J and K what is S?

Response: "S" means "saturated", indicating the initial phase of the plating parathyroid cells one day before the experiment.

7) Methods lacking on how the rat SHPT model was developed, rat strain, mouse strain and vendor information.

Response: Thanks for the comments. We have added this detailed information into the Methods section.

- On page 62, lines 12- 19 (Methods):

Animals

Adult male C57BL/6 mice (6–8 weeks old), Sprague Dawley rats (8–10 weeks old) and CD-1 nude mice (8–10 weeks old) were used in this study (Guangdong Medical Laboratory Animal Center, Guangzhou, China). Mice and rats were housed at 22–25 °C on a 12–hour light/dark circadian cycle with ad-libitum access to food and water. Animal care and experimental procedures were performed with approval by the Research Committee of the Shenzhen Institutes of Advanced Technology, Chinese Academy of Sciences

- On page 66, lines 6- line 17 (Methods):

The rat model of secondary hyperparathyroidism was established as previously described with minor modification³². Rats were divided into two groups based on the diet fed: (1) experimental group on low calcium and high phosphate diet (0.2% Ca,

1.2 % P); (2) control on a normal phosphate and calcium diet (0.6% Ca, 0.6% P). All diets were customized and had the same constituents except for the calcium or phosphate contents: 20% protein and vitamin D 100 IU/100 g (Beijing Keao Xieli Feed Co. Ltd., Beijing, China). All rats were housed at 22-25 °C on a 12-hour light/dark circadian cycle with ad-libitum access to food and water for two weeks. For serum biochemical determinations, blood samples were collected from anesthetized animals (intraperitoneal pentobarbital sodium) by aortic puncture. The immunohistochemistry and HE staining strategies were employed to confirm the parathyroid hyperplasia. Then, virus injection and light regulation were performed to modulate PTH secretion.

8) How long were cells stored after incubation with fura 2?

Response: After incubation with Fura2, we performed light stimulation procedures or the calcium challenge in less than 3 hours.

9) Methods for calcium imaging - what is the LED device?

Response: We used a home-made light-emitting diode (LED) device for the calcium imaging. The LED device consist of LED units and electric circuit board. The protocol has been described in previous publications (Yang et al., *Nature Communications*. 2014; Tu et al., *Glia*. 2014). During the calcium imaging we fixed the LED device on the microscope (besides the objective lens) and we control the light on/off manually for the optogenetic stimulation.

Yang F et al., Activated astrocytes enhance the dopaminergic differentiation of stem cells and promote brain repair through bFGF. Nat Commun. 2014 Dec 17;5:5627.

Tu et al., Light-controlled astrocytes promote human mesenchymal stem cells toward neuronal differentiation and improve the neurological deficit in stroke rats. *Glia*. 2014 Jan;62(1):106-21.

10) What was the source and stereotype of the AAVs?

Response: We have added these details in our revised Methods section. The AAV is rAAV9-PTH-ChR2-eYFP-WPRE-pA virus (titer: 6.21E12), and the virus were purchased from BrainVTA Co., Ltd., China.

11) Why 20Hz stimulation?

Response: Based on our previous experience, 20 Hz light stimulation can trigger optogenetic manipulation on non-excitabile cells, including astrocytes and glioma cells (Yang et al., *Nature Communications*. 2014; Tu et al, *Glia*. 2014). In the electrophysiological recordings in the current study, we also found that 20 Hz light stimulation could successfully induce ChETA depolarization spike trains in parathyroid cells (Fig. 2f), suggesting that 20 Hz stimulation was sufficient to induce the membrane depolarization of ChETA-expressing parathyroid cells.

Based on your comments, we also investigated the effects of different light stimulation frequencies on inhibiting PTH secreted from normal rat parathyroid cells, and found that PTH levels were comparable over 10 Hz, 20 Hz and 30 Hz frequencies. These data thus suggest that 20 Hz of light stimulation can be used to effectively inhibit the PTH secretion from parathyroid cells.

Yang F et al., Activated astrocytes enhance the dopaminergic differentiation of stem cells and promote brain repair through bFGF. Nat Commun. 2014 Dec 17;5:5627.

Tu et al., Light-controlled astrocytes promote human mesenchymal stem cells toward neuronal differentiation and improve the neurological deficit in stroke rats. Glia. 2014 Jan;62(1):106-21.

12) Calcium Monitoring using electrode and blue light stimulation - Methods need more details on how the system was programmed to work automatically? What hardware was used to monitor calcium levels and deliver light?

Response: Thanks for the suggestions and we fully agree. We have now added more details about calcium monitoring and light stimulation into our revised Methods section. A schematic diagram of calcium monitoring using electrode and blue light stimulation is also summarized in Extended Data Fig. 7. We hope the detailed information and description is helpful in explaining how the system was programmed to work automatically.

- On page 73, lines 20- page 75, line 9 (Methods):

Calcium responding, PTH secretion control set up

A calcium-responding, light-controlling system was set up to detect calcium ion changes and to control the on/off state of blue light (Extended Data Fig. 7). In detail, an electrode was immersed in the dish, which determined the calcium concentration in a real-time manner. At the same time, a home made electronic microchip (and associated software) attached to the electrode compared the calcium concentration with the setting threshold. If the concentration was lower than the threshold, the light above the dish remained off and when the concentration exceeded the threshold, the software then controlled a connected LED device, which turned on to illuminate with blue light (470 nm).

For *in vitro* tests, the program-controlled calcium ion detector was immersed in RPMI-1640 medium in one well of a 6-well plate containing 4 pieces of ChETA/eYFP-expressing PTG organoid. To increase the calcium concentration, 1 ml high concentration calcium chloride (CaCl₂, 100mg/mL, Sigma) solution was added to the well. Once the concentration reached the threshold (50 mg/L), the monitoring program turned the light on. PTH levels in the ChETA and eYFP groups were determined after light stimulation.

For *in vivo* tests, the established system was upgraded by combining with a microdialysis system. The mice were anesthetized by an intraperitoneal (i.p.) injection of pentobarbital (80 mg/kg) before probe implantation surgery and remained anesthetized throughout the experimental period with a rodent anesthesia machine (R520IP, RWD, China). Then, the neck hairs were cut and the blood microdialysis probes (MD-2310, IV-10, BASi, USA) were positioned within the jugular vein. The probe was perfused with 30% ethanol calcium-free physiological saline at a rate of 2.5 μ L/min using a microinjection pump (MD-1000-B, BASi) and micro syringe (MDN-0100, BASi) for 1 h to reach baseline values. Blood dialysates were collected in one well of a 12-well plate, which was filled with 1.5 ml saline and the calcium detector electrode. The calcium ion concentration was read and replaced with fresh saline every 10 min. The resting value of calcium concentration was set as the threshold, and light stimulation was triggered by the detection of the elevated calcium ion. For calcium challenge, CaCl_2 (30 mM, 0.5ml) was delivered by i.p. injection. PTH levels and serum calcium levels were determined after light stimulation in ChETA and eYFP groups.

Reviewers' Comments:

Reviewer #1:

Remarks to the Author:

The authors have addressed the issues I have raised, generally satisfactorily. This is an important paper that shows that optical activation of a mutant rhodopsin can bypass the defects in PTH secretion found in patients with hyperparathyroidism and renal failure to normalize secretion. I have only a few minor comments:

1. I continue to think that the authors are misleading in the way that they say in the Abstract, on line 126, on lines 481-484, and in the Discussion that light-induced suppression of PTH leads to an increase in bone formation. That makes it sound as if the bones of the mice are better than normal. In fact, what the authors show is that their transplantation protocol leads to loss of bone formation and lower trabecular bone mass. This is partially restored with light-induced suppression of PTH secretion. Thus, their therapy prevents bone loss; it doesn't increase bone formation over that in control mice. In fact, the authors provide no statistical tests or comments about the comparison of the treated bones and control bones. So, they should always say that their treatment prevents the loss of bone seen after transplantation. This is not a therapy for osteoporosis! Suppression of PTH has been tried as a therapy for osteoporosis in humans; it doesn't work.
2. Table 1. GFR should have units specified.
3. The paper does not describe the structure of the ChETA opsin or provide a reference that does so. An appropriate reference that describes the mutation in the channel rhodopsin in ChETA would be a good thing to include.
4. Figure 5e calculates the fraction of eGFP+ cells that express PTH. It would be good if they included a calculation of the fraction of PTH-expressing cells that express eGFP.
5. Figure 7j. In the legend and/or the methods, the authors should specify whether the mineral apposition rate was measured in cortical or trabecular bone. Also, they talk about bone formation and probably have the data to allow measure of true bone formation rate. They omitted the measurements of mineralized surface and bone formation rate. Those should be included.

Reviewer #2:

Remarks to the Author:

I appreciate the time and effort that the authors have undertaken to address the critiques that were delivered in the initial review of their manuscript. However, I have a number of concerns with the authors' rebuttal to reviewer comments.

In summary, these are:

1. The authors do not demonstrate a depth of knowledge regarding parathyroid physiology and prior research in the field.
2. The authors do not provide sufficient evidence to support one of the central arguments of the paper, that rhythmic optogenetic suppression of PTH alleviates the impact of SHPT model PTH production on bone *in vivo*.
3. The authors' initial methods of normalization and analysis of data raise high concerns about the integrity and appropriateness of their approach.
4. Inconstancies in the figures are of high concern.
5. The authors do not support their mechanistic argument for PTH secretion suppression with sufficient weight of evidence.
6. The transfection efficiency in parathyroid glands raise serious questions regarding the observed effects *in vivo*.
7. The disregard of the endogenous PTH in the transplanted nude mice is of concern.

More detail on these points below:

1. There are a number of places in the original manuscript where references were either reported incorrectly or were used to support misleading or incorrect statements. The authors have responded to each of the reviewer comments on these errors by revising the text. However, the number of errors made and the lack of rebuttal from the authors on each critique raises the question of whether the authors performed due diligence in their review and understanding of the relevant background in the field.

2. Regarding critiques raised by Reviewer 1 (comments 2 and 19): One of the central discussion points in the manuscript is that rhythmic optogenetic suppression of PTH secretion from secondary hyperparathyroid (SHPT) parathyroid glands can rescue the decrease in bone mass caused by chronic overproduction of PTH in the setting of the disease. This is represented by the *in vivo* experiments in Figure 7.

The authors' claim that rhythmic PTH secretion rescues bone loss in the human parathyroid gland (PTG) transplanted nude mice is not supported by a weight of evidence. Specifically, they provide no point of comparison to show that rhythmic inhibition (rather than nonrhythmic inhibition) protects against bone loss. Of note is the fact that light exposure does not bring the bone parameters to the level of control (no implant) mice, which suggests that removing the human PTGs would have a greater effect on bone (and in fact, this is why primary hyperparathyroidism in humans can be effectively treated with surgical intervention). One would also predict that constant treatment with calcimimetics, which the authors show can effectively suppress PTH production in the human PTGs from SHPT patients (Extended Data Figure 3c), would be able to rescue the observed deleterious effects on bone in the nude mice. Also of concern is a clear discrepancy in the way that the authors characterize rhythmic inhibition in their rebuttal ("small fluctuations during the whole day") and the actual pattern of inhibition used in the experiments, which is 30 minutes of light exposure every other day resulting in a transient (5-10 minutes after light exposure) suppression of the high levels of PTH produced by the human PTGs.

3. I and Reviewer 1 requested non-normalized data for PTH and serum calcium, which the authors kindly provide. However, the non-normalized data raises some strong concerns regarding the approach that the authors chose for initial analysis and presentation of their data:

- The normalization that the authors performed on the data should not have influenced the relative signal intensity for any of the signaling molecules that they tested. Yet, a comparison of the revised (non-normalized) figure and the original shows differences in the relationship between parameters measured in the control and experiment groups. For example, in Figure 3K, in the "normalized" original figure, 15-LO in the ChETA cells is higher at baseline than in the control cells. In the new figure presented in the revised manuscript, 15-LO is lower in the ChETA cells than the control cells at baseline. I do not know of any normalization method that should have this effect on the data. Figure 3k is not the only figure in which this effect can be observed. This is of high concern.

- In addition, from the scatter of data points it is clear that the distribution of the data was also impacted by the original "normalization" process. The calculated p-values for comparison between groups have also changed, which raises suspicions that the authors are not conducting the correct analyses for comparison and representation of their data.

- In Figure 5f, the authors have not changed the labeling on the Y axis. Comparing the figure with the original, the scatter plot appears to show the data points in different positions relative to each other; this cannot be accounted for by the revised scaling. In Figure 5g and 5h, the fact that the original figure has no point scatter at baseline is strange. In the revised figure, there is a clear scatter. The authors must have used the average of the data points at baseline for normalization, but this would eliminate any variability in the baseline data and compromise the statistical comparison of time points after exposure with baseline. Once again, the calculated p-values are different in the revised data compared to the original.

- As above, similar issues arise in Figure 6h, 6i, and 7b. Of particular concern is the way the comparative amount of serum calcium at A30 in eYFP and ChETA conditions in Figure 6i has changed in the revised figure compared to the original. The associated p-value has changed by an order of magnitude for this comparison. This supports the possibility that normalizing each set (eYFP and ChETA) to itself at baseline was an invalid approach for comparing the sets with each other at later time points. It would have exaggerated or minimized differences between the data sets dependent on the differences at baseline – using an average for normalization would further obfuscate some the variation in the data between groups.

4. Extended Fig. 3 shows data from parathyroid cells obtained from patients 1 through 8. It shows the PTH response 1 hour after light stimulation. Fig. 2 shows data from the same experiments with more time points from patients 1-3. However, the data appear to be different. For example, Patient 2 shows an eYFP response at 1 hour of <1.0 PTH/Total Protein (pg/mg) (Fig 2). Yet, in Ext Fig 3 the same patient has an eYFP response of >1.0 PTH/Total Protein (pg/mg) at 1 hour. Scatter plots appear different for the 1 hour PTH levels for several patients. For example, the scatter plot of Patient 3 ChETA appears to be different in Fig. 2 compared to Ext Fig. 3.

5. Regarding the mechanisms of PTH secretion suppression by optogenetic stimulation, the authors

do not provide a satisfactory response to Reviewer 1. Extra ion channels, opened by exposure to blue light, causing depolarization of the membrane and could have a number of effects on cells, including changes in intracellular PKA and cAMP levels. The authors write: "The immediate decrease of PKA and cAMP after stimulation indicates that the secretion of PTH was reversibly inhibited by optogenetic treatment. The light-induced elevation and recovery of the level of PLA2, arachidonic acid, 12-LO and 15-LO all suggest that light stimulation might effectively inhibit the release of PTH through PLA2-arachidonic acid (AA) signaling pathways." They do not provide sufficient weight of evidence for these statements.

6. Regarding my concerns of transfection efficiency, the authors state that transfection efficacy in Figure 5d is around 50%. I would appreciate if the authors provided the means by which they calculated this efficiency and showed the data, as an examination of the figures that they provided does not support a claim of 50% ChETA-transfected cells. The authors say that specificity of transfection (ie, the fact that most transfected cells are PTH-producing chief cells) makes up for this based on their own prior experience with astrocytes and inhibition of FGF-2 synthesis. However, this does not address a known compensatory mechanism within parathyroid glands, in which functional cells can compensate for decreased PTH production from inactive or damaged cells. In fact, the authors argue that this mechanism of increasing PTH production to make up for lost cells/tissue may explain why PTH secretion recovers within 15 minutes of light stimulation as shown in Fig. 5 and in Extended Data Figure 8. It is inconceivable that it takes as long as 15 minutes or more for this compensation to happen.

7. The nude mice transplanted with human parathyroid tissue have their own regulated PTH, which is downregulated in hypercalcemia and upregulated in hypocalcemia. Ext Fig. 8 shows significantly lower serum calcium in mice when the human parathyroid tissue is exposed to blue light with reduction in PTH. This is inconceivable because the endogenous mouse parathyroids would immediately increase the secretion of (mouse) PTH to maintain serum calcium within the normal range at all times. The authors make no attempt to explain that discrepancy or to even measure endogenous PTH as a control.

Additional points:

- Extracellular calcium does not lead to CASR-mediated signals in the human parathyroid cells of secondary hyperparathyroidism (Fig. 1), but the calcimimetic cinacalcet does (extended Fig 3c). Both act on the CASR. How do the authors explain that discrepancy?
- Fig 3a It is not possible to draw conclusions from the data in this panel because it does not contain control cells (e.g., a mix of ChETA-transfected and eYFP transfected cells).
- Fig 4g: it is unclear what the figure is supposed to show
- Fig 4h: it is unclear where the PTH staining is – it looks to the reviewer that all structures in the picture are positive.
- Using normal parathyroid cells, high extracellular calcium (2.6 mM) decreases PTH from 0.8 to 0.65 (Ext Fig 3b). Yet optogenetic inhibition of PTH achieves a much stronger suppression of PTH (Ext Fig 5) despite a transfection efficiency of less than 50%. How do the authors explain that?

Reviewer #3:

Remarks to the Author:

The Authors addressed most of my concerns and the added the necessary details to the methods. A few remaining concerns are listed below

- 1) Figure 1 G the x-axis should be labeled in time (seconds)
- 2) Figure 3 C and D convert and label the x axis in seconds
- 3) What is the PTG group in figure 7? Transplanted PTG with no-transfection? This needs to be more clear in the manuscript. How is this different from control? Line 422 – What is the PTG group? Same question for extended figure 9.
- 4) Line 1332 Seven days after transplantation, light stimulation (470 nm, 20 Hz, 9 mW) was conducted. (how long was the stimulation?)
- 5) Line 420 – "the transplanted tissue every other day during a period of 28 days" How long was the light stimulation each day?
- 6) Line 433 – Please include more details here about the stimulation protocol – 28 days after illumination is too vague. I believe in the methods you mentioned doing it every other day for X? time. This should be included in the main text as well.

7) Line 1331 – implantation of parathyroid organoids – control, efyp and Cheta groups. In these experiments you have the graphs labeled control, PTG and Cheta. This needs to be consistent. Please also describe what 'control' is?

Response to Reviewer #1

The authors have addressed the issues I have raised, generally satisfactorily. This is an important paper that shows that optical activation of a mutant rhodopsin can bypass the defects in PTH secretion found in patients with hyperparathyroidism and renal failure to normalize secretion. I have only a few minor comments:

1. I continue to think that the authors are misleading in the way that they say in the Abstract, on line 126, on lines 481-484, and in the Discussion that light-induced suppression of PTH leads to an increase in bone formation. That makes it sound as if the bones of the mice are better than normal. In fact, what the authors show is that their transplantation protocol leads to loss of bone formation and lower trabecular bone mass. This is partially restored with light-induced suppression of PTH secretion. Thus, their therapy prevents bone loss; it doesn't increase bone formation over that in control mice. In fact, the authors provide no statistical tests or comments about the comparison of the treated bones and control bones. So, they should always say that their treatment prevents the loss of bone seen after transplantation. This is not a therapy for osteoporosis! Suppression of PTH has been tried as a therapy for osteoporosis in humans; it doesn't work.

Response: Thanks for the suggestion. We agree that it would be more accurate to say that our treatment prevents the loss of bone seen after transplantation. As suggested, we have revised the related text in the abstract and discussion part.

- On page 3, lines 65- 67 (Abstract):

In mice, the rhythmic inhibition of PTH induced by optogenetic treatment significantly prevented hyperplastic parathyroid tissue-induced bone loss by influencing the bone remodeling process.

- On page 23, lines 492- 495 (Discussion):

Such an optogenetic approach could be used to regulate the systematic calcium balance and to markedly prevent hyperplastic parathyroid tissue-induced bone loss by influencing the bone remodeling process in mice.

- On page 28, lines 601- 604 (Discussion):

Bone structure parameters, including BV/TV and TbN, together with the trabecular bone morphology, reliably demonstrate that optogenetic inhibition of PTH partially attenuated bone loss induced by PTG transplantation.

- On page 29, lines 622- 625 (Discussion):

The chronic and rhythmic inhibition of PTH induced by optogenetic treatment altered trabecular bone formation and influenced the bone remodeling process, which partially

attenuated bone loss induced by PTG transplantation in mice.

2. Table 1. GFR should have units specified.

Response: Thanks for the suggestion. We added the units for GFR (ml/min) in the Table 1.

- On page 44, lines 890 (Table 1)

3. The paper does not describe the structure of the ChETA opsin or provide a reference that does so. An appropriate reference that describes the mutation in the channel rhodopsin in ChETA would be a good thing to include.

Response: Thanks for the suggestion. As suggested, we have added a reference (Gunaydin LA et al, *Nature Neuroscience*, 2010) to the main text which describes the mutation in the channel rhodopsin in ChETA in detail.

- On page 9, lines 195-198 (Results)

We used ChETA, an engineered opsin gene which has specific advantages over the ChR2 gene in optogenetic control. These include a reduced level of undesirable extra spiking seen in ChR2 and the ability to drive temporally-stationary spiking up to around 200 Hz¹⁸.

18. Gunaydin, L. A. et al. Ultrafast optogenetic control. *Nat Neurosci.* 13, 387-392

4. Figure 5e calculates the fraction of eGFP+ cells that express PTH. It would be good if they included a calculation of the fraction of PTH-expressing cells that express eGFP.

Response: Thanks for this suggestion - we agree. The fraction of eYFP⁺ cells that express PTH show the specificity of virus expression, and the fraction of PTH⁺ cells which express eYFP will show the efficiency of the virus transfection. As suggested, we performed experiments and calculated the fraction of PTH expressing cells which merges to the eYFP positive signals. Our data show that, on average, 46.69% PTH⁺ cells expressed ChETA-eYFP. We added this new data in Figure 5f (page 39).

- On page 16, lines 341-345 (Results)

To evaluate the efficiency of virus transfection, we counted the PTH positive cells expressing green fluorescent protein. On average, 46.69% of PTH+ cells expressed ChETA-eYFP (Fig. 5f). To reveal the transfection specificity, we double stained ChETA-eYFP and PTH and found that ChETA expression (green) was mainly restricted to the parathyroid gland (red) (Fig. 5e).

- On page 39, lines 785-789 (Figure 5f)

f, Quantification of the number of PTH⁺/ChETA⁺ double positive cells per section. A total of 86% (865/1000) of ChETA-positive cells expressed PTH from 20 sections from 3 rats per group. Quantification of the number of ChETA⁺/PTH⁺ double positive cells per section; A total of 46.7% of PTH-positive cells expressed ChETA⁺.

5. Figure 7j. In the legend and/or the methods, the authors should specify whether the mineral apposition rate was measured in cortical or trabecular bone. Also, they talk about bone formation and probably have the data to allow measure of true bone formation rate. They omitted the measurements of mineralized surface and bone formation rate. Those should be included.

Response: Thanks for the suggestion. In this study, we measured mineral apposition rate (MAR) in cortical bone and we have added the details of the MAR measurement in the legend and the methods.

- On page 21, lines 456-458 (Results)

In vivo calcein labeling confirmed that new bone formation and mineral apposition rate (MAR) in cortical bone were both higher in the ChETA group than in the PTG group (Figure 7j).

In order to reflect the actual change of mineral apposition rate in cortical bone, we replaced the MAR ratio with MAR absolute value in Figure 7j.

- On page 42, lines 838 (Figure.7j)

- On page 43, lines 863-865 (Legends- Fig.7)

j, In vivo calcein labeling and calculation of mineral apposition rate (MAR) in cortical bone in the control, PTG and ChETA groups (n=12 sections per group).

- On page 75, lines 1462-1464 (Materials and Methods)

Mineral apposition rate (MAR, $MAR = \text{interlabel width} / \text{labeling period}$) in cortical bone was determined using Zen softwares 2.3 (Carl Zeiss, Germany).

As suggested, we also added the value of the mineralized surface/bone surface (MS/BS), and bone formation rate (BFR) in the Methods, Result section and Figures (Extended Data Fig.9d-e; page 59). Please refer to the figures and related text in the manuscript.

- On page 21, lines 458-462 (Results)

Furthermore, the mineralizing surface versus the bone surface (MS/BS) level in the ChETA group was significantly higher than that of the PTG group, but similar to that of the control group (Extended Data Fig. 9d). Nevertheless, the bone formation rate in cortical bone was higher in the ChETA group than in the PTG group (Extended Data Fig. 9e).

- On page 59, lines 1110-1114 (Extended Data Fig. 9)

d, The mineralizing surface vs. the bone surface in the control, PTG and ChETA groups. values represent mean \pm SEM (n=12 per group). e, The bone formation rate in the control, PTG and ChETA groups. Values represent mean \pm SEM (n=12 per group). All statistical tests in c-e used: * $p < 0.05$; ** $p < 0.01$; *** $p < 0.0001$; one-way analysis of variance (ANOVA) with Tukey's multiple comparisons test.

- On page 75, lines 1464-1466 (Materials and Methods)

The bone formation rate was calculated according to the following formula: $BFR = MAR \times (MS/BS)$, where MS is the mineralizing surface and BS is the bone surface³⁴.

34. Nguyen-Yamamoto, L., Bolivar, Strugnell, S.A & Goltzman, D. Comparison of active vitamin D compounds and a calcimimetic in mineral homeostasis. *J Am Soc Nephrol.* 21, 1713-1723 (2010).

Response to Reviewer #2:

I appreciate the time and effort that the authors have undertaken to address the critiques that were delivered in the initial review of their manuscript. However, I have a number of concerns with the authors' rebuttal to reviewer comments.

Response: We sincerely appreciate that the reviewer put so much effort to help us to improve our manuscript. We will address the reviewer's concerns and explain specific comments in detail.

In summary, these are:

1. The authors do not demonstrate a depth of knowledge regarding parathyroid physiology and prior research in the field.

Response: In order to improve the manuscript, we have done in-depth literature research and paid attention to the latest frontier developments in this field and many relevant citations are included in this manuscript. Incidentally, we recently published a review paper on the parathyroid field on ***J Tissue Eng Regen Med***. This review summarized the latest research relating to parathyroid physiology and tissue engineering.

Reference

Li D, Guo B, Liang Q, Liu Y, Zhang L, Hu N*, Zhang X, **Yang F***, Ruan C*. Tissue-engineered Parathyroid Gland and Its Regulatory Secretion of Parathyroid Hormone, ***J Tissue Eng Regen Med***. 2020 Jun 8. doi: 10.1002/term.3080 (*, Co-corresponding author)

2. The authors do not provide sufficient evidence to support one of the central arguments of the paper, that rhythmic optogenetic suppression of PTH alleviates the impact of SHPT model PTH production on bone in vivo.

Response: Thanks for the constructive suggestion. To address this, we have now added experiments that further explore the nonrhythmic inhibition on PTH secretion in PTG transplanted nude mice. In these experiments, we removed the PTG graft, or performed cinacalcet treatment, to test bone mass and bone structure changes. We will describe these results in detail below in response to point 2.

3. The authors' initial methods of normalization and analysis of data raise high concerns about the integrity and appropriateness of their approach.

Response: Normalization is a widely accepted data processing method, which can directly reflect the rate of change of values among different experimental groups. We think that normalization may change the p value, but will not influence the integrity of our study. Furthermore, we have provided the non-normalized data according to the reviewer's suggestions. Based on the comparison of the normalized and non-normalized data, we demonstrate that

normalization does not affect the accuracy of the final conclusion.

4. Inconstancies in the figures are of high concern.

Response: We apologize that we didn't make it clear in the previous manuscript. The data we have shown in Figure 2 and Extended Figure 3 were collected from different experimental cohorts. We will explain the experimental protocols in detail below in response to point 4.

5. The authors do not support their mechanistic argument for PTH secretion suppression with sufficient weight of evidence.

Response: Thanks for the comment and we agree. We have now rewritten this argument with accuracy and clarity in mind.

6. The transfection efficiency in parathyroid glands raise serious questions regarding the observed effects in vivo.

Response: we have shown in our previous version that the transfection efficiency in parathyroid glands is around 50%. In order to confirm the *in vivo* transfection rate, we performed new experiments including the virus injection and the immunofluorescence, and our new data further confirmed this transfection rate.

In our revised manuscript, we also explain in detail why the compensation of PTH secretion was not as rapid as the reviewer expected, based on our newly obtained experimental data.

7. The disregard of the endogenous PTH in the transplanted nude mice is of concern.

Response: Thanks for the constructive comment. We agree that endogenous PTH in nude mice plays a role in regulating the serum calcium concentration. To address this concern, we conducted new experiments to study the response of endogenous parathyroid gland to the changes of calcium concentration in nude mice. We will describe our new findings in detail below in response to point 7.

More detail on these points below:

1. There are a number of places in the original manuscript where references were either reported incorrectly or were used to support misleading or incorrect statements. The authors have responded to each of the reviewer comments on these errors by revising the text. However, the number of errors made and the lack of rebuttal from the authors on each critique raises the question of whether the authors performed due diligence in their review and understanding of the relevant background in the field.

Response: To improve the manuscript in this regard, we have made great efforts to conduct an in-depth literature review and have studied the related

latest progress in the field of parathyroid biology and physiology. Recently, we also published a review paper on tissue engineering of the parathyroid gland.

Importantly, we highly value each reviewers' comments. Based on these suggestions, we have made careful revision of the original manuscript. We think that the constructive comments would guide us to improve this manuscript. For example, we truly believe the comments about the recommended references from the reviewer 1 resulted in a more refined and precise manuscript. Thus, we accepted most of these suggestions.

Reference

Li D, Guo B, Liang Q, Liu Y, Zhang L, Hu N*, Zhang X, **Yang F***, Ruan C*. Tissue-engineered Parathyroid Gland and Its Regulatory Secretion of Parathyroid Hormone, *J Tissue Eng Regen Med*. 2020 Jun 8. doi: 10.1002/term.3080 (*, Co-corresponding author)

2. Regarding critiques raised by Reviewer 1 (comments 2 and 19): One of the central discussion points in the manuscript is that rhythmic optogenetic suppression of PTH secretion from secondary hyperparathyroid (SHPT) parathyroid glands can rescue the decrease in bone mass caused by chronic overproduction of PTH in the setting of the disease. This is represented by the in vivo experiments in Figure 7.

Response: Thank you for your attention to Reviewers 1's comments and helping us improve the manuscript. Yes, it is one of our claims that rhythmic optogenetic suppression of PTH from SHPT parathyroid glands (PTGs) can prevent the decrease in the bone mass in PTG transplanted nude mice.

The authors' claim that rhythmic PTH secretion rescues bone loss in the human parathyroid gland (PTG) transplanted nude mice is not supported by a weight of evidence. Specifically, they provide no point of comparison to show that rhythmic inhibition (rather than nonrhythmic inhibition) protects against bone loss. Of note is the fact that light exposure does not bring the bone parameters to the level of control (no implant) mice, which suggests that removing the human PTGs would have a greater effect on bone (and in fact, this is why primary hyperparathyroidism in humans can be effectively treated with surgical intervention). One would also predict that constant treatment with calcimimetics, which the authors show can effectively suppress PTH production in the human PTGs from SHPT patients (Extended Data Figure 3c), would be able to rescue the observed deleterious effects on bone in the nude mice.

Response: Thank you for the comments and we agree. It would be more informative to compare the effects of rhythmic PTH inhibition and non-rhythmic PTH inhibition (PTG removal or Cinacalcet treatment) on bone-mass changes in nude mice.

We assigned 21 nude mice to four groups: Blank (no PTG transplantation, n=6), PTG transplant (PTG, n=5), PTG removal (PTG-PTGx, n=5) and Cinacalcet treatment (PTG-Cina, n=5) groups (the latter three groups also had PTG transplants). Two weeks after transplantation, we removed graft and bone samples of mice were collected for analysis two weeks after PTG removal. Two weeks after transplantation, mice in the PTG-Cina group were treated with cinacalcet (30 µg/g daily) for two weeks and bone samples were collected for analysis after treatment.

We found that, compared to the PTG transplanted group, nude mice in the graft removal group (PTG-PTGx) showed significantly higher bone mineral density (BMD) and bone structure improvement. However, the Cinacalcet treated nude mice did not show any improvement in either BMD or bone structure. Previous studies (*Joint Bone Spine*. 2020; *Bone*. 2019) have demonstrated that parathyroidectomy can effectively improve BMD and bone remodeling process in patients with hyperthyroidism, which are consistent with our findings. Our cinacalcet treatment results are also consistent with the other findings that show cinacalcet corrects abnormality of serum calcium and PTH in hyperthyroidism of human or animal models, but has no effect on bone mineral density (*J Clin Endocrinol Metab*. 2009; *J Am Soc Nephrol*. 2010).

Collectively, rhythmic optogenetic inhibition of PTH and parathyroidectomy, but not cinacalcet treatment, can partially reverse bone loss resulting from hyperparathyroidism. All three strategies can correct serum calcium and PTH, but it is not clear why cinacalcet treatment exerts no effects on bone mineral density. The analysis of the mechanism underlying this phenomenon is worthy of future investigation.

Figure 1. BMD and bone structure changes following PTG removal or Cinacalcet

treatment. Compared with control or PTG groups, PTG removal attenuated bone loss, but cinacalcet treatments had no effect on bone

References

1. Guillaume Couture, Michel Laroche. Improvement in bone involvement of secondary hyperparathyroidism post-parathyroidectomy. *Joint Bone Spine*. Available online 26 June 2020. <https://doi.org/10.1016/j.jbspin.2020.06.006>
2. Geovanna O. Pires, Itamar O. Vieira, Fabiana R. Hernandez, Andre L. Teixeira, Ivone B. Oliveira, Wagner V. Dominguez, Luciene M. dos Reis, Fábio M. Montenegro, Rosa M. Moysés, Aluizio B. Carvalho, Vanda Jorgetti. *Bone*. Volume 121, April 2019, Pages 277-283
3. Munro Peacock, Michael A Bolognese, Michael Borofsky, Simona Scumpia, Lulu Ren Sterling, Sunfa Cheng, Dolores Shoback. Cinacalcet treatment of primary hyperparathyroidism: biochemical and bone densitometric outcomes in a five-year study. *J Clin Endocrinol Metab*. 2009 Dec;94(12):4860-7.
4. Loan Nguyen-Yamamoto, Isabel Bolivar, Stephen A Strugnell, David Goltzman. Comparison of active vitamin D compounds and a calcimimetic in mineral homeostasis. *J Am Soc Nephrol*. 2010 Oct;21(10):1713-23.

Also of concern is a clear discrepancy in the way that the authors characterize rhythmic inhibition in their rebuttal (“small fluctuations during the whole day”) and the actual pattern of inhibition used in the experiments, which is 30 minutes of light exposure every other day resulting in a transient (5-10 minutes after light exposure) suppression of the high levels of PTH produced by the human PTGs.

Response: Thanks for the comments. We indeed mentioned that the normal human PTH have rhythms and show small fluctuations during the whole day. In our experiment, we used 30 minutes of light exposure every other day resulting in a transient (5-10 minutes after light exposure) suppression of the high levels of PTH produced by the human PTGs. When we said “**small fluctuations during the whole day**”, what we meant was short time periods of fluctuation, and did not necessarily mean that the magnitude was small, which it was not. So, we think that our optogenetic stimulus protocol does result in a general rhythmic regulation for PTH in every sense of the word rhythm, even although this rhythm is clearly different to that of endogenous physiological regulation of PTH in humans.

3. I and Reviewer 1 requested non-normalized data for PTH and serum calcium, which the authors kindly provide. However, the non-normalized data raises some strong concerns regarding the approach that the authors chose for initial analysis and presentation of their data:

- ***The normalization that the authors performed on the data should not have influenced the relative signal intensity for any of the signaling***

molecules that they tested. Yet, a comparison of the revised (non-normalized) figure and the original shows differences in the relationship between parameters measured in the control and experiment groups. For example, in Figure 3K, in the “normalized” original figure, 15-LO in the ChETA cells is higher at baseline than in the control cells. In the new figure presented in the revised manuscript, 15-LO is lower in the ChETA cells than the control cells at baseline. I do not know of any normalization method that should have this effect on the data. Figure 3k is not the only figure in which this effect can be observed. This is of high concern.

Response: We are grateful to both reviewers for the careful examination of our experimental data, and we would like suggest that there must be some degree of misunderstanding. **In the first round of reviews, reviewer 3 suggested that we add positive controls (Comment 2: -Figure 3 F-K - + control for the Elissa assays - example – F and G Forskolin -> increase cAMP).** In order to keep the data consistent and confirm that our findings can be repeated, we conducted new experiments on all key components of the relevant signaling pathways and replaced all the old data with new data (**We did describe this amendment of positive controls in our previous rebuttal letter**). So, the level of related signaling molecules we presented in the revised manuscript are different from original ones for this reason. That is why the graphs in Fig 3 in the revised and original manuscripts are different, including the inconsistency in 15-LO in original and revised figure.

To be clear, here we list the original non-normalized and revised non-normalized data. We want to emphasize that the absolute values obtained by different experiments are different, but the data trend is relatively stable and does not change out original interpretation. A large number of studies have shown that chief cells from different hyperparathyroidism patients have different PTH secretion capacities. This may explain why the absolute values from different experiments (using different patients) were different. On the other hand, the stable trends we observed demonstrate that optogenetic regulation of the signaling molecules is effective and repeatable.

Figure 2. Original and revised non-normalized signaling pathway data. The absolute values obtained by different experiments are different, but the data trends are relatively stable. We also included the experimental data showing the actual values in the following table for your reference.

Revised non-normalized data

15-LO	Control					ETA					eYFP				
Pre	67.98647	77.29414	96.18827	94.78406	97.63471	65.93162	56.91533	62.23452	55.95877	45.92479	77.96295	114.8486	106.8092	40.76689	106.8092
Post	47.37427	54.87911	45.80818	51.80178	53.29606	335.0667	267.8329	291.8323	227.4335	208.7723	70.09705	55.73982	52.57278	92.5281	100.574
1h-post	113.8212	69.3811	85.3152	147.4629	203.6393	121.3951	99.23571	69.08122	73.98639	72.27572	65.06155	64.39765	50.89492	63.1097	40.19726
12-LO	Control					ETA					eYFP				
Pre	1.891784	2.310679	2.396259	2.231	2.396259	1.75932	2.666186	1.636835	2.119836	1.783586	3.235587	3.821248	4.641333	2.566798	2.920173
Post	2.30936	2.613224	2.498176	2.665839	2.579286	31.56827	24.3806	25.18583	13.80888	19.56258	0.713643	0.97738	1.152808	0.795743	1.167779
1h-post	1.470675	2.066552	2.3972	1.858295	1.664722	0.807317	0.954808	1.28961	1.527692	0.769767	1.864722	1.363046	2.768247	3.678356	1.525682
AA	Control					ETA					eYFP				
Pre	150.4729	138.2416	137.9808	149.8562	160.3724	82.79305	102.1005	94.2753	137.0368	73.59382	104.3565	94.91246	135.9114	120.5497	163.4467
Post	164.1192	157.0349	164.1192	99.51907	174.1584	267.5771	355.7201	250.1798	427.8708	344.8997	93.83765	80.96678	115.9045	81.74906	129.5055
1h-post	202.5827	188.9018	180.4645	164.9143	177.3834	142.1878	0.077188	183.2723	204.3203	146.1771	154.1792	170.1655	155.4775	219.1449	193.3294
cAMP	Control					ETA					eYFP				
Pre	44.474	42.48263	44.474	45.17994	44.474	43.36215	41.42056	43.36215	44.05044	43.36215	42.032	40.032	43.6575	41.032	39.42545
Post	42.3282	42.3282	41.677	42.3282	40.43291	31.53451	31.53451	31.04937	31.53451	30.12252	37.63522	39.80453	38.16404	38.28257	36.15337
1h-post	43.06374	44.62182	44.94742	44.55723	39.48647	38.74926	38.85156	40.03073	41.62176	42.87626	37.22337	39.22374	39.44483	39.76812	36.28667
PKA	Control					ETA					eYFP				
Pre	369.7941	335.6832	333.891	380.3089	399.5872	376.9628	333.3534	343.1505	412.5755	406.7646	334.9065	338.6104	334.3689	382.645	375.624
Post	350.3192	329.4106	302.528	323.4623	326.5429	142.3678	132.7498	138.0068	150.2834	157.1666	328.1561	331.143	326.8418	313.1439	330.5408
1h-post	325.4678	330.008	287.4738	433.6834	444.1757	418.7204	345.4804	356.1139	523.6668	527.5032	360.1762	292.91	288.1907	464.2404	464.2404
PLA2	Control					ETA					eYFP				
Pre	32.45607	32.88895	21.41758	15.02457	16.7524	17.60073	17.15626	15.37077	18.26312	19.63549	15.1222	16.94634	17.48742	14.43011	16.46626
Post	21.35817	17.99733	25.19261	21.19732	23.38359	75.16543	71.8815	74.66012	73.18414	77.12286	16.58555	14.5061	20.86326	17.72356	21.45005
1h-post	16.05883	15.33162	18.20501	28.43843	31.21563	25.33958	23.29292	20.12013	27.03643	27.90857	20.98853	24.72805	14.90611	28.9776	31.37244

Original non-normalized data

15-LO	Control					ETA					eYFP				
Pre	61.793738	64.817497	61.541754	84.82742	47.13339	68.924769	76.761348	64.666306	93.429844	84.537403	72.603679	72.250908	80.213476	97.731182	65.037379
Post	62.398485	66.68215	63.607995	91.054584	79.284599	156.91618	157.54613	123.85641	142.94287	122.18356	77.214912	76.458974	77.567683	101.7756	68.327126
1h-post	63.859971	63.532396	58.517995	90.017831	83.540301	86.991736	84.74911	80.616643	63.585034	84.866363	78.273231	74.644718	66.68215	84.876342	64.142854
12-LO	Control					ETA					eYFP				
Pre	2.831902	3.151622	1.573465	2.606102	2.43052	4.0217095	2.9769244	2.8209699	2.6398138	2.4822654	3.2355874	2.9201728	4.4920146	1.5901112	2.6570034
Post	2.5937428	2.4225048	2.3419397	4.5371865	3.0601518	31.568275	24.380603	25.185828	25.833716	20.216586	2.9283348	3.1469598	5.4763274	4.9421842	4.8197795
1h-post	2.3336531	2.9558952	2.2660736	4.1180234	2.5798515	2.8097299	4.8578537	3.2044766	2.3463855	2.6241783	4.213982	4.5285873	5.2067113	4.5959583	5.6348544
AA	Control					ETA					eYFP				
Pre	115.47968	166.70818	168.48448	155.40561	174.08783	136.79528	127.06115	195.55529	155.75981	155.75891	170.61603	164.43452	215.80511	155.75981	180.80688
Post	184.61327	146.74256	212.11038	169.31913	218.23503	353.14863	283.30453	435.07159	365.0186	330.54539	214.38409	202.02102	158.9635	174.5206	190.0088
1h-post	155.97933	148.87413	212.74987	178.90447	171.40131	205.99994	158.60824	258.29422	208.04944	206.40195	190.72376	152.42673	195.27108	160.7198	187.42365
cAMP	Control					ETA					eYFP				
Pre	127.04348	124.90863	127.49111	121.86955	138.66306	123.91008	121.08657	128.69627	126.01519	116.94063	128.35194	128.4208	112.67384	126.00659	127.09659
Post	133.72348	133.51689	131.5542	139.43839	132.44534	105.21296	101.1843	90.267167	98.534097	100.34833	124.9775	132.18126	126.73358	127.12721	120.05666
1h-post	127.52555	132.13956	133.10369	131.94825	116.93211	117.67696	117.9876	121.56863	126.40038	130.21015	121.7408	128.28307	129.00616	130.06352	118.67728
PKA	Control					ETA					eYFP				
Pre	333.89099	369.79414	331.50573	324.17126	335.68317	319.28623	362.31987	376.9628	343.15054	333.35336	341.42477	334.36891	334.90654	293.74341	338.61037
Post	295.49139	302.52801	329.41057	313.26959	350.31923	175.0346	138.00681	168.54515	132.74978	142.36775	312.61475	326.84179	304.9836	331.143	328.15606
1h-post	271.30492	287.47381	325.27881	325.46778	330.00795	325.28541	340.14128	317.01038	345.48036	324.68223	300.73348	288.19066	360.17617	312.67784	292.91004
PLA2	Control					ETA					eYFP				
Pre	25.916513	21.417583	32.456075	28.184796	32.888948	33.796201	35.712122	42.771552	15.632994	28.160289	16.989615	24.135996	27.901582	15.122198	17.487415
Post	28.961293	25.192612	21.358169	27.513621	17.997329	137.99676	211.63204	121.54268	125.10641	151.31981	22.76476	20.863261	14.506098	26.609007	16.58555
1h-post	14.009406	24.817018	18.769451	15.331623	16.058827	14.12915	23.17592	15.302304	18.226507	15.683035	14.129778	14.906108	17.321027	25.606177	20.988531

- ***In addition, from the scatter of data points it is clear that the distribution of the data was also impacted by the original “normalization” process. The calculated p-values for comparison between groups have also changed, which raises suspicions that the authors are not conducting the correct analyses for comparison and representation of their data.***

Response: Thanks for the comment. As mentioned above, for the signaling pathway molecules in Figure 3, the original normalized data and the revised non-normalized data were collected in different experiments, so the p -values are different.

With this in mind, we do not think that normalization is an incorrect form of analysis, especially when we focus on the fold changes among groups during the experiments (<https://thenode.biologists.com/data-normalization/research/>). In the original version, we intended to evaluate the fold changes of signaling molecules when we performed the optogenetic stimulus. To this end, normalizing with the average of the control group would be a meaningful strategy to make the fold changes more intuitive.

In addition, we also noticed a slight change in the p value between the original normalized data and the original non-normalized data. This was caused by the previous data we used for p analysis is deducted by the mean of control group, and the mean is only saved in two decimal places instead of six decimal places. In the revised manuscript, the p value is generated from the original non-normalized data, which is saved in six decimal places, therefore, the p value changed slightly. For instance, in the PKA data, the $p=0.9996$ between control and ETA group (Pre) in original normalized version, whereas the $p=0.9995$ in original non-normalized data. This is due to the normalization process of rounding all data.

Nevertheless, as the reviewers suggested, we want to reflect the changes of the absolute value, so we accepted the reviewer's suggestion to present non-normalized data in the revised manuscript. Based on the comparison of the normalized and non-normalized data, we demonstrated that normalization does not affect the accuracy of the final conclusion.

- ***In Figure 5f, the authors have not changed the labeling on the Y axis. Comparing the figure with the original, the scatter plot appears to show the data points in different positions relative to each other; this cannot be accounted for by the revised scaling.***

Response: Thanks for this. We changed the labeling in the previous revision on the Y axis in Figure 5f. In the original manuscript, we showed the specificity of the virus transfection using a percentage of PTH⁺/eYFP⁺ (PTH⁺/eYFP⁺, %), whereas in the previous revised version we presented the actual cell numbers

with PTH⁺ and eYFP⁺ labeled (PTH⁺/eYFP⁺ cells). After in-depth consideration, we still think (as we did originally) that percentages better reflect the specificity of the virus transfection, so in the newest revision, we have now reverted back to using percentage, and the graph has the correct Y-axis label in Figure 5f. Furthermore, we also re-explored the virus transfection efficiency and have presented this new data in Figure 5f.

- On page 16, lines 341-348 (Results)

To evaluate the efficiency of virus transfection, we counted the PTH positive cells expressing green fluorescent protein. On average, 46.69% of PTH⁺ cells expressed ChETA-eYFP (Fig. 5f). To reveal the transfection specificity, we double stained ChETA-eYFP and PTH and found that ChETA expression (green) was mainly restricted to the parathyroid gland (red) (Fig. 5e). Quantification revealed that around 86% of the ChETA-positive cells expressed PTH (Fig. 5f), suggesting that we successfully achieved *in situ* expression of the optogene ChETA in the hyperplastic rat parathyroid gland.

- On page 39, lines 785-789 (Figure 5f)

f. Quantification of the number of PTH⁺/ChETA⁺ double positive cells per section. A total of 86% (865/1000) of ChETA-positive cells expressed PTH from 20 sections from 3 rats per group. Quantification of the number of ChETA⁺/PTH⁺ double positive cells per section; A total of 46.7% of PTH-positive cells expressed ChETA⁺.

In Figure 5g and 5h, the fact that the original figure has no point scatter at baseline is strange. In the revised figure, there is a clear scatter. The authors must have used the average of the data points at baseline for normalization, but this would eliminate any variability in the baseline data and compromise the statistical comparison of time points after exposure with baseline. Once again, the calculated p-values are different in the revised data compared to the original.

Response: Thank you for the comment. In the original manuscript, we did not show points scatter at baseline in Figure 5g and 5h because we had normalized the data and defined the baseline as 1. This normalization enabled us to visualize the fold changes after light stimulation more clearly. This approach has been widely used by many researchers in previous studies. For example, to investigate the fold change in microvascular volume under insulin or other treatments, Vincent and colleagues normalized data and defined the baseline as 1 (*Diabetes*, 2004a). In another similar case, to reveal the fold change in cells migration with the mutation, α -actin-R258C in smooth muscle, Zhennan Liu and co-workers also defined the baseline (0 nmol/L) as 1 (*PNAS*, 2017). These studies suggest that the normalization and definition the baseline as 1 is a feasible approach in the analysis of the data.

Yes, in the original manuscript we did normalize the data with the average of the data points at baseline. We agree that this normalization process might

change the p value, however, the normalization may show the data trends in a more intuitive way. As explained above, the change of P value was due to the baseline difference and the rounding during the normalization. Furthermore, we found that the difference of non-normalized baseline values of the ChETA and eYFP groups was not significant. The normalization will not significantly affect the data trend and our conclusion.

Furthermore, according to the reviewer's suggestion, we have provided the non-normalized data in the revised manuscript, and we agree that the new graphs better reflect the original PTH and calcium changes.

References

1. Vincent MA, Clerk LH, Lindner JR, Klibanov AL, Clark MG, Rattigan S, Barrett EJ. Microvascular recruitment is an early insulin effect that regulates skeletal muscle glucose uptake in vivo. *Diabetes*. 2004 Jun;53(6):1418-23.
2. Zhenan Liu, Audrey N Chang, Frederick Grinnell, Kathleen M Trybus, Dianna M Milewicz, James T Stull, Kristine E Kamm. Vascular disease-causing mutation, smooth muscle α -actin R258C, dominantly suppresses functions of α -actin in human patient fibroblasts *PNAS*. 2017 Jul 11;114(28):E5569-E5578.

As above, similar issues arise in Figure 6h, 6i, and 7b. Of particular concern is the way the comparative amount of serum calcium at A30 in eYFP and ChETA conditions in Figure 6i has changed in the revised figure compared to the original. The associated p-value has changed by an order of magnitude for this comparison. This supports the possibility that normalizing each set (eYFP and ChETA) to itself at baseline was an invalid approach for comparing the sets with each other at later time points. It would have exaggerated or minimized differences between the data sets dependent on the differences at baseline – using an average for normalization would further obfuscate some the variation in the data between groups.

Response: Thanks for the comments. Normalization is a widely accepted data processing method in previous studies, which can directly reflect the change rate of different experimental groups. For example, in a recently published paper in *Natural Metabolism* (Chang et al., 2020), the authors normalized cAMP changes induced by calcium and forskoline in parathyroid cells (Fig 4i-k). They also normalized the PTH to basal secretion of PTGs (Figure 6e). In our previous paper published on *Nature Communications* (Yang et al., 2014), we also used a normalization method to reflect the change rate of FGF2 secreted by astrocytes after light stimulation. These studies demonstrate that normalization is a widely accepted method to process data.

Figure 3. Normalization are used to reveal cAMP changes under calcium or other drug treatment (Chang et al., *Nat Metab.* 2020; Fig 4g-k)

Figure 4. Normalization are used to reveal fold change of PTH in PTG (Chang et al., *Nature Metabolism.* 2020; Fig 6)

We agree that normalization may minimize the difference among groups dependent on the baseline difference (eYFP and ChETA groups) and could change p -value. So, we provided the non-normalized data in the previous revised manuscript and re-did the statistical analysis. Our new data analysis shows the changed p value (Figure 6i) compared with the previous normalized data, but this change does not influence the data trend, nor compromise the conclusion of our study.

References

1. Wenhan Chang, Chia-Ling Tu, Frederic G Jean-Alphonse, Amanda Herberger, Zhiqiang Cheng, Jenna Hwong, Hanson Ho, Alfred Li, Dawei Wang, Hongda Liu, Alex D White, Insoo Suh, Wen Shen, Quan-Yang Duh, Elham Khanafshar, Dolores M Shoback, Kunhong Xiao, Jean-Pierre Villardaga. PTH hypersecretion triggered by a GABA_{B1} and Ca²⁺-sensing receptor heterocomplex in hyperparathyroidism. *Nature Metabolism*. 2020 Mar;2(3):243-255.
2. Fan Yang, Yunhui Liu, Jie Tu, Jun Wan, Jie Zhang, Bifeng Wu, Shanping Chen, Jiawei Zhou, Yangling Mu, Liping Wang. Activated astrocytes enhance the dopaminergic differentiation of stem cells and promote brain repair through bFGF. *Nature Communication*. 2014 Dec 17;5:5627.

4. Extended Fig. 3 shows data from parathyroid cells obtained from patients 1 through 8. It shows the PTH response 1 hour after light stimulation. Fig. 2 shows data from the same experiments with more time points from patients 1-3. However, the data appear to be different. For example, Patient 2 shows an eYFP response at 1 hour of <1.0 PTH/Total Protein (pg/mg) (Fig 2). Yet, in Ext Fig 3 the same patient has an eYFP response of >1.0 PTH/Total Protein (pg/mg) at 1 hour. Scatter plots appear different for the 1 hour PTH levels for several patients. For example, the scatter plot of Patient 3 ChETA appears to be different in Fig. 2 compared to Ext Fig. 3.

Response: Thanks for the careful examination. In our study, we first investigated the effect of optogenetic stimulation on PTH secretion of parathyroid cells in 3 patients with SHPT at different time points (Fig 2). We found that PTH secretion in all patients decreased significantly within 1 hour after light stimulation. To verify this finding, we performed further experiments with cells from patients 2-8 to test whether this inhibition of PTH secretion within 1 hour after light exposure can be replicated. Patient 1 had only enough tissue for one test, so whilst Fig 2 and Ext Fig 3 shows the same data for patient 1, whereas patients 2 and 3 we shown in the Ext Fig 3 are from a second test. This is now made clear in the methods section.

- On page 67, lines 1277-1284 (Materials and Methods)

To test the effect of optogenetic stimulation on PTH secretion of parathyroid cells in 1-3 patients with SHPT, PTH concentration was determined at 1 h, 2 h, 4 h, 6 h, 24 h after light stimulation. To verify the inhibitory effects of optogenetic stimulation on PTH secretion at 1 h, PTH determination were conducted with cells from 2-8 patient to test whether this inhibition of PTH secretion within 1 hour after light exposure can be replicated. We did not collect enough parathyroid tissue from patient 1 to perform a new 1 h post light experiment, so we used same data for patient 1 in Extended Fig 3 and Fig 2.

Furthermore, Brown and colleagues found that expression of CaSR in a primary monolayer culture of parathyroid cells gradually decreased over time (*Biochem Biophys Res Commun*, 1995). This may explain why Patient 2 shows lower PTH secretion in Fig 2 than in Extended Fig 3. In the early phase of cell culture, cells may express more CaSR, which respond to extracellular calcium more efficiently to inhibit PTH secretion. However, in the later phase of cell culture, chief cells may express less CaSR, which may weaken the response to extracellular calcium and lead to the increase of PTH secretion. This is why in Ext Fig 3, cells from the same patient have higher concentrations of PTH than those in Fig 2.

Reference

1. Brown AJ, Zhong M, Ritter C, Brown EM, Stetson E. Loss of calcium responsiveness in cultured bovine parathyroid cells is associated with decreased calcium receptor expression. *Biochem Biophys Res Commun*. 1995 Jul 26;212(3):861-7.

5. Regarding the mechanisms of PTH secretion suppression by optogenetic stimulation, the authors do not provide a satisfactory response to Reviewer 1. Extra ion channels, opened by exposure to blue light, causing depolarization of the membrane and could have a number of effects on cells, including changes in intracellular PKA and cAMP levels. The authors write: "The immediate decrease of PKA and cAMP after stimulation indicates that the secretion of PTH was reversibly inhibited by optogenetic treatment. The light-induced elevation and recovery of the level of PLA2, arachidonic acid, 12-LO and 15-LO all suggest that light stimulation might effectively inhibit the release of PTH through PLA2-arachidonic acid (AA) signaling pathways." They do not provide sufficient weight of evidence for these statements.

Response: Thanks for the comment. We agree and have rewritten the sentence to make it clearer and more accurate:

- On page 25, lines 536-540 (Discussion)

The decrease of PKA and cAMP shortly after stimulation implies that the secretion of PTH may be reversibly inhibited by optogenetic treatment. The light-induced elevation and recovery of the levels of PLA2, arachidonic acid, 12-LO and 15-LO suggest that PLA2-arachidonic acid (AA) signaling pathways might be, at least partially, involved in the light stimulation-induced inhibition the release of PTH.

6. Regarding my concerns of transfection efficiency, the authors state that transfection efficacy in Figure 5d is around 50%. I would appreciate if the authors provided the means by which they calculated this efficiency and showed the data, as an examination of the figures that they provided does not support a claim of 50% ChETA-transfected cells.

Response: To support the 50% transfection efficiency claim, we have provided

data to demonstrate this transfection efficiency. Please see the following staining data.

Table: Transfection efficiency

ChETA	eYFP
40.42%	48.678%
47.649%	50.748%
26.222%	88.104%
50%	79.008%
63.322%	31.492%
54.756%	24.471%
28.308%	70.529%
65.964%	31.698%
43.534%	60.656%

Figure 5. The transfection efficiency is about 50%, PTH (red), eYFP (green), DAPI (blue)

In details, we injected lenti-CMV-ChETA-eYFP or lenti-CMV-eYFP virus into the PTG in rats (n=5 per group). Four weeks later, the PTG was removed for immunostaining. The transfection efficiency was calculated by counting eYFP⁺ cells versus PTH⁺ cells. Each picture counted the number of positive cells in two to three visual fields and we used a student's T test (using Prism software) to test the statistical significance of the differences between group means.

The authors say that specificity of transfection (ie, the fact that most transfected cells are PTH-producing chief cells) makes up for this based on their own prior experience with astrocytes and inhibition of FGF-2 synthesis. However, this does not address a known compensatory mechanism within parathyroid glands, in which functional cells can compensate for decreased PTH production from inactive or damaged

cells. In fact, the authors argue that this mechanism of increasing PTH production to make up for lost cells/tissue may explain why PTH secretion recovers within 15 minutes of light stimulation as shown in Fig. 5 and in Extended Data Figure 8. It is inconceivable that it takes as long as 15 minutes or more for this compensation to happen.

Response: Thanks for the comments. First of all, in our previous article published in *Nature Communications* 2014, we demonstrated that optogenetic activation could promote FGF2 synthesis in astrocytes, rather than inhibit synthesis.

Theoretically, we agree that the light stimulation only suppresses PTH secretion in about half of the chief cells, and the rest of the uninfected chief cells would compensate for PTH secretion in a short time period. However, compared with *in vitro* experiments, the secretion and compensation of PTH *in vivo* is very complicated. As we shown in Figure 5g, after the optogenetic inhibition of PTH secretion *in vivo*, the recovery of PTH level is not as rapid as expected. It take 15 minutes or more for the recovery of the PTH level.

To further investigate the compensatory capacity of the parathyroid gland, we did a new experiment where we unilaterally removed parathyroid glands in mice and rat SPHT models and monitored changes in PTH secretion after resection. Our new results showed that serum PTH levels in SHPT mice and rats decreased significantly immediately after parathyroidectomy, and did not return to the original level within 30 minutes after the operation.

In a nude mice experiment, serum PTH decreased significantly within 5 minutes after the removal of the transplanted PTG graft and returned to normal level after 15 minutes. These results consistently indicate that inhibition or partial removal of parathyroid glands can rapidly reduce serum parathyroid levels; however, it will take at least 15 minutes for compensation to be detected.

Figure 6. Unilateral removal PTG in SHPT mice or rat decreased serum PTH and did not recover to pre-operation at least for 30min. It takes 15 minutes for endogenous PTH to compensate for the decrease in PTH caused by graft removal in nude mice

7. The nude mice transplanted with human parathyroid tissue have their own regulated PTH, which is downregulated in hypercalcemia and upregulated in hypocalcemia. Ext Fig. 8 shows significantly lower serum calcium in mice when the human parathyroid tissue is exposed to blue light with reduction in PTH. This is inconceivable because the endogenous mouse parathyroids would immediately increase the secretion of (mouse) PTH to maintain serum calcium within the normal range at all times. The authors make no attempt to explain that discrepancy or to even measure endogenous PTH as a control.

Response: Thanks for the comment. We agree that decreased serum calcium in the nude mice would stimulate endogenous PTH to secrete more PTH. However, the *in vivo* change of serum calcium and PTH is an accumulative process, and it would take time for the endogenous PTH to induce the recovery of the serum calcium level.

To confirm this and further study the response time window of endogenous parathyroid hormone in nude mice to the changes in serum calcium, CaCl_2 (12 mM, 0.1 ml) was injected intravenously in mice, and we found that intravascular injection of CaCl_2 increased serum calcium concentration immediately. However, the decrease of PTH occurred 15 minutes after CaCl_2 injection, rather than immediately. In addition, we also intravenously injected EDTA (12 mM, 0.1

ml), a calcium chelating agent, into mice and continuously monitored changes in serum calcium and PTH concentrations. We found that serum calcium concentration decreased rapidly after injection and returned to the pre-injection level 15 minutes later. Importantly, the increase of PTH secretion was observed at 15 minutes after the EDTA-induced decrease of serum calcium.

Figure 7. CaCl₂ or EDTA injection and the monitoring of serum calcium and PTH response. CaCl₂ and EDTA resulted in rapid change of serum calcium concentration, However, PTH response took longer at around 15 min after injection.

Additionally, we also tested response times of calcium and PTH change after removal of the transplant PTG graft in nude mice. As shown in the following Figure, after parathyroidectomy, the serum PTH level of nude mice decreased significantly within 5 minutes and returned to normal levels 15 minutes after parathyroidectomy, and the serum calcium also returned to normal level at 30 minutes after the partial parathyroidectomy in nude mice.

Figure 8. After parathyroidectomy, the serum PTH level of nude mice decreased significantly within 5 minutes and returned to normal level 15 minutes after parathyroidectomy

Taken together, these *in vivo* data consistently show that there was a time window between the changes in serum calcium and changes in PTH. The recovery of serum calcium was not an instantaneous response.

Additional points:

- **Extracellular calcium does not lead to CASR-mediated signals in the human parathyroid cells of secondary hyperparathyroidism (Fig. 1), but the calcimimetic cinacalcet does (extended Fig 3c). Both act on the CASR. How do the authors explain that discrepancy?**

Response: Extracellular calcium and cinacalcet act on CaSRs by different mechanisms. Extracellular calcium binds to CaSRs directly to regulate PTH secretion, whereas Cinacacet improves the calcium sensitivity of CaSRs on parathyroid cells in patients with hyperparathyroidism to inhibit parathyroid secretion.

Under physiological conditions, Ca^{2+} regulates PTH secretion through interaction with CaSR. However, most secondary hyperparathyroidism patients exhibit elevated PTH despite elevated extracellular Ca^{2+} . These patients show a rightward shift of the PTH- Ca^{2+} curve compared with controls, suggesting a **reduced sensitivity** of parathyroid cells to Ca^{2+} (*Nephrol Dial Transplant*, 2008). Furthermore, Koh and colleagues found that, based on calcium EC_{50} values, the parathyroid cells in primary hyperparathyroidism could be segregated into two distinct categories: one group manifested mean EC_{50} of 2.40 mM, closely aligned to the established normal range. The second group of parathyroid cells were less responsive to calcium stimulus with a mean EC_{50} of 3.61 mM (*J. Cell. Mol. Med.* 2016). This could explain why the 2.6 mM calcium stimulus in our study did not induce a significant decrease of PTH secretion: it could be the case that some parathyroid cells ($\text{EC}_{50}=2.40\text{mM}$) responded to the calcium stimulus by decreasing PTH secretion, whereas another group of cells ($\text{EC}_{50}=3.61\text{mM}$) did not respond, making the total outcome where 2.6 mM calcium did not result in a significant decrease in PTH secretion, which is what we found. A clinical study also showed that, after oral calcium load, ionized calcium increased significantly in hyperparathyroidism patients, whereas PTH remained stable (*EUR J ENDOCRINOL*, 2012). This study offers *in vivo* evidence that elevated calcium in hyperthyroidism patients does not significantly inhibit PTH secretion.

In addition, Cinacalcet, a widely-used drug for hyperparathyroidism patients, is an allosteric modulator of CaSR, which can induce a conformational change that increases the sensitivity of CaSRs to extracellular ionized calcium through enhanced signal transduction (*Nephrol Dial Transplant*, 2008; *PNAS*, 1998). By increasing the sensitivity of parathyroid CaSRs, cinacalcet decreases the set point of the PTH-Calcium curve (*J Am Soc Nephrol*, 2008). Our data showed that cinacalcet treatments (100 nM and 1000 nM) resulted in an approximately 40%-50% reduction in PTH levels in primary culture parathyroid cells from SHPT patients, which is consistent with Kawata's finding that cinacalcet treatments (100 nM and 1000 nM) decreased PTH secretion by 40%-

80% in primary cultured parathyroid cells from SHPT patients (*J Bone Miner Metab*, 2006).

References:

1. Angel L M de Francisco, Maria Izquierdo, John Cunningham, Celestino Piñera, Rosa Palomar, Gema Fernandez Fresnedo, Jose A Amado, Mayte Garcia Unzueta, Manuel Arias. Calcium-mediated parathyroid hormone release changes in patients treated with the calcimimetic agent cinacalcet. *Nephrol Dial Transplant*. 2008 Sep;23(9):2895-901
2. James Koh, Joyce A Hogue, Yuli Wang, Matthew DiSalvo, Nancy L Allbritton, Yuhong Shi, John A Olson Jr, Julie A Sosa. Single-cell functional analysis of parathyroid adenomas reveals distinct classes of calcium sensing behaviour in primary hyperparathyroidism. *J Cell Mol Med*. 2016 Feb;20(2):351-9.
3. Marco Invernizzi, Stefano Carda, Velella Righini, Alessio Baricich, Carlo Cisari, Maurizio Bevilacqua. Different PTH response to oral peptone load and oral calcium load in patients with normocalcemic primary hyperparathyroidism, primary hyperparathyroidism, and healthy subjects. *Eur J Endocrinol*. 2012 Oct;167(4):491-7
4. EDWARD F. NEMETH*, MICHAEL E. STEFFEY, LANCE G. HAMMERLAND, BENJAMIN C. P. HUNG, BRADFORD C. VAN WAGENEN, ERIC G. DELMAR, AND MANUEL F. BALANDRIN. Calcimimetics with potent and selective activity on the parathyroid calcium receptor. *Proc Natl Acad Sci U S A*. 1998 Mar 31;95(7):4040-5
5. Casimiro Valle, Mariano Rodriguez, Rafael Santamaría, Yolanda Almaden, Maria E Rodriguez, Sagrario Cañadillas, Alejandro Martin-Malo, Pedro Aljama. Cinacalcet reduces the set point of the PTH-calcium curve. *J Am Soc Nephrol*. 2008 Dec;19(12):2430-6
6. Takehisa Kawata, Yasuo Imanishi, Keisuke Kobayashi, Naoyoshi Onoda, Senji Okuno, Yoshiaki Takemoto, Takeshi Komo, Hideki Tahara, Michihito Wada, Nobuo Nagano, Eiji Ishimura, Takami Miki, Tetsuro Ishikawa, Masaaki Inaba, Yoshiki Nishizawa. Direct in vitro evidence of the suppressive effect of cinacalcet HCl on parathyroid hormone secretion in human parathyroid cells with pathologically reduced calcium-sensing receptor levels. *J Bone Miner Metab*. 2006;24(4):300-6

• **Fig 3a It is not possible to draw conclusions from the data in this panel because it does not contain control cells (e.g., a mix of ChETA-transfected and eYFP transfected cells).**

Response: In figure 3 (previous version of manuscript), we included both the experimental cells and the control cells. Fig 3a shows two representative ChETA expressing cells (infected with lenti-CMV-ChETA-eYFP virus) which were be activated by blue light. As a control, two representative eYFP expressing cells (infected with lenti-CMV-eYFP virus) were also shown in Fig 3b, which do not respond to the light stimulus. The only difference between these two groups was that the ChETA group contained opsin (expressed ChETA-eYFP), while the eYFP did not (only expressed eYFP). Therefore, the eYFP group was an effective control which can demonstrate that optogenetic

regulation can modulate the activity of parathyroid gland cells. As such, this figure has been retained in the new revised manuscript

- ***Fig 4g: it is unclear what the figure is supposed to show***

Response: Fig 4g shows the expression of fluorescent protein in parathyroid gland tissues cultured *ex vivo* in plate. The images were taken through an inverted microscope. In order to reflect the expression efficiency of eYFP, we also sectioned the parathyroid tissue and used immunofluorescence to display the viral transfection rate in the parathyroid tissue (Figure 4h).

- ***Fig 4h: it is unclear where the PTH staining is – it looks to the reviewer that all structures in the picture are positive.***

Response: Yes, we agree that the image we showed in Fig 4h is not clear enough to distinguish positive signals from background, especially in low magnification images. We have now re-selected and replaced these with new images (Fig 4h) which show the structures and signals more clearly.

- On page 36, lines 740 (Figure.4)
- On page 37, lines 752-754 (Legends- Figure.4)

h, Double staining of PTH and eYFP was performed on sections of the cultured gland and co-localization of ChETA-eYFP and PTH were observed in the parathyroid gland. Scale bar=50 \$\mu\$ m.

- ***Using normal parathyroid cells, high extracellular calcium (2.6 mM) decreases PTH from 0.8 to 0.65 (Ext Fig 3b). Yet optogenetic inhibition of PTH achieves a much stronger suppression of PTH (Ext Fig 5) despite a transfection efficiency of less than 50%. How do the authors explain that?***

Response: In Extended Fig 3b in the previous manuscript, we showed the normal rat parathyroid cell response to optogenetic stimulation, but not the response to the calcium challenge.

We guess the reviewer may be referring to Ext Fig 2g, in which we showed calcium challenge induced responses in normal rat parathyroid cells. As we showed in the Ext Fig 5, the calcium challenge, for example, the 2.5 mM induced decreasing is comparable to 20 Hz blue light induced suppression of PTH. Since these data were collected from primary rat parathyroid cells culture, the transfection efficiency in culture was much higher than 50% (Fig 2b and Ext Fig 2a), which might explain the much stronger effect of PTH suppression.

Response to Reviewer #3:

The Authors addressed most of my concerns and the added the necessary details to the methods. A few remaining concerns are listed below

1) Figure 1 G the x-axis should be labeled in time (seconds)

Response: Thanks for the comment. As suggested, we labeled the Figure 1g with seconds.

- On page 30, lines 645 (Figure.1g)
- On page 31, lines 664-665 (Figure.1g)

g, Quantification of the calcium fluorescence intensity over time in the 4 representative parathyroid chief cells shown in b.

2) Figure 3 C and D convert and label the x axis in seconds

Response: Thanks for the comment. As suggested, we labeled the Figure 1g with Time (seconds).

- On page 34, lines 708 (Figure.3c, 3d)
- On page 34-35, lines 716-720 (Figure.3c, 3d)

c, Quantification of the intensity of calcium fluorescence signals over time in ChETA-expressing cells. Blue light stimulation effectively induced elevation and return to baseline of the calcium fluorescence. d, Quantification of calcium fluorescence signals with time in control eYFP-expressing cells; elevation of signal was not significant during light stimulation.

We also labeled the Extended Figure 2f with seconds.

- On page 48, lines 937 (Extended Figure 2f)
- On page 49, lines 952-953 (Extended Figure 2f)

f, Quantification of the calcium fluorescence intensity over time in 3 representative parathyroid chief cells.

3) What is the PTG group in figure 7? Transplanted PTG with no-transfection? This needs to be more clear in the manuscript. How is this different from control? Line 422 – What is the PTG group? Same question for extended figure 9.

Response: Yes, the PTG group we presented in Figure 7 are the group of nude mice transplanted with no-transfected PTG, this group was used to evaluate changes in serum calcium, PTH and bone metabolism after transplantation. The control group in Figure 7 were normal nude mice without PTG transplantation, which was used to compare the PTH and bone structural difference between PTG transplanted and normal nude mice. The PTG group we described in Line 422 is the nude mice transplanted with no-transfection, the same as extended figure 9. For clarity, we now explain these details in the revised manuscript.

- On page 19, lines 415-418 (Results)

We found that serum levels of human PTH in nude mice were significantly lower in the ChETA group than in the PTG group (transplanted PTG without transfection) 5 and 10 min after blue-light stimulation, and had recovered at 15 min (Fig. 7b).

- On page 20, lines 441-443 (Results)

An obvious lower bone mass in the PTG (transplanted PTG without transfection) group and less trabecular bone structure were revealed by HE staining (Extended Data Fig. 9a).

- On page 42, lines 841-842 (Figure7)

a, Schematic showing transplantation of ChETA-transfected (ChETA group) or non-transfected (PTG group) human parathyroid tissue into the backs of nude mice.

- On page 58, lines 1098-1102 (Extended Figure. 9)

a, HE staining showing trabecular bone structure in the control, PTG (transplanted PTG without transfection) and ChETA (transplanted with ChETA-transfected PTG) groups. Lower bone mass and less trabecular bone structure were observed in PTG group compared with control group; however, in ChETA group, bone mass increased notably and the trabecular bone structure recovered. Scale bar=100 μ m.

4) Line 1332 Seven days after transplantation, light stimulation (470 nm, 20 Hz, 9 mW) was conducted. (how long was the stimulation?)

Response: Thanks. We conducted the light stimulation for 30 minutes per session and we have now added these experimental details in the revised manuscript.

- On page 68, lines 1304-1305 (Materials and Methods)

Seven days after transplantation, light stimulation (470 nm, 20 Hz, 9 mW) was conducted for 30 min per session.

- On page 68, lines 1312-1314 (Materials and Methods)

For long-term stimulation evaluation, light stimulation was conducted three times per week (30 min per session) and bone samples were collected for bone metabolism tests at 28 days after the first stimulation.

5) Line 420 –“ the transplanted tissue every other day during a period of 28 days” How long was the light stimulation each day?

Response: Thanks again. We conducted the light stimulation for 30 minutes per session and we have now added the experiment details in the manuscript.

- On page 19, lines 412-415 (Results)

Human ChETA-transfected parathyroid tissue from SHPT patients was transplanted into the backs of nude mice (Extended Data Fig. 8a, b) and blue light was used to illuminate the transplanted tissue for 30 min every other day during a period of 28 days (Fig. 7a).

6) Line 433 – Please include more details here about the stimulation protocol – 28 days after illumination is too vague. I believe in the methods you mentioned doing it every other day for X? time. This should be included in the main text as well.

Response: Thanks for the suggestion. As suggested, we added the details in main text.

- On page 20, lines 426-429 (Results)

MicroCT scanning was used to evaluate structural changes of both cortical bone and trabecular bone in control (without PTG transplantation), PTG and ChETA groups 28 days after illumination (470 nm, 20 Hz, 9 mW; 30 min per session every other day over a 28-day period) (Fig. 7c).

7) Line 1331 – implantation of parathyroid organoids – control, efyp and Cheta groups. In these experiments you have the graphs labeled control, PTG and Cheta. This needs to be consistent. Please also describe what ‘control’ is?

Response: Thank you very much for picking this up. We have corrected the sentence (line1331) to make the text and graphs consistent. The control group were normal nude mice without PTG transplantation.

- On page 68, lines 1300-1304 (Materials and Methods)

After that, every 8th piece of human parathyroid tissue from different groups were transplanted to nude mice including control (without PTG transplantation), PTG (transplanted with no-transfected PTG) and ChETA (transplanted with ChETA-transfected PTG) groups.

Reviewers' Comments:

Reviewer #1:

Remarks to the Author:

The authors have addressed my concerns satisfactorily. I would like to comment, however, on the authors' response to Reviewer 2 regarding their new cinacalcet experiment. Reviewer 2 comments, "The authors' claim that rhythmic PTH secretion rescues bone loss in the human parathyroid gland (PTG) transplanted nude mice is not supported by a weight of evidence. Specifically, they provide no point of comparison to show that rhythmic inhibition (rather than nonrhythmic inhibition) protects against bone." In response, the authors show that removal of parathyroid transplants but not cinacalcet administration works as well as pulsatile suppression of PTH secretion in preventing bone loss. They use the poor bone response to cinacalcet as evidence that pulsatile PTH suppression is special. But this experiment is flawed. It is well known that, though cinacalcet is good at normalizing blood calcium, it does a terrible job of normalizing PTH levels in patients with hyperparathyroidism. PTH levels generally fall some but do not normalize. This limitation of cinacalcet therapy is probably the reason it does not help bone mass (PTH remains elevated). To claim that rhythmicity of PTH suppression (as opposed simply to constant suppression), the authors would need to show that cinacalcet lowered PTH as much as the optogenetic approach did. They don't measure PTH after cinacalcet, so that no real conclusion can be drawn from their experiment. I don't think that anyone has shown that rhythmicity in PTH secretion is important physiologically and this experiment does not do so either.

Reviewer #3:

Remarks to the Author:

I am satisfied with the responses to my inquires and recommend the manuscript for publication

Reviewer #4:

Remarks to the Author:

NCOMMS-19-1125137B

Liu et al

"An Optogenetic Approach for Regulating Human Parathyroid Hormone Secretion"

This manuscript reports that lentiviral transduction with a construct encoding a recombinant rhodopsin variant (ChETA) can render parathyroid chief cells subject to light-activated plasma membrane depolarization, conferring the capacity for transient, reversible suppression of parathyroid hormone secretion following light pulse stimulation. The authors engineered a calcium probe sensor that triggers light pulses once culture media calcium concentrations exceed a designated threshold and then show that this system can replicate calcium-responsive negative feedback on parathyroid hormone secretion in ChETA-positive cells. Parathyroid hormone secretion by human xenografts of hyperplastic parathyroid tissue transduced with ChETA lentiviruses could be partially suppressed by rhythmic light exposure cycles in vivo. While optogenetic technology has been previously described in other tissue contexts, primarily in modulating neuronal or electrophysiological signaling, the current report is the first to demonstrate utilization of this approach for reconstitution of physiological, ambient ligand sensing and regulatory feedback on hormone secretory activity in an endocrine organ.

As requested by the editorial team, the focus of the comments appended below is on the responsiveness of the authors in addressing issues and concerns raised during the two prior rounds of peer review, with particular attention to the questions highlighted in the thorough and comprehensive critiques articulated by Reviewer 2. In general, the authors have made a good faith effort to address the primary concerns of both reviewers, but a number of critical points remain outstanding.

The first point enumerated by Reviewer 2 was that the authors did not demonstrate sufficient command of parathyroid pathophysiology and in-depth knowledge of prior research in this field.

The authors responded by noting that they have published a recent review article on parathyroid biology in a bioengineering journal; the primary focus of the article is on tissue bioengineering and replacement therapies for mitigating parathyroid diseases. While it is commendable that the senior author's team has published the review, I feel citing that paper rather misses the point, which is that the current manuscript does not provide sufficient pathophysiological understanding and context for evaluating the experimental approach and results. Two examples of this issue are in the selection of SHPT patient tissues for the study and the choice of the rat SHPT model. These two systems are implicitly viewed as essentially equivalent from the standpoint of parathyroid dysfunction, but that is almost certainly not the case. Based on the clinical data in Table 1, all of the patients listed are in acute renal failure with GFRs of less than 10% of normal and profoundly elevated creatinine levels. Parathyroid glands taken from patients with such advanced disease are highly likely to have developed tertiary foci of calcium-non-responsive parathyroid cells and cannot be viewed as simply hyperplastic (this could explain the relative lack of calcium responsiveness in the experiments in figure 1); moreover, PTH levels in these patients are constitutively and dramatically elevated despite chronic hypercalcemia. The limited, relatively low-resolution H/E images in figure 1B are not sufficient to evaluate the histopathological state of these tissues. In contrast, the rat model utilized in figure 5 induces a hyperparathyroid state via dietary calcium deficiency; this causes compensatory gland hyperplasia and appropriately elevated PTH in the context of serum hypocalcemia. In other words, the parathyroid glands in the rat model have undergone hyperplasia as an appropriate calcium sensing response to hypocalcemia - these glands are intrinsically capable of regulating PTH secretion appropriately. There is insufficient methodological detail presented for the figure 5 experiments; how long were the rats kept on the calcium-deficient diet? Gland hyperplasia is reversible upon restoration to calcium-replete diets in many rodent models. Where is the evidence for gland hyperplasia? (the H/E images are not sufficient; no gland weights are given, etc).

A second point raised by both reviewers is that insufficient data were presented to support the assertion that optogenetic suppression of PTH secretion in human SHPT parathyroid tissue xenografts could mitigate the bone mass attrition phenotype of recipient host nude mice. The authors have effectively addressed this criticism by incorporating a new series of nude mouse experiments where bone remodeling effects are evaluated in mice receiving cinacalcet treatment as compared to xenograft removal; these data are intended as reference standards for comparison to the optogenetic response of xenografted mice. As expected, the graft-removal group showed the greatest BMD recovery, while cinacalcet had minimal effect. The magnitude of recovery following graft removal was similar to ChETA-transduced/light-treated grafts for trabecular bone, but graft removal also reversed the cortical bone loss phenotype while ChETA did not. These data (Figure 1 from the response document) should be included in manuscript figure 7. The selective recovery of trabecular but not cortical bone in the ChETA model needs to be discussed, and the absence of any comment on this pattern is similar to the issue in point #1. In humans, PTH injections can reduce cortical mass and increase trabecular mass - generally this is believed to be because trabecular bone has a higher turnover rate, which could also partially explain the experimental result in the xenografted animals. In hyperparathyroidism, though, BMD loss occurs primarily in cortical bone while the trabecular compartment is unaffected. Particularly in patients with SHPT due to renal failure (like the patients in Table 1), elevated PTH is associated with cortical bone loss but trabecular bone is preserved. Basically, then, the murine experiments presented here do show that BMD recovery from the trabecular bone loss induced by the human gland xenografts does occur in the ChETA group, but the recovery does not reproduce the effects of gland removal and do not model the clinical presentation of SHPT.

The third point regarding data normalization in figure 3 and throughout the paper is troubling. While I believe the authors suggestion that the trends and basic findings of the figures are not changed by normalizing, I agree with Reviewer 2 that it is not clear why the relative levels of certain metrics have changed (e.g.: Fig 3k/1 hr post/control > ETA; in the revised non-normalized version of 3k ETA>control). The authors need to provide a clear explanation. The change in the serum calcium values in fig. 6i before and after normalization is not adequately explained and citing the Chang figure is not helpful. The authors need to show the arithmetic for how the normalized values are derived.

The experiments noting changes in PKA, cAMP, PLA2, AA, 12-LO, and 15-LO are of limited

probative value and do not provide evidence of a specific mechanism. The authors have not shown for example, whether the change in PTH release influences PTH production (transcription/translation), vesicular trafficking, or secretion. Mechanistic experiments evaluated the consequences of interfering with the light-associated changes of the second messengers would be required to make any mechanistic inferences.

As Reviewer 2 noted, there is a disconnect between the relatively modest transduction efficiencies of the lentiviral ChETA construct and the magnitude of the reported physiological effects. Particularly in the context of murine or rat hosts where the endogenous PTH/calcium axis remains intact, it is surprising that manipulating ~half of the target tissue could produce the observed, statistically significant changes in serum calcium. Is it possible that ChETA transduction could also be producing non-cell autonomous effects via some unknown paracrine mechanism?

The authors have not satisfactorily addressed Reviewer 2's concerns regarding transduction efficiency and compensatory secretion. It is certainly possible that part of the issue in understanding recovery of PTH levels after the light signal is quenched could potentially be confounded by endogenous host PTH. The authors presumably are measuring only human PTH in the xenograft experiments. In the cited control experiment where PTH levels were evaluated by unilateral, partial parathyroidectomy in the rat SHPT model, the length of time to recovery to pre-resection PTH levels is not simply a function of compensatory secretion. In SHPT, the circulating PTH levels are commonly dependent on gland mass; recovery to pre-resection PTH levels following partial removal of hyperplastic SHPT glands would imply an elevated, fixed setpoint, which is not observed clinically and has no mechanistic basis as far as I know.

Finally, I agree with Reviewer 2's comment that the epifluorescence image quality is poor and is difficult to interpret as presented. In certain cases, double labelling would have been very helpful. For example, in figure 3A, a white arrow indicates a transduced cell where intracellular calcium flux is being measured. It would be helpful to show the GFP channel for the same images so a positively transduced cell can be distinguished from a non-transduced cell; then a comparison flux trace (transduced vs non-transduced) could be shown to demonstrate specificity. Throughout, formal quantitation metrics should be shown to support the assertions of co-localization (figure 6g), successful expression levels (figure 6 b), etc.

Response to Reviewer #1

The authors have addressed my concerns satisfactorily. I would like to comment, however, on the authors' response to Reviewer 2 regarding their new cinacalcet experiment. Reviewer 2 comments, "The authors' claim that rhythmic PTH secretion rescues bone loss in the human parathyroid gland (PTG) transplanted nude mice is not supported by a weight of evidence. Specifically, they provide no point of comparison to show that rhythmic inhibition (rather than nonrhythmic inhibition) protects against bone." In response, the authors show that removal of parathyroid transplants but not cinacalcet administration works as well as pulsatile suppression of PTH secretion in preventing bone loss. They use the poor bone response to cinacalcet as evidence that pulsatile PTH suppression is special. But this experiment is flawed. It is well known that, though cinacalcet is good at normalizing blood calcium, it does a terrible job of normalizing PTH levels in patients with hyperparathyroidism. PTH levels generally fall some but do not normalize. This limitation of cinacalcet therapy is probably the reason it does not help bone mass (PTH remains elevated). To claim that rhythmicity of PTH suppression (as opposed simply to constant suppression), the authors would need to show that cinacalcet lowered PTH as much as the optogenetic approach did. They don't measure PTH after cinacalcet, so that no real conclusion can be drawn from their experiment. I don't think that anyone has shown that rhythmicity in PTH secretion is important physiologically and this experiment does not do so either.

Response: Thanks for the comments. Clinical evidence has indeed demonstrated that cinacalcet intervention did not completely normalize PTH levels in patients with secondary hyperparathyroidism (*Ren Fail.* 2019). However, some animal studies found that cinacalcet treatment (10 mg/kg) normalized PTH and blood ionized calcium to normal control levels in sHPT rat models (*Kidney International.* 2005).

To further investigate the effect of cinacalcet in the transplantation model in our study, we have assessed the serum PTH level of animals without human PTG graft (blank), with human PTG graft (PTG), with human PTG graft and performed graft removal after one week of transplantation (PTG+PTGx), with human PTG graft and treated with cinacalcet after one week of transplantation (PTG+cinacalcet). We found that cinacalcet treatment one week after transplant led to lower [33.5%±15.7%] serum PTH, whereas the removal of PTG transplantation graft group (PTG-PTGx) led to a [39.9%±14.9%] decrease of serum PTH (Extended Data Fig. 11a, b), and light stimulation lowered PTH [39.1%±12.8%] (Figure 7b), which is comparable to the PTG-PTGx group. Our

findings thus demonstrated that PTG-PTGx and light illuminated PTG-ChETA group significantly inhibited PTH secretion and partially reversed bone loss. **Although the cinacalcet group partially inhibited PTH secretion to a level similar to that following optogenetic inhibition, it had no obvious effect on bone loss. Based on these data, we concluded that PTH levels were related to bone mineral density, though the rhythmic inhibition of PTH by optogenetics also played an important role in partially reversing bone loss.**

With regards to the rhythmic secretion of PTH, previous reports have shown that PTH secretion is characterized by an ultradian rhythm with tonic and pulsatile components. In healthy subjects, the majority of PTH is secreted in tonic fashion, whereas approximately 30% is secreted in low-amplitude and high-frequency bursts occurring every 10–20 min. This rhythmic secretion of PTH is disordered in patients with secondary hyperparathyroidism (*Bone research*. 2015). Our findings show that optogenetic stimulation can induce rhythmic inhibition of PTH (Figure 6), and different light-stimulation timings have different inhibitory effects (Extended Data Fig. 6). Together with the increased trabecular bone mass following optogenetic stimulation (Figure 7), our data consistently support that rhythmic inhibition of PTH was involved in affecting the bone remodeling process in SHPT treatment.

Taken together, we agree that low PTH level was important in rescuing bone mass in SHPT. However, our results also demonstrated the effects of optogenetic rhythmic inhibition of PTH on bone remodeling, although the effects of PTH rhythmic inhibition on bone remodeling and the underlying mechanism need further study in the future.

- On page 23, lines 496- 509 (Results):

To further confirm the importance of rhythmic PTH secretion on bone-loss rescue in nude mice transplanted with human parathyroid gland (PTG) , we compared the effects of rhythmic PTH inhibition and non-rhythmic PTH inhibition (PTG removal or continuous cinacalcet treatment) on bone-mass changes in nude mice. First, we found that PTG transplantation significantly elevated serum PTH and calcium levels in nude mice. The removal of the graft or continuous cinacalcet treatment significantly reduced serum PTH and calcium levels; one week of cinacalcet treatment led to a [33.5%±15.7%] decrease in serum PTH, whereas removal of PTG transplantation graft (PTG-PTGx) for 1 week led to a [39.9%±14.9%] decrease of serum PTH (Extended Data Fig. 11a, b). Most importantly, we found that, compared to the PTG transplanted group, nude mice in the graft removal group (PTG-PTGx) had significantly higher bone mineral density (BMD) and bone structure improvement. However, the Cinacalcet treated nude mice did not show any improvement in either BMD or bone structure (Extended Data Fig. 11c-e).

- On page 29, lines 633- 637 (Discussion):

Rhythmic optogenetic inhibition of PTH and parathyroidectomy, but not cinacalcet treatment, can partially reverse bone loss resulting from hyperparathyroidism. All three strategies can correct serum calcium and PTH, but it is not clear why cinacalcet treatment exerts no effects on bone mineral density. The analysis of the mechanism underlying this phenomenon will be an important future investigation.

- On page 72, lines 1380- 1390 (Materials and Methods):

To investigate the importance of rhythmic PTH secretion on the rescue of bone loss in nude mice transplanted with human parathyroid glands (PTG), we compare the effects of rhythmic PTH inhibition and non-rhythmic PTH inhibition (PTG removal or continuous cinacalcet treatment) on bone-mass changes in the nude mice. Briefly, we assigned 21 nude mice to four groups: Blank (no PTG transplantation, n=6), PTG transplant (PTG, n=5), PTG removal (PTG-PTGx, n=5) and Cinacalcet treatment (PTG-Cina, n=5) groups (the latter three groups also had PTG transplants). Two weeks after transplantation, we removed the grafts in mice in the PTG-PTGx group and treated mice in the PTG-Cina group with cinacalcet (30 µg/g daily) for two weeks. Then, bone samples of mice in PTG-PTGx and PTG-Cina group were collected for analysis two weeks after treatments.

Reference

1. Susantitaphong P et al., The effectiveness of cinacalcet: a randomized, open label study in chronic hemodialysis patients with severe secondary hyperparathyroidism. **Ren Fail.** 2019 Nov;41(1):326-333.
2. Chiavistelli S et al., Parathyroid hormone pulsatility: physiological and clinical aspects. **Bone Res.** 2015 Jan 27;3:14049.

Response to Reviewer #3

I am satisfied with the responses to my inquires and recommend the manuscript for publication

Response: We sincerely thank you for your constructive comments in the previous review, which helped us significantly improve our manuscript.

Response to Reviewer #4

This manuscript reports that lentiviral transduction with a construct encoding a recombinant rhodopsin variant (ChETA) can render parathyroid chief cells subject to light-activated plasma membrane depolarization, conferring the capacity for transient, reversible suppression of parathyroid hormone secretion following light pulse stimulation. The authors engineered a calcium probe sensor that triggers light pulses once culture media calcium concentrations exceed a designated threshold and then show that this system can replicate calcium-responsive negative feedback on parathyroid hormone secretion in ChETA-positive cells. Parathyroid hormone secretion by human xenografts of hyperplastic parathyroid tissue transduced with ChETA lentiviruses could be partially suppressed by rhythmic light exposure cycles in vivo. While optogenetic technology has been previously described in other tissue contexts, primarily in modulating neuronal or electrophysiological signaling, the current report is the first to demonstrate utilization of this approach for reconstitution of physiological, ambient ligand sensing and regulatory feedback on hormone secretory activity in an endocrine organ.

As requested by the editorial team, the focus of the comments appended below is on the responsiveness of the authors in addressing issues and concerns raised during the two prior rounds of peer review, with particular attention to the questions highlighted in the thorough and comprehensive critiques articulated by Reviewer 2. In general, the authors have made a good faith effort to address the primary concerns of both reviewers, but a number of critical points remain outstanding.

Response: Thanks for the summary.

The first point enumerated by Reviewer 2 was that the authors did not demonstrate sufficient command of parathyroid pathophysiology and in-depth knowledge of prior research in this field. The authors responded by noting that they have published a recent review article on parathyroid biology in a bioengineering journal; the primary focus of the article is on tissue bioengineering and replacement therapies for mitigating parathyroid diseases. While it is commendable that the senior author's team has published the review, I feel citing that paper rather misses the point, which is that the current manuscript does not provide sufficient pathophysiological understanding and context for evaluating the experimental approach and results. Two examples of this issue are in the selection of SHPT patient tissues for the study and the choice of the rat

SHPT model. These two systems are implicitly viewed as essentially equivalent from the standpoint of parathyroid dysfunction, but that is almost certainly not the case. Based on the clinical data in Table 1, all of the patients listed are in acute renal failure with GFRs of less than 10% of normal and profoundly elevated creatinine levels. Parathyroid glands taken from patients with such advanced disease are highly likely to have developed tertiary foci of calcium-non-responsive parathyroid cells and cannot be viewed as simply hyperplastic (this could explain the relative lack of calcium responsiveness in the experiments in figure 1); moreover, PTH levels in these patients are constitutively and dramatically elevated despite chronic hypercalcemia. The limited, relatively low-resolution H/E images in figure 1B are not sufficient to evaluate the histopathological state of these tissues. In contrast, the rat model utilized in figure 5 induces a hyperparathyroid state via dietary calcium deficiency; this causes compensatory gland hyperplasia and appropriately elevated PTH in the context of serum hypocalcemia. In other words, the parathyroid glands in the rat model have undergone hyperplasia as an appropriate calcium sensing response to hypocalcemia - these glands are intrinsically capable of regulating PTH secretion appropriately. There is insufficient methodological detail presented for the figure 5 experiments; how long were the rats kept on the calcium-deficient diet? Gland hyperplasia is reversible upon restoration to calcium-replete diets in many rodent models. Where is the evidence for gland hyperplasia? (the H/E images are not sufficient; no gland weights are given, etc).

Response: Thanks for the comments. **In this study, we used the rat model mainly to demonstrate the *in vivo* inhibitory effect of optogenetic regulation of hyperplastic parathyroid.** We agree that the rat model we used in this study is different from human sHPT model considering the pathogenesis of hyperparathyroidism. However, in our study we observed similar pathological changes of rat and human gland tissues including histopathological changes (Figure 1' below), weight gain and enlargement of glands (Figure 4' below) and serum PTH increases (Figure 5' below).

1) To better demonstrate the histopathological state of patients from our study, we have replaced the previous figure 1b with the same photos but higher resolution (Figure 2',3' below). More of the HE staining results are also attached below in which it can be seen that both collected patient PTG tissues and rat tissue demonstrate a variety of histological changes including: thickening of connective tissues and packed chief cells with narrow perivascular spaces - both typical histopathological characters of parathyroid hyperplasia (*The American Journal of Pathology*, 1984) (Figure 1' below).

Figure 1'. H&E staining of parathyroid glands of patients and sHPT rats

Figure 2'. High resolution of human parathyroid tissue of Figure 1b

Figure 3'. High resolution of rat parathyroid glands of Figure 5a

2) More methodological details about the rat SHPT model have been added to the methods section.

- On page 68, lines 1298- 1303 (Materials and Methods):

Rats were fed with the high phosphate/low calcium (HPLCa) diet for 7 days before lentivirus injection. This was followed by 15 days during which lentivirus was expressed whilst the HPLCa-diet was not changed. Light stimulation or parathyroid tissue isolation was performed after animals were treated with HPLCa-diet for 21 days. To test whether the diet-induced PTH increase is reversible, we removed the HPLCa-diet and replaced it with standard chow on the 21st day after the HPLCa-diet began.

To demonstrate successful establishment of sHPT in rats, we compared the weight and size of the glands in the HPLCa group and control group. Microscopic surface area in the HPLCa group was significantly larger and parathyroid-gland weight was heavier than in the control group (Figure 4'). H&E staining of the parathyroid glands of HPLCa diet fed rat showed collagen thickening and condensation of chief cells in parathyroid glands, which is one of the histopathological features of hyperparathyroidism (Figure 5a in the manuscript). It could be said that the parathyroid chief cells had undergone pathological changes in the diet-induced sHPT rats.

Figure 4'. Parathyroid hyperplasia was observed following 3 weeks of the HPLCa diet in mice.

a, Schematic diagram of parathyroid hyperplasia in mice; **b**, The surface area of parathyroid glands in mice fed with the HPLCa diet increased significantly; **c,d**, The weight and diameter of parathyroid glands in mice fed with the HPLCa diet were increased

To test whether the HPLCa-diet-induced serum PTH elevation is reversible, we withdrew the HPLCa diet and replaced it with standard chow after 21 days of the HPLCa diet. Our data shows that PTH levels of the HPLCa group decreased in the following week but did not decrease to the previous level (or to the control group level) (Figure 5' below).

Figure 5'. PTH levels can be partially restored after a week of withdrawal of HPLCa diet, but it did not return to the control group level

In summary, we agree that the rat model of hyperparathyroidism induced by diet is not totally the same as sHPT patients with regard to pathogenesis. However, we indeed observed similar histopathological changes in parathyroid glands of the sHPT rat models and human hyperplastic parathyroid tissues. The gland hyperplasia is evident in the rat model of hyperparathyroidism, and the elevated PTH level was not totally reversible upon restoration to normal diet. To further confirm the inhibitory effect of optogenetics on human hyperplastic parathyroid tissue, we also performed human-derived PTG tissues transplantation experiment and these data confirmed the inhibitory effect in sHPT parathyroid graft (Figure 7 in the manuscript).

Reference:

1. Ghandur-Mnaymneh, L. and N. Kimura (1984). "The parathyroid adenoma. A histopathologic definition with a study of 172 cases of primary hyperparathyroidism." *The American journal of pathology* 115(1): 70.

A second point raised by both reviewers is that insufficient data were presented to support the assertion that optogenetic suppression of PTH secretion in human SHPT parathyroid tissue xenografts could mitigate the bone mass attrition phenotype of recipient host nude mice. The authors have effectively addressed this criticism by incorporating a new series of nude mouse experiments where bone remodeling effects are evaluated in mice receiving cinacalcet treatment as compared to xenograft removal; these data are intended as reference standards for comparison to the optogenetic response of xenografted mice. As expected, the graft-removal group showed the greatest BMD recovery,

while cinacalcet had minimal effect. The magnitude of recovery following graft removal was similar to ChETA-transduced/light-treated grafts for trabecular bone, but graft removal also reversed the cortical bone loss phenotype while ChETA did not. These data (Figure 1 from the response document) should be included in manuscript figure 7. The selective recovery of trabecular but not cortical bone in the ChETA model needs to be discussed, and the absence of any comment on this pattern is similar to the issue in point #1. In humans, PTH injections can reduce cortical mass and increase trabecular mass - generally this is believed to be because trabecular bone has a higher turnover rate, which could also partially explain the experimental result in the xenografted animals. In hyperparathyroidism, though, BMD loss occurs primarily in cortical bone while the trabecular compartment is unaffected. Particularly in patients with SHPT due to renal failure (like the patients in Table 1), elevated PTH is associated with cortical bone loss but trabecular bone is preserved. Basically, then, the murine experiments presented here do show that BMD recovery from the trabecular bone loss induced by the human gland xenografts does occur in the ChETA group, but the recovery does not reproduce the effects of gland removal and do not model the clinical presentation of SHPT.

Response: Thanks for the comments. According to these suggestions, we have included the cinacalcet treatment and graft removal result in the Extended Data Fig. 11 in the revised manuscript.

- On page 62, lines 1169- 1183 (Extended Data Fig. 11):

Extended Data Fig. 11 The effect of rhythmic and non-rhythmic inhibition of PTH secretion on bone loss caused by secondary hyperparathyroidism
a,b, Quantification of serum PTH (a), calcium (b) levels in the blank, PTG, PTG+PTGx and PTG+cina groups before and 1 week after treatment; n=5 per group. c, MicroCT scanning was used to evaluate the structural changes of both cortical and trabecular bone in blank, PTG, PTG+PTGx and PTG+cina groups. d, Quantification of bone mineral density (BMD) and bone volume (BV) of cortical bone from blank, PTG, PTG+PTGx and PTG+cina groups; n=5 per group. e, Quantification of bone mineral density (BMD), bone volume/tissue volume (BV/TV), trabecular number (TbN) and trabecular separation (Tb.Sp) of trabecular bone from normal control, PTG, and ChETA groups; n=5 per group. P evaluated from Student's t test. Parathyroid graft removal for PTG+PTGx group, Cinacalcet treatment for PTG+cina group, sham surgery for blank and PTG group.

- On page 23, lines 496- 509 (Results):

To further confirm the importance of rhythmic PTH secretion on bone-loss rescue in nude mice transplanted with human parathyroid gland (PTG) , we compared the effects of rhythmic PTH inhibition and non-rhythmic PTH inhibition (PTG removal or continuous cinacalcet treatment) on bone-mass changes in nude mice. First, we found

that PTG transplantation significantly elevated serum PTH and calcium levels in nude mice. The removal of the graft or continuous cinacalcet treatment significantly reduced serum PTH and calcium levels; one week of cinacalcet treatment led to a [33.5%±15.7%] decrease in serum PTH, whereas removal of PTG transplantation graft (PTG-PTGx) for 1 week led to a [39.9%±14.9%] decrease of serum PTH (Extended Data Fig. 11a, b). Most importantly, we found that, compared to the PTG transplanted group, nude mice in the graft removal group (PTG-PTGx) had significantly higher bone mineral density (BMD) and bone structure improvement. However, the Cinacalcet treated nude mice did not show any improvement in either BMD or bone structure (Extended Data Fig. 11c-e).

- On page 30, lines 633- 637 (Discussion):

Rhythmic optogenetic inhibition of PTH and parathyroidectomy, but not cinacalcet treatment, can partially reverse bone loss resulting from hyperparathyroidism. All three strategies can correct serum calcium and PTH, but it is not clear why cinacalcet treatment exerts no effects on bone mineral density. The analysis of the mechanism underlying this phenomenon will be an important future investigation.

- On page 72, lines 1381- 1391 (Materials and Methods):

To investigate the importance of rhythmic PTH secretion on the rescue of bone loss in nude mice transplanted with human parathyroid glands (PTG), we compare the effects of rhythmic PTH inhibition and non-rhythmic PTH inhibition (PTG removal or continuous cinacalcet treatment) on bone-mass changes in the nude mice. Briefly, we assigned 21 nude mice to four groups: Blank (no PTG transplantation, n=6), PTG transplant (PTG, n=5), PTG removal (PTG-PTGx, n=5) and Cinacalcet treatment (PTG-Cina, n=5) groups (the latter three groups also had PTG transplants). Two weeks after transplantation, we removed the grafts in mice in the PTG-PTGx group and treated mice in the PTG-Cina group with cinacalcet (30 µg/g daily) for two weeks. Then, bone samples of mice in PTG-PTGx and PTG-Cina group were collected for analysis two weeks after treatments.

We agree with the reviewer that PTH injection induces decreased BMD in cortical bone and increase trabecular bone (*Drugs*, 2005); in some hyperparathyroidism patients, clinical findings suggest that BMD loss occurs primarily in cortical bone (*J Bone Miner Res*, 2009).

However, in some animal studies, both trabecular and cortical bone were affected following by PTH injection (*Scientific reports*, 2020). Accordingly, both trabecular and cortical bone are affected during the process of secondary hyperparathyroidism in animal models (*BMC nephrology*, 2018). We noted that the definition of 'trabecular bone' and 'cortical bone' during the BMD assessment is inconsistent in some human and animal studies. In clinical studies, the 'trabecular bone' and 'cortical bone' are not analyzed on the same bone, but that assessment targets mainly comprised of trabecular bone are

regarded as 'trabecular bone' (e.g. spine) and assessment targets mainly comprised of cortical bone is defined as 'cortical bone' (e.g. radius). Though, for animal studies, the trabecular and cortical bone are usually assessed at the same time through drawing the specific ROI. These methodological inconsistencies might explain the disparate findings between human and animal studies.

On the other hand, clinical studies have found that parathyroidectomy treatment induces recovery of BMD in trabecular bone prior to cortical bone (*J Clin Endocr Metab*, 2007), which is consistent with our finding that suppression of PTH in hyperparathyroidism induces increase of BMD primary from trabecular bone. Bone biopsy work has also indicated that mild PHPT induces microstructure change within trabecular bone (*J Clin Endocr Metab*, 1990). Therefore, it is expected that cortical and trabecular bone both respond to changes in PTH, even although there may be latency following PTH change prior to change in BMD. The latency is expected induced by the fact that the large number of bone cells conjugated in trabecular bone makes it easier maintaining the BMD value in this area under the effect of PTH in the early stage. At the same time, the cortical bone serves as a reservoir of minerals (*J Biomed Mater Res A*, 2012) which would be primarily etched by PTH serving the calcium requirements which could be observed easier from BMD assessment. We agree that the long-term effect of optogenetic treatment on the recovery of cortical bone mass would also be possible, and this would require future study.

Reference

1. Rubin MR, & Bilezikian JP.. Parathyroid Hormone as an Anabolic Skeletal Therapy. *Drugs*, 2005. 65(17), 2481–2498.
2. Silverberg SJ et al., Skeletal disease in primary hyperparathyroidism. *J Bone Miner Res*, 2009. 4(3), 283–291.
3. Roberts BC et al., PTH (1–34) treatment and/or mechanical loading have different osteogenic effects on the trabecular and cortical bone in the ovariectomized C57BL/6 mouse. *Scientific reports*. 2020. 10(1): 1-16.
4. Bajwa NM et al., Cortical and trabecular bone are equally affected in rats with renal failure and secondary hyperparathyroidism. *BMC nephrology*, 2018. 19(1): 1-11.
5. Ambrogini E et al., Surgery or surveillance for mild asymptomatic primary hyperparathyroidism: a prospective, randomized clinical trial. *J Clin Endocr Metab*, 2007. 92(8): 3114-3121.
6. Parisien M et al. The histomorphometry of bone in primary hyperparathyroidism: preservation of cancellous bone structure. *J Clin Endocr Metab*. 1990 70(4): 930-938.

7. Nakamura M et al., Bone mineral as an electrical energy reservoir. *J Biomed Mater Res A*. 2012. 100(5): 1368-1374.

The third point regarding data normalization in figure 3 and throughout the paper is troubling. While I believe the authors suggestion that the trends and basic findings of the figures are not changed by normalizing, I agree with Reviewer 2 that it is not clear why the relative levels of certain metrics have changed (e.g.: Fig 3k/1 hr post/control > ETA; in the revised non-normalized version of 3k ETA>control). The authors need to provide a clear explanation. The change in the serum calcium values in fig. 6i before and after normalization is not adequately explained and citing the Chang figure is not helpful. The authors need to show the arithmetic for how the normalized values are derived.

Response: Thanks for your comments. We would like to explain our data normalization process.

For **figure 3k**, in the first round of reviews, reviewer 3 suggested that we add positive controls (**Comment 2: -Figure 3 F-K - + control for the Elissa assays - example – F and G Forskolin -> increase cAMP**). In order to keep the data consistent with the positive controls and confirm that our findings can be repeated, we conducted new experiments on all key components of the relevant signaling pathways and replaced all the old data with new data (We have described this amendment of positive controls in our previous rebuttal letter). So, the level of 15-LO in the previously revised manuscript are different from original ones for this reason.

For **figure 6i**, the calculation process of normalization is as follow:

$$\text{Normalized data} = \frac{\text{value of each eYFP or ChETA group (A0 or A5 or A15 or A30)}}{\text{average value of baseline B (eYFP or ChETA)}}$$

1) For eYFP (labeled light gray in the form below)

Firstly, we calculated average mean of baseline (B) of eYFP: $(1.7371+1.7107+1.6737+1.4568+1.4751+1.4819)/6=1.5892$; Secondly, to visualize the fold changes after light stimulation more clearly, we defined the normalized baseline as 1. Then, we divided all data with the average mean of baseline (1.5892) to obtain normalized data:

$A0^{eYFP}_1=1.8321/1.5892=1.1595$;	$A0^{eYFP}_2=1.6384/1.5892=1.0369$;	$A0^{eYFP}_3=2.5673/1.5892=1.6249$; ...
$A5^{eYFP}_1=1.9359/1.5892=1.2253$;	$A5^{eYFP}_2=2.4613/1.5892=1.5578$;	$A5^{eYFP}_3=2.6203/1.5892=1.6584$; ...
$A15^{eYFP}_1=2.1593/1.5892=1.3667$;	$A15^{eYFP}_2=1.9883/1.5892=1.2584$;	$A15^{eYFP}_3=2.6113/1.5892=1.6528$; ...
$A30^{eYFP}_1=1.6296/1.5892=1.0314$;	$A30^{eYFP}_2=1.9854/1.5892=1.2566$;	$A30^{eYFP}_3=1.8081/1.5892=1.1444$; ...

2) For ETA (labeled light orange in the form below)

First, we calculated the average mean of baseline (B) of ChETA:

$(1.7065+1.7104+1.8684+1.6556+1.9478+1.6021)/6=1.7485$; to visualize the fold changes after light stimulation more clearly, we defined the normalized baseline as 1. Then, we divided all data with the average mean of baseline (1.7485) to obtain normalized data:

$A0^{ChETA_1}=2.6477/1.7485=1.5129$; $A0^{ChETA_2}=2.8428/1.7485=1.6338$; $A0^{ChETA_3}=2.1674/1.7485=1.2456$; ...
 $A5^{ChETA_1}=2.1885/1.7485=1.2506$; $A5^{ChETA_2}=1.7348/1.7485=0.9969$; $A5^{ChETA_3}=2.839/1.7485=1.6316$; ...
 $A15^{ChETA_1}=1.8711/1.7485=1.0692$; $A15^{ChETA_2}=1.5736/1.7485=0.9043$; $A15^{ChETA_3}=1.4869/1.7485=0.8546$; ...
 $A30^{ChETA_1}=1.8088/1.7485=1.0336$; $A30^{ChETA_2}=1.6376/1.7485=0.9411$; $A30^{ChETA_3}=1.4503/1.7485=0.8335$; ...

Figure 6i (raw data)

non-normalized Calcium data (relate to figure 6i)

B	eYFP						ETA					
	1.7371	1.7107	1.6737	1.4568	1.4751	1.4819	1.7065	1.7104	1.8684	1.6556	1.9478	1.6021
A0	1.8321	1.6384	2.5673	2.9009	2.5727	2.7363	2.6477	2.8428	2.1674	3.4043	3.2069	2.1849
A5	1.9359	2.4613	2.6203	2.9181	2.6136	2.6881	2.1885	1.7348	2.839	2.965	2.3127	3.178
A15	2.1593	1.9883	2.6113	2.7331	2.2389	2.0666	1.8711	1.5736	1.4869	1.7133	1.7124	1.5708
A30	1.6296	1.9854	1.8081	1.8279	2.3244	2.246	1.8088	1.6376	1.4503	1.8133	1.7403	1.6969

Normalized Calcium data (relate to figure 6i)

B	eYFP						ETA					
	1	1	1	1	1	1	1	1	1	1	1	1
A0	1.1595	1.0369	1.6249	1.8361	1.6283	1.7318	1.5129	1.6338	1.2456	1.9565	1.8431	1.2557
A5	1.2253	1.5578	1.6584	1.8469	1.6542	1.7013	1.2506	0.9969	1.6316	1.704	1.3291	1.8265
A15	1.3667	1.2584	1.6528	1.7298	1.4171	1.3079	1.0692	0.9043	0.8546	0.9846	0.9841	0.9028
A30	1.0314	1.2566	1.1444	1.1569	1.4712	1.4215	1.0336	0.9411	0.8335	1.0422	1.0002	0.9753

All the source data and the normalization process can be found in attached letter.

The experiments noting changes in PKA, cAMP, PLA2, AA, 12-LO, and 15-LO are of limited probative value and do not provide evidence of a specific mechanism. The authors have not shown for example, whether the change in PTH release influences PTH production (transcription/translation), vesicular trafficking, or secretion. Mechanistic experiments evaluated the consequences of interfering with the light-associated changes of the second messengers would be required to make any mechanistic inferences.

Response: Thanks for the comment. As suggested, we added new experiments and have now assessed *MafB*, *Gcm2* and *PTH* for PTH production (*J Endocr Soc.* 2020), and *Rab8a* and *Rab11a* for vesicular trafficking (*Cell Death Dis.* 2016) in the control, ChETA and eYFP groups.

We found that light stimulation reversibly suppressed *MafB*, *Gcm2*, *PTH*. *Rab8a* and *Rab11a* were significantly lower in the ChETA group compared with eYFP group immediately after light stimulation, then had recovered by 1 h after light stimulation (Extended Data Fig. 5a-e in the manuscript). These results demonstrate that light stimulation can reversibly inhibit PTH production and vesicle trafficking in sHPT-derived parathyroid chief cells. Together with the data we presented in figure 3 in the manuscript, our data suggests that light stimulation can inhibit PTH secretion in sHPT-derived parathyroid chief cells.

We added primer information and process of these new experiments in the Materials and Methods section.

- On page 76-77, lines 1482- 1492 (Materials and Methods):

For PTH production and vesicle trafficking analysis, primers used for GCM2 were: 5'CGCTTCTTCTAGCTTCTGTCTC3', 5'TGATGGCAACGCGATCTT3' (Final product 89 bp); MAFB: 5' GGGTTCATCTGCTGGTAGTT3', 5'GCTCAGCACTCCGTGTAG3' (Final product 114 bp); PTH: 5'CGTAAGAAGCTGCAG3', 5'CACATCAGCTTTGTC3' (Final product 153 bp); Rab8a: 5'TTGGATTCCGGAACATTGAGG3', 5'GCCTCTTGTCGTTACATCAC3' (Final product 86 bp); Rab11a: 5'GCAACAAGAAGCATCCAGG3', 5'GCACCTACTGCTCCACGAT3' (Final product 116 bp); GAPDH was used as internal control to normalize RNA content: 5'GCATGGCCTTCCGTGTTC3', 5'CCTGCTTACCACCTTCTTGAT' (Final product 105 bp).

- On page 13-14, lines 282- 293 (Results):

To further study whether light stimulation can regulate PTH production, we evaluated changes in *GCM2*, *MAFB*, and *PTH*, genes closely related to PTH production, before and after the light illumination. We found that light stimulation effectively inhibited expression levels of *GCM2*, *MAFB*, and *PTH*, and this inhibition returns to normal levels within 1 hour after cessation of light stimulation (Extended Data Fig. 5a-c). In addition, we also tested the effect of light stimulation on expression levels of genes related to intracellular membrane vesicle transport in PTG cells, such as *Rab8a* and *Rab11a*. The results show that light stimulation can effectively inhibit expression levels of *Rab8a* and *Rab11a*, and that this inhibition returns to normal levels within one hour after the light stimulation (Extended Data Fig. 5d-e). These results indicate that optogenetic stimulation can inhibit PTH synthesis, vesicle transport and secretion processes in PTG cells..

- On page 54, lines 1041- 1057 (Extended Data Fig. 5):

Extended Data Fig. 5 RT-PCR analysis key elements of PTH production and vesicle trafficking in PTG cells before and after light stimulation.

a, *GCM2* levels in the control, eYFP and ChETA groups before (Pre), immediate after (Post) and 1 h after light stimulation (1h-post) values represent mean \pm SEM (n=5 per group); b, *MAFB* in the control, eYFP and ChETA groups before (Pre), immediate after

(Post) and 1 h after light stimulation (1h-post); values represent mean \pm SEM (n=5 per group); c, PTH in the control, eYFP and ChETA groups before (Pre), immediate after (Post) and 1 h after the light stimulation (1h-post); values represent mean \pm SEM (n=5 per group); d, Rab8a in the control, eYFP and ChETA groups before (Pre), immediate after (Post) and 1 h after the light stimulation (1h-post). Values represent mean \pm SEM (n=5 per group); e, Rab11a in the control, eYFP and ChETA groups before (Pre), immediate after (Post) and 1 h after the light stimulation (1h-post); values represent mean \pm SEM (n=5 per group). All statistical tests in a-e used: *p<0.05; **p<0.01; ***p<0.0001; one-way analysis of variance (ANOVA) with Tukey's multiple comparisons test.

Reference

- 1 . Fabbri S, Zonefrati R, Galli G, Gronchi G, Perigli G, Borrelli A, Brandi M. In Vitro Control of Genes Critical for Parathyroid Embryogenesis by Extracellular Calcium. *J Endocr Soc*. 2020 May 25;4(7):bvaa058..
- 2 . Zhu H, Xue CB, Xu X, Guo YB, Li XH, Lu JJ, Ju SQ, Wang YJ, Cao Z, Gu XS. Rab8a/Rab11a regulate intercellular communications between neural cells via tunneling nanotubes. *Cell Death Dis*. 2016 Dec 22;7(12):e2523.

As Reviewer 2 noted, there is a disconnect between the relatively modest transduction efficiencies of the lentiviral ChETA construct and the magnitude of the reported physiological effects. Particularly in the context of murine or rat hosts where the endogenous PTH/calcium axis remains intact, it is surprising that manipulating ~half of the target tissue could produce the observed, statistically significant changes in serum calcium. Is it possible that ChETA transduction could also be producing non-cell autonomous effects via some unknown paracrine mechanism?

Response: Thanks for the constructive suggestion. We agree that there may be non-cell autonomous effects between parathyroid chief cells. Actually, previous work has indicated paracrine mechanisms in bovine parathyroid chief cells: chief cells of close proximity are stimulated to secrete more PTH than those at lesser densities; parathyroid cells are recruited to secrete PTH when plated at high density (*J Bone Miner Res*, 1994). It may be that there is a positive feed-back regulation loop between the parathyroid chief cells, which might be able to explain our data that even although only half of the parathyroid cells are transfected and suppressed by optogenetic regulation, there is a significant decrease of serum PTH in animals.

To confirm the paracrine effects, we compared the expression of PTH and the ChETA in situ on the rat parathyroid gland cells using immunofluorescence staining (Figure 6' below). We found that the cells with high expression of ChETA (**green arrow**) had low expression of PTH after light stimulation. Chief

cells with high PTH expression (**red arrow**) were typically not transfected with ChETA. However, there were also cells that expressed low PTH without expression of ChETA (**white arrow**). These cells might have been regulated by non-cellular autonomous effects such as paracrine signaling from the ChETA-affected cells, but the mechanism of this phenomenon requires further study.

Figure 6'. non-cell autonomous effects were observed in parathyroid cells

Reference

1. Sun F et al., Paracrine interactions among parathyroid cells: effect of cell density on cell secretion. *J Bone Miner Res*, 1994. 9(7): p. 971-976.

The authors have not satisfactorily addressed Reviewer 2's concerns regarding transduction efficiency and compensatory secretion. It is certainly possible that part of the issue in understanding recovery of PTH levels after the light signal is quenched could potentially be confounded by endogenous host PTH. The authors presumably are measuring only human PTH in the xenograft experiments. In the cited control experiment where PTH levels were evaluated by unilateral, partial parathyroidectomy in the rat SHPT model, the length of time to recovery to pre-resection PTH levels is not simply a function of compensatory secretion. In SHPT, the circulating PTH levels are commonly dependent on gland mass; recovery

to pre-resection PTH levels following partial removal of hyperplastic SHPT glands would imply an elevated, fixed setpoint, which is not observed clinically and has no mechanistic basis as far as I know.

Response: Thanks for the careful examination. We would like to explain this more clearly. There might be a misunderstanding here regarding interpretation of the PTGx results that we demonstrated in our previous reply letter. In the previous revision, we demonstrated that we performed one-side parathyroidectomy surgery for both SD rat and C57 mice, and found that the serum PTH level of the animals decrease and did not recover to normal within 30 mins. These results indicate that the intact parathyroid gland on the other side could not completely compensate the serum PTH decrease within 30 mins (Figure 7' below).

Figure 7'. PTH changes in unilateral parathyroidectomy rat (left) or mice (right)

However, in the nude mice transplantation study, we found the serum PTH level of nude mice transplanted with human parathyroid gland (PTG) graft decreased 5 mins after the removal of the graft, and recovered to previous level at 15 min (Figure 8' below). This phenomenon is observed because the endogenous parathyroid gland was not affected in the nude mice transplanted with human PTG graft; therefore, the endogenous PTG compensates for optogenetically induced PTH suppression (Figure 8' below).

Figure 8'. PTH changes in PTG graft removed nude mice

Taken together, our data consistently demonstrates that the compensation of PTH existed in animal models. We agree with the reviewer that the degree of compensation was dependent on the residue endogenous PTG mass.

Finally, I agree with Reviewer 2's comment that the epifluorescence image quality is poor and is difficult to interpret as presented. In certain cases, double labelling would have been very helpful. For example, in figure 3A, a white arrow indicates a transduced cell where intracellular calcium flux is being measured. It would be helpful to show the GFP channel for the same images so a positively transduced cell can be distinguished from a non-transduced cell; then a comparison flux trace (transduced vs non-transduced) could be shown to demonstrate specificity. Throughout, formal quantitation metrics should be shown to support the assertions of co-localization (figure 6g), successful expression levels (figure 6 b), etc.

Response: Thanks for the comment. As suggested, we now show the calcium fluorescence dynamic of the transduced and non-transduced cells in the same images. The results show that uninfected cells do not respond to light illumination, which is indicated by unchanged intracellular calcium dynamic (Figure 9' below).

Figure 9'. Uninfected cells do not response to light stimulation

As for the quantitation of the colocalization in figure 6b, we have performed cryosection of the virus transfected human parathyroid graft and performed immunofluorescence with PTH antibody. The percentage of double positive cells is 44.0% and 49.3% in eYFP and ChETA groups, respectively (Figure 10' and Figure 11' below). The figure 6g demonstrates the state of the transplanted graft after 4 weeks of transplantation which is the same graft shown in 6b in the manuscript. Therefore, quantification of 6g data is not shown.

Figure 10'. Co-localization of PTH and GFP in eYFP, ChETA and eYFP groups

Figure 11'. Quantification of eYFP and ChETA infection efficiency

Attached letter

Source data and the arithmetic for how the normalized values are derived for figure 3, 5, 6 and 7

Figure 3f-k

In general, the normalization calculation process is as follow:

$$\text{Normalized data} = \frac{\text{value of each group (pre/post/1h-post, respectively)}}{\text{average value of control (pre or post or 1h-post)}}$$

Here we take the **PKA data (correlated to Figure 3f)** as an example to demonstrate the calculation process. Normalization of the other pathway components (cAMP, PLA2, AA...) were achieved using the same calculation method.

1) For Pre (labeled light gray in the form)

First, we calculated the average mean of Control (Pre): $(369.794+335.683+333.891+324.171+331.506)/5=346.456$; Then, we divided all Pre data with 346.456 to obtain normalized data:

$$\begin{array}{lll} \text{Control}^{\text{Pre}_1}=369.794/346.456=1.0674; & \text{Control}^{\text{Pre}_2}=335.683/346.456=0.9689; & \text{Control}^{\text{Pre}_3}=333.891/346.456=0.9637; \dots \\ \text{ETA}^{\text{Pre}_1}=376.963/346.456=1.0881; & \text{ETA}^{\text{Pre}_2}=333.353/346.456=0.9622; & \text{ETA}^{\text{Pre}_3}=343.151/346.456=0.9905; \dots \\ \text{eYFP}^{\text{Pre}_1}=334.907/346.456=0.9667; & \text{eYFP}^{\text{Pre}_2}=338.61/346.456=0.9774; & \text{eYFP}^{\text{Pre}_3}=334.369/346.456=0.9651; \dots \end{array}$$

2) For Post (labeled light blue in the form)

Average mean of Control (Post): $(350.319+329.411+302.528+295.491+313.269)/5=327.419$;

Then, we divided all Post data with 327.419 to obtain the normalized data:

$$\begin{array}{lll} \text{Control}^{\text{Post}_1}=350.319/327.419=1.0699; & \text{Control}^{\text{Post}_2}=329.411/327.419=1.0061; & \text{Control}^{\text{Post}_3}=302.528/327.419=0.9239; \dots \\ \text{ETA}^{\text{Post}_1}=142.368/327.419=0.43481; & \text{ETA}^{\text{Post}_2}=132.749/327.419=0.4054; & \text{ETA}^{\text{Post}_3}=138.007/327.419=0.4215; \dots \\ \text{eYFP}^{\text{Post}_1}=328.156/327.419=1.0023; & \text{eYFP}^{\text{Post}_2}=331.143/327.419=1.0114; & \text{eYFP}^{\text{Post}_3}=326.842/327.419=0.9982; \dots \end{array}$$

3) For 1h-Post (labeled light orange in the form)

Average mean of Control (1h-Post): $(325.468+330.008+287.474+271.305+325.279)/5=314.317$;

Then, we divided all Post data with 314.317 to obtain the normalized data:

Control^{1h-Post}₁= $325.468/314.317=1.0355$; Control^{1h-Post}₂= $330.008/314.317=1.0499$; Control^{1h-Post}₃= $287.474/314.317=0.9146$; ...

ETA^{1h-Post}₁= $340.141/314.317 =1.0822$; ETA^{1h-Post}₂= $345.48/314.317 =1.0991$; ETA^{1h-Post}₃= $324.682/314.317 =1.0329$; ...

eYFP^{1h-Post}₁= $360.176/314.317 =1.1459$; eYFP^{1h-Post}₂= $292.91/314.317 =0.9319$; eYFP^{1h-Post}₃= $288.191/314.317 =0.9169$; ...

PKA non-normalized data (relates to figure 3f)

PKA	Control					ETA					eYFP				
Pre	369.794	335.683	333.891	324.171	331.506	376.963	333.353	343.151	319.286	362.319	334.9075	338.61	334.369	293.743	341.425
Post	350.319	329.411	302.528	295.491	313.269	142.368	132.749	138.007	175.035	168.545	328.156	331.143	326.842	312.615	304.984
1h-post	325.468	330.008	287.474	271.305	325.279	340.141	345.48	324.682	325.285	317.01	360.176	292.91	288.191	300.734	312.678

PKA Normalized (relates to figure 3f)

PKA	Control					ETA					eYFP				
Pre	1.0674	0.9689	0.9637	0.9357	0.9568	1.0881	0.9622	0.9905	0.9216	1.0458	0.9667	0.9774	0.9651	0.8479	0.9855
Post	1.0699	1.0061	0.9239	0.9025	0.9568	0.4348	0.4054	0.4215	0.5346	0.5148	1.0023	1.0114	0.9982	0.9548	0.9315
1h-post	1.0355	1.0499	0.9146	0.8632	1.0349	1.0822	1.0991	1.0329	1.0349	1.0086	1.1459	0.9319	0.9169	0.9568	0.9948

All other non-normalized or normalized data related figure 3 are listed below.

cAMP non-normalized data (relates to figure 3g)

cAMP	Control					ETA					eYFP				
Pre	127.043	124.91	127.49	121.869	138.663	123.91	121.087	128.697	126.02	116.941	128.352	128.421	112.674	126.01	127.097
Post	133.723	133.517	131.554	139.438	132.4453	105.213	101.184	90.267	98.534	100.348	124.978	132.181	126.734	127.127	120.057
1h-post	127.526	132.139	133.104	131.948	116.932	117.677	117.988	121.569	126.4	130.21	121.741	128.2831	129.006	130.064	118.677

cAMP Normalized (relates to figure 3g)

cAMP	Control					ETA					eYFP				
Pre	1.0045	0.9876	1.0079	0.9635	1.0963	0.9797	0.9574	1.0175	0.9963	0.9246	1.0148	1.0153	0.8908	0.9963	1.0049
Post	1.0059	1.1044	0.9897	1.0489	0.9964	0.7915	0.7612	0.6791	0.7412	0.7549	0.9402	0.9944	0.9534	0.9563	0.9032
1h-post	0.9741	1.0093	1.0167	1.0079	0.8932	0.8988	0.9012	0.9286	0.9655	0.9946	0.9299	0.9799	0.9854	0.9935	0.9065

PLA2 non-normalized data (relates to figure 3h)

PLA2	Control					ETA					eYFP				
Pre	32.4561	32.8889	21.4176	25.9165	28.1848	15.6329	28.1603	33.7962	35.7121	42.7716	15.1222	16.9896	17.4874	24.136	27.9016
Post	21.3582	17.9973	25.1926	28.9613	27.5136	151.319	211.632	125.106	121.543	137.997	16.5856	14.5061	20.8633	22.7648	26.609
1h-post	16.0588	15.3316	24.817	14.0094	18.7695	15.683	15.3023	23.1759	14.1292	18.2265	20.9885	25.6062	14.9061	14.1298	17.321

PLA2 Normalized (relates to figure 3h)

PLA2	Control					ETA					eYFP				
Pre	1.1223	1.1372	0.7406	0.8961	0.9746	0.5406	0.9737	1.1686	1.2349	1.4789	0.7027	0.5875	0.8126	0.8346	0.9648
Post	0.9925	0.8363	1.1707	1.3458	1.2785	5.2324	7.3178	4.3259	5.6479	6.4125	0.7707	0.6741	0.9695	1.0578	1.2365
1h-post	0.9715	0.9275	1.5013	0.8475	1.1355	0.9488	0.9258	1.4021	0.8548	1.1026	0.9753	1.5491	0.6927	0.8548	1.0479

AA non-normalized data (relates to figure 3i)

AA	Control					ETA					eYFP				
Pre	115.479	166.708	168.484	155.406	174.088	136.795	127.061	195.555	155.759	155.759	170.616	164.435	215.805	155.759	180.807
Post	184.613	146.743	212.11	169.319	218.235	353.149	283.305	435.072	365.019	330.545	214.384	202.021	158.964	174.521	190.01
1h-post	155.979	148.874	212.749	178.904	171.401	205.999	158.608	258.294	208.049	206.402	190.724	152.427	195.271	160.719	187.424

AA Normalized (relates to figure 3i)

AA	Control					ETA					eYFP				
Pre	0.7688	1.1098	1.1217	1.0346	1.1589	0.9107	0.8459	1.3019	1.0369	1.0369	1.1359	1.0947	1.4367	1.0369	1.2037
Post	1.019	0.8099	1.1708	0.9346	1.2046	1.9493	1.5638	2.4015	2.0148	1.8245	1.1835	1.1153	0.8776	0.9635	1.0489
1h-post	0.9041	0.8629	1.2331	1.0369	0.9935	1.1939	0.9193	1.4971	1.2059	1.1963	1.1055	0.8835	1.1318	0.9315	1.0863

12-LO non-normalized data (relates to figure 3j)

12-LO	Control					ETA					eYFP				
Pre	2.8319	3.1516	1.5735	2.6061	2.4305	4.0217	2.9769	2.8209	2.6398	2.4823	3.2356	2.9202	4.492	1.5901	2.657
Post	2.5937	2.4225	2.3419	4.5372	3.0602	31.5683	24.3806	25.1858	25.833716	20.2166	2.9283	3.1469	5.4763	4.9422	4.8198
1h-post	2.3337	2.9559	2.2661	4.118	2.5799	2.8097	4.8579	3.2045	2.3464	2.6242	4.2139	4.5286	5.2067	4.5959	5.6349

12-LO Normalized (relates to figure 3j)

12-LO	Control					ETA					eYFP				
Pre	1.1242	1.2511	0.6246	1.0346	0.9649	1.5966	1.1818	1.1199	1.0479	0.9854	1.2845	1.1593	1.7833	0.6312	1.0548
Post	1.0574	0.9876	0.9547	1.8496	1.2475	12.8693	9.9391	10.2674	10.5315	8.2416	1.1938	1.2829	2.2325	2.0148	1.9649
1h-post	0.9264	1.1734	0.8996	1.635	1.0242	1.1154	1.9285	1.2721	0.9315	1.0418	1.6729	1.7978	2.0669	1.8245	2.2369

15-LO non-normalized data (relates to figure 3k)

15-LO	Control					ETA					eYFP				
Pre	61.7937	64.8175	61.5418	84.8274	47.1334	68.9248	76.7613	64.6663	93.4298	84.5374	72.6037	72.2509	80.2135	97.7312	65.0374
Post	62.3985	66.6822	63.6079	91.0546	79.2846	156.916	157.546	123.856	142.943	122.184	77.2149	76.4589	77.5677	101.776	68.3271
1h-post	63.8599	63.5324	58.5179	90.0178	83.5403	86.9917	84.7491	80.6166	63.585	84.8664	78.2732	74.6447	66.6822	84.8763	64.1429

15-LO Normalized (relates to figure 3k)

15-LO	Control					ETA					eYFP				
Pre	0.9852	1.0334	0.9812	1.3525	0.7515	1.0989	1.2239	1.031	1.4896	1.3479	1.1576	1.1519	1.2789	1.5582	1.0369
Post	0.9716	1.0383	0.9905	1.4179	1.2346	2.4434	2.4532	1.9286	2.2258	1.9026	1.2024	1.1906	1.2078	1.5848	1.0639
1h-post	1.0303	1.025	0.9441	1.4524	1.3479	1.4035	1.3674	1.3007	1.0259	1.3693	1.2629	1.2043	1.0759	1.3694	1.0349

Figure 5g

non-normalized PTH data (relates to figure 5g)

	eYFP						ChETA					
B	106.708	101.972	143.772	136.096	129.024	69.392	74.084	136.552	79.56	130.524	99.288	192.652
A0	111.334	112.755	116.821	125.181	146.145	114.49	91.9383	110.955	111.389	105.255	90.9522	133.512
A5	146.145	141.279	99.7578	175.011	167.472	105.378	95.4514	71.1762	97.5975	98.2343	90.9379	77.4359
A15	96.3486	161.583	144.102	116.5	136.275	175.011	77.0943	77.5099	99.4708	89.8975	101.276	93.0724
A30	105.549	80.2798	112.411	173.406	103.159	178.027	82.4191	114.482	106.751	116.948	169.823	163.149

Normalized PTH data (relates to figure 5g)

	eYFP						ChETA					
B	1	1	1	1	1	1	1	1	1	1	1	1
A0	0.9724	0.9848	1.0204	1.0934	1.2765	1	0.774	0.9341	0.9378	0.8861	0.7657	1.124
A5	1.2765	1.2339	0.8713	1.5286	1.4628	0.9204	0.8036	0.5992	0.8217	0.827	0.7656	0.6519
A15	0.8415	1.4113	1.2586	1.0176	1.1903	1.5286	0.6491	0.6526	0.8374	0.7568	0.8526	0.7836
A30	0.9219	0.7012	0.9818	1.5146	0.901	1.5549	0.6939	0.9638	0.8987	0.9846	1.4297	1.3735

The normalization calculation process is as follow:

$$\text{Normalized data} = \frac{\text{value of eYFP or ChETA groups (A0 or A5 or A15 or A30)}}{\text{average value of baseline B (eYFP or ChETA)}}$$

To visualize the fold changes after light stimulation more clearly, we defined the normalized baseline as 1.

1) For eYFP (labeled light gray in the form)

First, we calculated average mean of the baseline (B) of eYFP: $(106.708+101.972+143.772+136.096+129.024+69.392)/6=114.494$; Second, to visualize the fold changes after light stimulation more clearly, we defined the normalized baseline as 1. Then, we divided all data with the average mean of the

baseline (114.494) to obtain the normalized data:

$$\begin{array}{lll} A0^{eYFP}_1=111.334/114.494=0.9724; & A0^{eYFP}_2=112.755/114.494=0.9848; & A0^{eYFP}_3=116.821/114.494=1.0204; \dots \\ A5^{eYFP}_1=146.145/114.494=1.2765; & A5^{eYFP}_2=141.279/114.494=1.2339; & A5^{eYFP}_3=99.7578/114.494=0.8713; \dots \\ A15^{eYFP}_1=96.3486/114.494=0.8415; & A15^{eYFP}_2=161.583/114.494=1.4113; & A15^{eYFP}_3=144.102/114.494=1.2586; \dots \\ A30^{eYFP}_1=105.549/114.494=0.9219; & A30^{eYFP}_2=80.2798/114.494=0.7012; & A30^{eYFP}_3=112.411/114.494=0.9818; \dots \end{array}$$

2) For ChETA (labeled light orange in the form)

First, we calculated average mean of the baseline (B) of ChETA: $(74.084+136.552+79.56+130.524+99.288+192.652)/6=118.777$; to visualize the fold changes after light stimulation more clearly, we defined the normalized baseline as 1. Then, we divided all data with the average mean of the baseline (118.777) to obtain the normalized data:

$$\begin{array}{lll} A0^{ChETA}_1=91.9383/118.777=0.774; & A0^{ChETA}_2=110.955/118.777=0.9341; & A0^{ChETA}_3=111.389/118.777=0.9378; \dots \\ A5^{ChETA}_1=95.4514/118.777=0.8036; & A5^{ChETA}_2=71.1762/118.777=0.5992; & A5^{ChETA}_3=97.5975/118.777=0.8217; \dots \\ A15^{ChETA}_1=77.0943/118.777=0.6491; & A15^{ChETA}_2=77.5099/118.777=0.6526; & A15^{ChETA}_3=99.4708/118.777=0.8374; \dots \\ A30^{ChETA}_1=82.4191/118.777=0.6939; & A30^{ChETA}_2=114.482/118.777=0.9638; & A30^{ChETA}_3=106.751/118.777=0.8987; \dots \end{array}$$

Figure 5h

non-normalized Calcium data (relates to figure 5h)

	eYFP						ChETA					
B	2.5776	1.7723	2.4753	2.3433	2.6585	2.3647	2.38	2.5148	2.2138	2.6303	2.5915	2.7496
A0	2.9725	2.4577	2.6659	2.4555	2.437	2.4454	3.0446	2.3438	2.4008	2.4008	2.3671	2.4026
A5	2.8916	2.5069	2.333	2.4811	2.2851	2.3731	2.3859	2.5024	2.3383	2.2691	2.5831	2.8274
A15	2.5174	2.2714	2.258	2.4407	2.1735	2.3888	2.6838	2.2699	2.1449	2.4715	2.4701	2.2659
A30	2.4341	2.9781	2.7121	2.7418	3.0127	2.1219	2.0921	1.6092	2.0921	1.4163	2.4098	1.3394

Normalized Calcium data (relates to figure 5h)

	eYFP						ChETA					
B	1	1	1	1	1	1	1	1	1	1	1	1
A0	1.2595	1.0369	1.1249	1.0361	1.0283	1.0318	1.2129	0.9338	0.9565	0.9565	0.9431	0.9572
A5	1.2253	1.0578	0.9844	1.0469	0.9642	1.0013	0.9506	0.9969	0.9316	0.904	1.0291	1.1265
A15	1.0667	0.9584	0.9528	1.0298	0.9171	1.0079	1.0692	0.9043	0.8546	0.9846	0.9841	0.9028
A30	1.0314	1.2566	1.1444	1.1569	1.2712	0.8953	0.8335	0.6411	0.8335	0.5643	0.9601	0.5336

The normalization calculation process is as follow:

$$\text{Normalized data} = \frac{\text{value of eYFP or ChETA groups (A0 or A5 or A15 or A30)}}{\text{average value of baseline B (eYFP or ChETA)}}$$

To visualize the fold changes after light stimulation more clearly, we defined the normalized baseline as 1.

1) For eYFP (labeled light gray in the form)

First, we calculated the average mean of the baseline (B) of eYFP: $(2.5776+1.7723+2.4753+2.3433+2.6585+2.3647)/6=2.3653$; Second, to visualize the fold changes after light stimulation more clearly, we defined the normalized baseline as 1. Then, we divided all data with the average mean of the

baseline (2.3653) to obtain the normalized data:

$$\begin{array}{lll} A0^{eYFP}_1=2.9725/2.3653=1.2595; & A0^{eYFP}_2=2.4577/2.3653=1.0369; & A0^{eYFP}_3=2.6659/2.3653=1.1249; \dots \\ A5^{eYFP}_1=2.8916/2.3653=1.2253; & A5^{eYFP}_2=2.5069/2.3653=1.0578; & A5^{eYFP}_3=2.333/2.3653=0.9844; \dots \\ A15^{eYFP}_1=2.5174/2.3653=1.0667; & A15^{eYFP}_2=2.2714/2.3653=0.9584; & A15^{eYFP}_3=2.258/2.3653=0.9528; \dots \\ A30^{eYFP}_1=2.4341/2.3653=1.0314; & A30^{eYFP}_2=2.9781/2.3653=1.2566; & A30^{eYFP}_3=2.7121/2.3653=1.1444; \dots \end{array}$$

2) For ChETA (labeled light orange in the form)

First, we calculated the average mean of the baseline (B) of ChETA: $(2.38+2.5148+2.2138+2.6303+2.5915+2.7496)/6=2.5133$; to visualize the fold changes after light stimulation more clearly, we defined the normalized baseline as 1. Then, we divided all data with the average mean of the baseline (2.5133) to obtain the normalized data:

$$\begin{array}{lll} A0^{ChETA}_1=3.0446/2.5133=1.2129; & A0^{ChETA}_2=2.3438/2.5133=0.9338; & A0^{ChETA}_3=2.4008/2.5133=0.9565; \dots \\ A5^{ChETA}_1=2.3859/2.5133=0.9506; & A5^{ChETA}_2=2.5024/2.5133=0.9969; & A5^{ChETA}_3=2.3383/2.5133=0.9316; \dots \\ A15^{ChETA}_1=2.6838/2.5133=1.0692; & A15^{ChETA}_2=2.2699/2.5133=0.9043; & A15^{ChETA}_3=2.1449/2.5133=0.8546; \dots \\ A30^{ChETA}_1=2.0921/2.5133=0.8335; & A30^{ChETA}_2=1.6092/2.5133=0.6411; & A30^{ChETA}_3=2.0921/2.5133=0.8335; \dots \end{array}$$

Figure 6d

non-normalized PTH data (relates to figure 6d)

1st	Control					ETA					eYFP				
Base	20.5999	20.5121	20.2426	20.4363	19.7632	18.8439	19.7918	19.3371	19.5369	20.1575	21.6545	20.9692	19.7953	22.0766	19.8386
Pre	20.5999	20.4901	20.039	18.9418	20.0248	19.2273	18.7258	19.3371	21.5748	20.5293	22.5602	22.7331	19.3514	19.9593	19.7039
2h	20.1273	20.9633	19.5575	20.2561	19.7641	13.8194	13.8194	15.7541	12.4631	15.4259	21.8867	21.2296	20.8516	20.5563	19.0976
4h	20.5349	20.556	20.2561	19.7636	19.5575	21.9743	22.3005	18.739	17.7181	17.7438	23.8277	19.1829	22.7219	18.9561	20.6249
2nd															
Base	20.058	21.0619	20.4601	20.6104	20.26	20.6673	20.2678	21.1306	20.0606	21.6823	22.9716	22.7933	21.443	22.0025	20.1873
Pre	20.5385	20.5815	19.4009	18.5749	18.9799	19.0958	19.376	17.726	18.1752	18.3155	21.476	23.4285	22.7135	20.9833	19.706
2h	20.6852	20.4348	20.2545	19.5612	21.6136	20.7502	18.5209	19.3478	21.0974	18.2955	22.002	22.2751	21.5898	20.9814	20.1454
4h	20.6031	20.4953	19.5576	19.4366	20.2198	21.1535	20.598	19.5703	20.6694	19.6899	22.005	21.8192	20.9826	22.9481	20.6211

Normalized PTH data (relates to figure 6d)

1st	Control					ETA					eYFP				
Base	1.0021	0.9979	0.9848	0.9942	0.9614	0.9595	0.9568	0.9846	0.9948	0.9245	1.0346	1.0019	0.9458	1.0548	0.9479
Pre	1.0021	0.9968	0.9749	0.9215	0.9742	0.9789	0.9535	0.9846	1.0985	1.0453	1.0779	1.0862	0.9246	0.9536	0.9414
2h	0.9791	1.0198	0.9514	0.9854	0.9615	0.7036	0.7036	0.8021	0.6346	0.7854	1.0457	1.0143	0.9963	0.9821	0.9125
4h	0.9989	1	0.9854	0.9615	0.9514	1.1189	1.1355	0.9541	0.9021	0.9035	1.1384	1.1554	1.0856	1.0012	0.9854
2nd															
Base	0.9756	1.0244	0.9951	1.0025	0.9854	1.0523	1.0319	0.9231	1.0214	0.9512	1.0975	1.089	1.0245	1.0512	0.9645
Pre	0.9989	1.001	0.9436	0.9035	0.9231	0.9723	0.9866	0.9025	0.9254	0.9326	1.0261	1.1194	1.0852	1.0025	0.9415
2h	1.006	0.9939	0.9851	0.9514	1.0512	0.9547	0.943	0.9851	0.9215	0.9315	1.0512	1.0643	1.0315	1.0025	0.9625
4h	1.0021	0.9969	0.9512	0.9454	0.9835	1.0771	1.0488	0.9965	1.0524	1.0025	1.0514	1.0425	1.0025	1.0964	0.9852

The normalization calculation process is as follow:

$$\text{Normalized data} = \frac{\text{value of each Control, ETA or eYFP groups (Pre or 2h or 4h)}}{\text{average value of base (Control or ETA or eYFP)}}$$

1) For Control (labeled light gray in the form)

First, we calculated the average mean of baseline (Base) of Control: $(20.5999+20.5121+20.2426+20.4363+19.7632)/5=20.56$; Then, we divided all data in the Control (Base^{1st}, Pre^{1st}, 2h^{1st}, 4h^{1st}, Base^{2nd}, Pre^{2nd}, 2h^{2nd}, 4h^{2nd}) with the average mean of the baseline (20.56) to obtain the normalized data:

$$\begin{array}{lll} \text{Base}^{1\text{st}}_1=20.5999/20.56=1.0021; & \text{Base}^{1\text{st}}_2=20.5121/20.56=0.9979; & \text{Base}^{1\text{st}}_3=20.2426/20.56=0.9848; \dots \\ \text{Pre}^{1\text{st}}_1=20.5999/20.56=1.0021; & \text{Pre}^{1\text{st}}_2=20.4901/20.56=0.9968; & \text{Pre}^{1\text{st}}_3=20.039/20.56=0.9749; \dots \\ 2\text{h}^{1\text{st}}_1=20.1273/20.56=0.9791; & 2\text{h}^{1\text{st}}_2=20.9633/20.56=1.0198; & 2\text{h}^{1\text{st}}_3=19.5575/20.56=0.9514; \dots \\ 4\text{h}^{1\text{st}}_1=20.5349/20.56=0.9989; & 4\text{h}^{1\text{st}}_2=20.556/20.56=1; & 4\text{h}^{1\text{st}}_3=20.2561/20.56=0.9854; \dots \end{array}$$

$$\begin{array}{lll} \text{Base}^{2\text{nd}}_1=20.058/20.56=0.9756; & \text{Base}^{2\text{nd}}_2=21.0619/20.56=1.0244; & \text{Base}^{2\text{nd}}_3=20.4601/20.56=0.9951; \dots \\ \text{Pre}^{2\text{nd}}_1=20.5385/20.56=0.9989; & \text{Pre}^{2\text{nd}}_2=20.5815/20.56=1.001; & \text{Pre}^{2\text{nd}}_3=19.4009/20.56=0.9436; \dots \\ 2\text{h}^{2\text{nd}}_1=20.6852/20.56=1.0061; & 2\text{h}^{2\text{nd}}_2=20.4348/20.56=0.9939; & 2\text{h}^{2\text{nd}}_3=20.2545/20.56=0.9851; \dots \\ 4\text{h}^{2\text{nd}}_1=20.6031/20.56=1.0021; & 4\text{h}^{2\text{nd}}_2=20.4953/20.56=0.9969; & 4\text{h}^{2\text{nd}}_3=19.5576/20.56=0.9512; \dots \end{array}$$

2) For ETA (labeled light blue in the form)

First, we calculated the average mean of the baseline (Base) of ETA: $(18.8439+19.7918+19.3371+19.5369+20.1575)/5=19.64$; Then, we divided all data in the ETA (Base^{1st}, Pre^{1st}, 2h^{1st}, 4h^{1st}, Base^{2nd}, Pre^{2nd}, 2h^{2nd}, 4h^{2nd}) with the average mean of the baseline (19.64) to obtain the normalized data:

$$\begin{array}{lll} \text{Base}^{1\text{st}}_1=18.8439/19.64=0.9595; & \text{Base}^{1\text{st}}_2=19.7918/19.64=0.9568; & \text{Base}^{1\text{st}}_3=19.3371/19.64=0.9846; \dots \\ \text{Pre}^{1\text{st}}_1=19.2273/19.64=0.9789; & \text{Pre}^{1\text{st}}_2=18.7258/19.64=0.9535; & \text{Pre}^{1\text{st}}_3=19.3371/19.64=0.9846; \dots \\ 2\text{h}^{1\text{st}}_1=13.8194/19.64=0.7036; & 2\text{h}^{1\text{st}}_2=13.8194/19.64=0.7036; & 2\text{h}^{1\text{st}}_3=15.7541/19.64=0.8021; \dots \\ 4\text{h}^{1\text{st}}_1=21.9743/19.64=1.1189; & 4\text{h}^{1\text{st}}_2=22.3005/19.64=1.1355; & 4\text{h}^{1\text{st}}_3=18.739/19.64=0.9541; \dots \end{array}$$

$\text{Base}^{2\text{nd}}_1=20.6673/19.64=1.0523;$	$\text{Base}^{2\text{nd}}_2=20.2678/19.64=1.0319;$	$\text{Base}^{2\text{nd}}_3=21.1306/19.64=0.9231; \dots$
$\text{Pre}^{2\text{nd}}_1=19.0958/19.64=0.9723;$	$\text{Pre}^{2\text{nd}}_2=19.376/19.64=0.9866;$	$\text{Pre}^{2\text{nd}}_3=17.726/19.64=0.9025; \dots$
$2\text{h}^{2\text{nd}}_1=20.7502/19.64=0.9547;$	$2\text{h}^{2\text{nd}}_2=18.5209/19.64=0.943;$	$2\text{h}^{2\text{nd}}_3=19.3478/19.64=0.9851; \dots$
$4\text{h}^{2\text{nd}}_1=21.1535/19.64=1.0771;$	$4\text{h}^{2\text{nd}}_2=20.598/19.64=1.0488;$	$4\text{h}^{2\text{nd}}_3=19.5703/19.64=0.9965; \dots$

3) For eYFP (labeled light orange in the form)

First, we calculated the average mean of the baseline (Base) of eYFP: $(21.6545+20.9692+19.7953+22.0766+19.8386)/5=20.93$; Then, we divided all data in the eYFP ($\text{Base}^{1\text{st}}, \text{Pre}^{1\text{st}}, 2\text{h}^{1\text{st}}, 4\text{h}^{1\text{st}}, \text{Base}^{2\text{nd}}, \text{Pre}^{2\text{nd}}, 2\text{h}^{2\text{nd}}, 4\text{h}^{2\text{nd}}$) with the average mean of the baseline (20.93) to obtain the normalized data:

$\text{Base}^{1\text{st}}_1=21.6545/20.93=1.0346;$	$\text{Base}^{1\text{st}}_2=20.9692/20.93=1.0019;$	$\text{Base}^{1\text{st}}_3=19.7953/20.93=0.9458; \dots$
$\text{Pre}^{1\text{st}}_1=22.5602/20.93=1.0779;$	$\text{Pre}^{1\text{st}}_2=22.7331/20.93=1.0862;$	$\text{Pre}^{1\text{st}}_3=19.3514/20.93=0.9246; \dots$
$2\text{h}^{1\text{st}}_1=21.8867/20.93=1.0457;$	$2\text{h}^{1\text{st}}_2=21.2296/20.93=1.0143;$	$2\text{h}^{1\text{st}}_3=20.8516/20.93=0.9963; \dots$
$4\text{h}^{1\text{st}}_1=23.8277/20.93=1.1384;$	$4\text{h}^{1\text{st}}_2=19.1829/20.93=1.1554;$	$4\text{h}^{1\text{st}}_3=22.7219/20.93=1.0856; \dots$
$\text{Base}^{2\text{nd}}_1=22.9716/20.93=1.0975;$	$\text{Base}^{2\text{nd}}_2=22.7933/20.93=1.089;$	$\text{Base}^{2\text{nd}}_3=21.443/20.93=1.0245; \dots$
$\text{Pre}^{2\text{nd}}_1=21.476/20.93=1.0261;$	$\text{Pre}^{2\text{nd}}_2=23.4285/20.93=1.1194;$	$\text{Pre}^{2\text{nd}}_3=22.7135/20.93=1.0852; \dots$
$2\text{h}^{2\text{nd}}_1=22.002/20.93=1.0512;$	$2\text{h}^{2\text{nd}}_2=22.2751/20.93=1.0643;$	$2\text{h}^{2\text{nd}}_3=21.5898/20.93=1.0315; \dots$
$4\text{h}^{2\text{nd}}_1=22.005/20.93=1.0514;$	$4\text{h}^{2\text{nd}}_2=21.8192/20.93=1.0425;$	$4\text{h}^{2\text{nd}}_3=20.9826/20.93=1.0025; \dots$

Figure 6h

non-normalized PTH data (relates to figure 6h)

	eYFP						ETA					
B	39.1706	39.1154	38.5816	40.1266	41.138	41.7179	35.8029	37.4616	43.5173	40.0795	42.2299	43.8569
A0	38.8682	39.3644	40.7838	43.7025	51.0212	39.97	31.3402	37.8224	37.9706	35.8797	35.053	41.7203
A5	30.7426	49.3223	34.8268	61.0986	58.4669	28.7948	32.5377	24.2627	33.2693	30.9991	26.3965	17.6435
A15	34.5177	29.6396	56.4108	40.6718	47.5753	49.1076	26.2801	26.3742	33.9078	30.6445	31.7268	26.4218
A30	50.4498	28.0268	39.2443	28.5622	36.014	54.1577	40.2422	30.9268	36.3894	53.2319	39.8655	42.905

Normalized PTH data (relates to figure 6h)

	eYFP						ETA					
B	1	1	1	1	1	1	1	1	1	1	1	1
A0	0.9724	0.9848	1.0204	1.0934	1.2765	1	0.774	0.9341	0.9378	0.8861	0.8657	1.0304
A5	0.7691	1.2339	0.8713	1.5286	1.4628	0.7204	0.8036	0.5992	0.8217	0.7656	0.6519	0.4357
A15	0.8636	0.7415	1.4113	1.0176	1.1903	1.2286	0.6491	0.6514	0.8374	0.7568	0.7836	0.6526
A30	1.2622	0.7012	0.9818	0.7146	0.901	1.3549	0.9939	0.7638	0.8987	1.3147	0.9846	1.0596

The normalization calculation process is as follow:

$$\text{Normalized data} = \frac{\text{value of eYFP or ChETA groups (A0 or A5 or A15 or A30)}}{\text{average value of baseline B (eYFP or ChETA)}}$$

To visualize the fold changes after light stimulation more clearly, we defined the normalized baseline as 1.

1) For eYFP (labeled light gray in the form)

First, we calculated the average mean of the baseline (B) of eYFP: $(39.1706+39.1154+38.5816+40.1266+41.138+41.7179)/6=39.975$; Second, to visualize the fold changes after light stimulation more clearly, we defined the normalized baseline as 1. Then, we divided all data with the average mean

of the baseline (39.975) to obtain the normalized data:

$$\begin{array}{lll} A0^{eYFP_1}=38.8682/39.975=0.9724; & A0^{eYFP_2}=39.3644/39.975=0.9848; & A0^{eYFP_3}=40.7838/39.975=1.0204; \dots \\ A5^{eYFP_1}=30.7426/39.975=0.7691; & A5^{eYFP_2}=49.3223/39.975=1.2339; & A5^{eYFP_3}=34.8268/39.975=0.8713; \dots \\ A15^{eYFP_1}=34.5177/39.975=0.8636; & A15^{eYFP_2}=29.6396/39.975=0.7415; & A15^{eYFP_3}=56.4108/39.975=1.4113; \dots \\ A30^{eYFP_1}=50.4498/39.975=1.2622; & A30^{eYFP_2}=28.0268/39.975=0.7012; & A30^{eYFP_3}=39.2443/39.975=0.9818; \dots \end{array}$$

2) For ETA (labeled light orange in the form)

First, we calculated the average mean of the baseline (B) of ChETA: $(35.8029+37.4616+43.5173+40.0795+42.2299+43.8569)/6=40.4914$; to visualize the fold changes after light stimulation more clearly, we defined the normalized baseline as 1. Then, we divided all data with the average mean of the baseline (40.4914) to obtain the normalized data:

$$\begin{array}{lll} A0^{ChETA_1}=31.3402/40.4914=0.774022; & A0^{ChETA_2}=37.8224/40.4914=0.9341; & A0^{ChETA_3}=37.9706/40.4914=0.9378; \dots \\ A5^{ChETA_1}=32.5377/40.4914=0.8036; & A5^{ChETA_2}=24.2627/40.4914=0.5992; & A5^{ChETA_3}=33.2693/40.4914=0.8217; \dots \\ A15^{ChETA_1}=26.2801/40.4914=0.6491; & A15^{ChETA_2}=26.3742/40.4914=0.6514; & A15^{ChETA_3}=33.9078/40.4914=0.8374; \dots \\ A30^{ChETA_1}=40.2422/40.4914=0.9939; & A30^{ChETA_2}=30.9268/40.4914=0.7638; & A30^{ChETA_3}=36.3894/40.4914=0.8987; \dots \end{array}$$

Figure 6i

non-normalized Calcium data (relates to figure 6i)

	eYFP						ETA					
B	1.7371	1.7107	1.6737	1.4568	1.4751	1.4819	1.7065	1.7104	1.8684	1.6556	1.9478	1.6021
A0	1.8321	1.6384	2.5673	2.9009	2.5727	2.7363	2.6477	2.8428	2.1674	3.4043	3.2069	2.1849
A5	1.9359	2.4613	2.6203	2.9181	2.6136	2.6881	2.1885	1.7348	2.839	2.965	2.3127	3.178
A15	2.1593	1.9883	2.6113	2.7331	2.2389	2.0666	1.8711	1.5736	1.4869	1.7133	1.7124	1.5708
A30	1.6296	1.9854	1.8081	1.8279	2.3244	2.246	1.8088	1.6376	1.4503	1.8133	1.7403	1.6969

Normalized Calcium data (relates to figure 6i)

	eYFP						ETA					
B	1	1	1	1	1	1	1	1	1	1	1	1
A0	1.1595	1.0369	1.6249	1.8361	1.6283	1.7318	1.5129	1.6338	1.2456	1.9565	1.8431	1.2557
A5	1.2253	1.5578	1.6584	1.8469	1.6542	1.7013	1.2506	0.9969	1.6316	1.704	1.3291	1.8265
A15	1.3667	1.2584	1.6528	1.7298	1.4171	1.3079	1.0692	0.9043	0.8546	0.9846	0.9841	0.9028
A30	1.0314	1.2566	1.1444	1.1569	1.4712	1.4215	1.0336	0.9411	0.8335	1.0422	1.0002	0.9753

The normalization calculation process is as follow:

$$\text{Normalized data} = \frac{\text{value of eYFP or ChETA groups (A0 or A5 or A15 or A30)}}{\text{average value of baseline B (eYFP or ChETA)}}$$

To visualize the fold changes after light stimulation more clearly, we defined the normalized baseline as 1.

3) For eYFP (labeled light gray in the form)

First, we calculated the average mean of the baseline (B) of eYFP: $(1.7371+1.7107+1.6737+1.4568+1.4751+1.4819)/6=1.5892$; Second, to visualize the fold changes after light stimulation more clearly, we defined the normalized baseline as 1. Then, we divided all data with the average mean of the

baseline (1.5892) to obtain the normalized data:

$$\begin{array}{lll} A0^{eYFP}_1=1.8321/1.5892=1.1595; & A0^{eYFP}_2=1.6384/1.5892=1.0369; & A0^{eYFP}_3=2.5673/1.5892=1.6249; \dots \\ A5^{eYFP}_1=1.9359/1.5892=1.2253; & A5^{eYFP}_2=2.4613/1.5892=1.5578; & A5^{eYFP}_3=2.6203/1.5892=1.6584; \dots \\ A15^{eYFP}_1=2.1593/1.5892=1.3667; & A15^{eYFP}_2=1.9883/1.5892=1.2584; & A15^{eYFP}_3=2.6113/1.5892=1.6528; \dots \\ A30^{eYFP}_1=1.6296/1.5892=1.0314; & A30^{eYFP}_2=1.9854/1.5892=1.2566; & A30^{eYFP}_3=1.8081/1.5892=1.1444; \dots \end{array}$$

4) For ETA (labeled light orange in the form)

First, we calculated the average mean of the baseline (B) of ChETA: $(1.7065+1.7104+1.8684+1.6556+1.9478+1.6021)/6=1.7485$; to visualize the fold changes after light stimulation more clearly, we defined the normalized baseline as 1. Then, we divided all data with the average mean of the baseline (1.7485) to obtain the normalized data:

$$\begin{array}{lll} A0^{ChETA}_1=2.6477/1.7485=1.5129; & A0^{ChETA}_2=2.8428/1.7485=1.6338; & A0^{ChETA}_3=2.1674/1.7485=1.2456; \dots \\ A5^{ChETA}_1=2.1885/1.7485=1.2506; & A5^{ChETA}_2=1.7348/1.7485=0.9969; & A5^{ChETA}_3=2.839/1.7485=1.6316; \dots \\ A15^{ChETA}_1=1.8711/1.7485=1.0692; & A15^{ChETA}_2=1.5736/1.7485=0.9043; & A15^{ChETA}_3=1.4869/1.7485=0.8546; \dots \\ A30^{ChETA}_1=1.8088/1.7485=1.0336; & A30^{ChETA}_2=1.6376/1.7485=0.9411; & A30^{ChETA}_3=1.4503/1.7485=0.8335; \dots \end{array}$$

Figure 7b

non-normalized PTH data (relates to figure 7b)

	PTG					ETA				
B	36.1234	38.3549	31.616	35.1397	33.7659	35.215	37.5631	35.8029	37.4616	43.5173
A5	30.2133	27.3563	39.8036	47.4117	27.4691	17.8353	18.0139	27.6278	26.8412	25.0704
A10	29.5128	36.5666	33.0229	43.2063	44.4102	23.3669	28.7701	31.7049	19.1794	27.9013
A15	29.9352	31.1678	35.8804	40.8285	32.4152	33.8871	40.0911	32.5454	35.4955	43.6565
2B	36.6846	38.3782	31.5868	33.2077	33.9026	39.2468	37.7419	36.621	38.4081	41.2144
2A5	29.9975	27.1609	29.5193	35.8821	30.3228	18.1835	31.0179	28.1598	28.1309	24.1538
2A10	40.2234	36.3769	32.7871	42.8977	29.7679	22.4264	29.3241	30.0434	21.8207	28.6312
2A15	29.7214	23.9951	35.6241	40.5369	18.2836	34.5487	40.8631	33.1721	36.1789	34.8372

Normalized PTH data (relates to figure 7b)

	PTG					ETA				
B	1	1	1	1	1	1	1	1	1	1
A5	0.8632	1.0959	1.1372	1.3546	0.7848	0.4705	0.4752	0.7288	0.708	0.6613
A10	0.8432	0.7816	0.9435	1.2345	1.2689	0.6164	0.7589	0.8363	0.5059	0.7359
A15	0.8553	1.0448	1.0252	1.1665	0.9261	0.8939	1.0575	0.8585	0.9363	1.1516
2B	1	1	1	1	1	1	1	1	1	1
2A5	0.8632	0.7816	0.8495	1.0326	0.8726	0.4706	0.8027	0.7288	0.728	0.6251
2A10	1.1575	1.0468	0.9435	1.2345	0.8566	0.5804	0.7589	0.7775	0.5647	0.7409
2A15	0.8553	0.6905	1.0252	1.1665	0.5261	0.8941	1.0575	0.8585	0.9363	0.9016

The normalization calculation process is as follow:

$$\text{Normalized data} = \frac{\text{value of PTG or ETA groups (A0 or A5 or A15 or A30)}}{\text{average value of baseline B (PTG or ETA)}}$$

To visualize the fold changes after light stimulation more clearly, we defined the normalized baseline as 1.

1) For PTG group

For Day 1 (data with gray background)

First, we calculated the average mean of the baseline (B) of PTG: $(36.1234+38.3549+31.616+35.1397+33.7659)/5=35$; Second, to visualize the fold changes after light stimulation more clearly, we defined the normalized baseline as 1. Then, we divided all data from different time points (A5, A10 or A15) with the average mean of the baseline (35) to obtain the normalized data:

$$\begin{aligned} A5^{\text{PTG}}_1 &= 30.2133/35 = 0.8632; & A5^{\text{PTG}}_2 &= 27.3563/35 = 0.7816; & A5^{\text{PTG}}_3 &= 39.8036/35 = 1.1372; \dots \\ A10^{\text{PTG}}_1 &= 29.5128/35 = 0.8432; & A10^{\text{PTG}}_2 &= 36.5666/35 = 1.0448; & A10^{\text{PTG}}_3 &= 33.0229/35 = 0.9435; \dots \\ A15^{\text{PTG}}_1 &= 29.9352/35 = 0.8553; & A15^{\text{PTG}}_2 &= 31.1678/35 = 0.8905; & A15^{\text{PTG}}_3 &= 35.8804/35 = 1.0252; \dots \end{aligned}$$

For Day 3 (data with orange background)

We calculated the average mean of the baseline (2B) of PTG: $(36.6846+38.3782+31.5868+33.2077+33.9026)/5=34.75$; Second, to visualize the fold changes after light stimulation more clearly, we defined the normalized baseline as 1. Then, we divided all data from different time points (2A5, 2A10 or 2A15) with the average mean of baseline 34.75 to gain normalized data:

$$\begin{aligned} 2A5^{\text{PTG}}_1 &= 29.9975/34.75 = 0.8632; & 2A5^{\text{PTG}}_2 &= 27.1609/34.75 = 0.7816; & 2A5^{\text{PTG}}_3 &= 29.5193/34.75 = 0.8495; \dots \\ 2A10^{\text{PTG}}_1 &= 40.2234/34.75 = 1.1575; & 2A10^{\text{PTG}}_2 &= 36.3769/34.75 = 1.0468; & 2A10^{\text{PTG}}_3 &= 32.7871/34.75 = 0.9435; \dots \\ 2A15^{\text{PTG}}_1 &= 29.7214/34.75 = 0.8553; & 2A15^{\text{PTG}}_2 &= 23.9951/34.75 = 0.6905; & 2A15^{\text{PTG}}_3 &= 35.6241/34.75 = 1.0252; \dots \end{aligned}$$

2) For ETA

For Day 1 (data with gray background)

First, we calculated the average mean of the baseline (B) of ETA: $(35.215+37.5631+35.8029+37.4616+43.5173)/5=37.91$; Second, to visualize the fold

changes after light stimulation more clearly, we defined the normalized baseline as 1. Then, we divided all data from different time points (A5, A10 or A15) with the average mean of the baseline (37.91) to obtain the normalized data:

$$\begin{array}{lll} A5^{\text{ETA}_1}=17.8353/37.91=0.4705; & A5^{\text{ETA}_2}=18.0139/37.91=0.4752; & A5^{\text{ETA}_3}=27.6278/37.91=0.7288; \dots \\ A10^{\text{ETA}_1}=23.3669/37.91=0.6164; & A10^{\text{ETA}_2}=28.7701/37.91=0.7589; & A10^{\text{ETA}_3}=31.7049/37.91=0.8363; \dots \\ A15^{\text{ETA}_1}=33.8871/37.91=0.8939; & A15^{\text{ETA}_2}=40.0911/37.91=1.0575; & A15^{\text{ETA}_3}=32.5454/37.91=0.8585; \dots \end{array}$$

For Day 3 (the data with orange background)

We calculated the average mean of the baseline (2B) of PTG: $(39.2468+37.7419+36.621+38.4081+41.2144)/5=38.64$; Second, to visualize the fold changes after light stimulation more clearly, we defined the normalized baseline as 1. Then, we divided all data from different time points (2A5, 2A10 or 2A15) with the average mean of the baseline (38.64) to obtain the normalized data:

$$\begin{array}{lll} 2A5^{\text{ETA}_1}=18.1835/38.64=0.4706; & 2A5^{\text{ETA}_2}=31.0179/38.64=0.8027; & 2A5^{\text{ETA}_3}=28.1598/38.64=0.7288; \dots \\ 2A10^{\text{ETA}_1}=22.4264/38.64=0.5804; & 2A10^{\text{ETA}_2}=29.3241/38.64=0.7589; & 2A10^{\text{ETA}_3}=30.0434/38.64=0.7775; \dots \\ 2A15^{\text{ETA}_1}=34.5487/38.64=0.8941; & 2A15^{\text{ETA}_2}=40.8631/38.64=1.0575; & 2A15^{\text{ETA}_3}=33.1721/38.64=0.8585; \dots \end{array}$$

Reviewers' Comments:

Reviewer #1:

Remarks to the Author:

1. The authors add a new experiment that shows that optogenetic manipulation of parathyroid cells not only affects PTH secretion but also leads to a number of changes in gene expression. This is a point well worth making. But, unfortunately, the authors also claim that changes in the levels of Rab8a mRNA and Rab11a mRNA mean that optogenetic stimulation "can inhibit PTH synthesis, vesicle transport and secretion processes in PTG cells." It is certainly possible that the changes in Rab8a and Rab11a might eventually lead to changes in vesicle transport and secretion processes, but the authors have not shown that. In order to claim changes in vesicle transport, the authors would need to measure vesicle transport in these cells, something that they did not do. So the authors need to soften their claim. They show that mRNA for genes involved in vesicle transport are suppressed in their levels. That is all that they should claim.
2. The authors show that transient suppression of PTH in mice with optogenetic suppression for 30 minutes every 2 days does lead to an increase in trabecular but not cortical bone mass. This is a clear study with a clear effect. In contrast, the new study comparing removal of a graft to cinacalcet treatment does not lead to a clear conclusion (graft removal but not cinacalcet led to an increase in bone mass). The authors do not really comment about why the effect of cinacalcet and graft removal on bone mass differ. Maybe this is interesting and maybe it's not. In any case, I don't see that experiment adding anything to this paper on the effects of optogenetic manipulation of PTH secretion. I don't think it adds anything regarding the possible effects of pulsatile PTH secretion. I would leave this new experiment out.

Reviewer #4:

Remarks to the Author:

NCOMMS-19-1125137C

Liu et al

"An Optogenetic Approach for Regulating Human Parathyroid Hormone Secretion"

The authors have adequately addressed my principal concerns in the latest revision.

I have a few minor comments.

1. It is not clear where the "prime" figure versions (Fig 1', etc) will appear. Are these data to be included as supplementary material, in the main manuscript, or were they provided solely to the reviewers as supporting information not intended for publication? Please clarify.
2. In figure 1', normal tissue controls should be included for comparison.
3. Although both the rat (diet-induced SHPT) and human (SHPT) parathyroid glands are clearly hyperplastic, this is not the same as demonstrating functional equivalence with regard to calcium sensing responsiveness. This caveat needs to be mentioned in the text.
4. The very informative discussion in the rebuttal letter of potentially divergent effects of PTH hypersecretion on trabecular vs cortical bone should be included in the manuscript discussion section.
5. With respect to the normalization of calcium levels described with regard to Figure 6i, it is not clear where the normalized data appear. The figure shows the actual calcium concentrations in mM, but does not display baseline-normalized values.
6. In Figure 6', one of the red arrows meant to denote non-infected cells with high PTH expression (upper panel series, red arrow in the lower left part of the image), appears to mark a ChETA-positive (green) cell that also expresses high levels of PTH. This suggests that ChETA expression is not on its own sufficient to suppress PTH production.

7. In Figure 9', the time course plot would be more informative if included a ChETA-positive cell so show the magnitude of the Fura-2 calcium flux change. Also, the timepoint at which the light stimulus is activated should be indicated on the x-axis.

8. The quality of the multichannel image overlays in Figure 10' is questionable. For example, the merged image panel of the control cells makes the PTH distribution appear sparse and non-uniform, while the single channel PTH channel for the same field appears uniform. The GFP signal intensity threshold may be too high to allow for meaningful assessment of colocalization with PTH. Can the authors provide a Manders score or similar colocalization metric to quantify the degree of overlap between the green and red color output channels?

9. Also related to this figure, how were positive cells defined for the graph? Was visual appearance the criterion? Do the authors have a quantitative definition of a positive cell (x-fold over background, something like that?).

Response to Reviewer #1

1. The authors add a new experiment that shows that optogenetic manipulation of parathyroid cells not only affects PTH secretion but also leads to a number of changes in gene expression. This is a point well worth making. But, unfortunately, the authors also claim that changes in the levels of Rab8a mRNA and Rab11a mRNA mean that optogenetic stimulation “can inhibit PTH synthesis, vesicle transport and secretion processes in PTG cells.” It is certainly possible that the changes in Rab8a and Rab11a might eventually lead to changes in vesicle transport and secretion processes, but the authors have not shown that. In order to claim changes in vesicle transport, the authors would need to measure vesicle transport in these cells, something that they did not do. So the authors need to soften their claim. They show that mRNA for genes involved in vesicle transport are suppressed in their levels. That is all that they should claim.

Response: Thanks for the comments. We agree that our data shows only that the mRNA levels for genes involved in vesicle transport are suppressed, so we have softened our claim: these data show that levels of mRNA for genes involved in PTH synthesis and vesicle transport are suppressed.

- On page 14, lines 291- 293 (Results):

These data show that levels of mRNA for genes involved in PTH synthesis and vesicle transport were suppressed in the optogenetic stimulated group.

2. The authors show that transient suppression of PTH in mice with optogenetic suppression for 30 minutes every 2 days does lead to an increase in trabecular but not cortical bone mass. This is a clear study with a clear effect. In contrast, the new study comparing removal of a graft to cinacalcet treatment does not lead to a clear conclusion (graft removal but not cinacalcet led to an increase in bone mass). The authors do not really comment about why the effect of cinacalcet and graft removal on bone mass differ. Maybe this is interesting and maybe it's not. In any case, I don't see that experiment adding anything to this paper on the effects of optogenetic manipulation of PTH secretion. I don't think it adds anything regarding the possible effects of pulsatile PTH secretion. I would leave this new experiment out.

Response: Thanks for the suggestion. As suggested, we have now removed this data and the related discussion paragraph comparing the removal of a graft to cinacalcet treatment from the manuscript.

Response to Reviewer #4

1. It is not clear where the "prime" figure versions (Fig 1', etc) will appear. Are these data to be included as supplementary material, in the main manuscript, or were they provided solely to the reviewers as supporting information not intended for publication? Please clarify.

Response: All 'prime' figures provided in the rebuttal letters (Figure 1', etc.) were solely for the reviewers as supporting information, and these data will not be included in the main manuscript. According to journal's transparency policy, all reviewer comments to authors and author rebuttal letters will be published together with the manuscript.

2. In figure 1', normal tissue controls should be included for comparison.

Response: Thanks for the suggestion. We now include normal tissue controls of human and rat for comparison (Figure 1').

Figure 1'. H&E staining of parathyroid glands of controls, patients and SHPT rats

3. Although both the rat (diet-induced SHPT) and human (SHPT) parathyroid glands are clearly hyperplastic, this is not the same as demonstrating functional equivalence with regard to calcium sensing responsiveness. This caveat needs to be mentioned in the text.

Response: Thank you and we agree. As suggested, we have now added the following information in the manuscript discussion.

- On page 27, lines 582- 584 (Discussion):

Although parathyroid glands from both diet-induced SHPT rats and human SHPT patients were hyperplastic, this does not necessarily mean that these two models are the equivalent with respect to functional calcium sensing responsiveness.

4. The very informative discussion in the rebuttal letter of potentially divergent effects of PTH hypersecretion on trabecular vs cortical bone should be included in the manuscript discussion section.

Response: Thanks for the suggestion. We have now added the discussion of potentially divergent effects of PTH hypersecretion on trabecular vs. cortical bone in the manuscript discussion.

- On page 29, lines 621- 636 (Discussion):

In our study, both trabecular and cortical bone were affected during the development of secondary hyperparathyroidism, which is consistent with previous work³⁰. However, after optogenetic inhibition of PTH secretion, we found an increase of BMD in trabecular bone, but not in cortical bone. This result is consistent with work by Ambrogini and colleagues who found that parathyroidectomy treatment induces recovery of BMD in trabecular bone prior to cortical bone³¹. It is possible that a longer course of optogenetic treatment may induce an increase in cortical bone mass.

Both human and animal studies investigating PTH injection have found both trabecular and cortical bone to be affected³²⁻³⁴. It is likely that cortical and trabecular bone both respond to changes in PTH, even though there may be latency between observable changes in cortical and trabecular bone. The large number of bone cells conjugated in trabecular bone allows easier maintenance of BMD under the effect of PTH in the early stage. In addition, cortical bone serves as a mineral reservoir³⁵, which would be primarily be utilized by PTH to serve calcium requirements. Long-term effects of optogenetic treatment on the recovery of cortical bone mass require careful study.

5. With respect to the normalization of calcium levels described with regard to Figure 6i, it is not clear where the normalized data appear. The figure shows the actual calcium concentrations in mM, but does not display baseline-normalized values.

Response: In the original version of the manuscript, we used the normalized calcium data (Figure 2' Left, below). After the first round of reviews, Reviewer #1 and Reviewer #2 suggested that we use absolute calcium values (Figure 2' right, below). Reviewer #2 required an explanation of the normalization process, which we provide below.

Figure 2'. Normalized calcium data in the original manuscript (left) and absolute calcium values in the latest manuscript (right)

The normalization process used in **figure 6i**:

$$\text{Normalized data} = \frac{\text{value of each eYFP or ChETA group (A0 or A5 or A15 or A30)}}{\text{average value of baseline B (eYFP or ChETA)}}$$

1) For eYFP (labeled light gray in the form below)

First, we calculated the average mean eYFP baseline (B): $(1.7371+1.7107+1.6737+1.4568+1.4751+1.4819)/6=1.5892$. To visualize the fold changes after light stimulation more clearly, we defined the normalized baseline as 1. Then, we divided all data with the average baseline mean (1.5892) to obtain normalized data:

$A0^{eYFP_1}=1.8321/1.5892=1.1595$;	$A0^{eYFP_2}=1.6384/1.5892=1.0369$;	$A0^{eYFP_3}=2.5673/1.5892=1.6249$; ...
$A5^{eYFP_1}=1.9359/1.5892=1.2253$;	$A5^{eYFP_2}=2.4613/1.5892=1.5578$;	$A5^{eYFP_3}=2.6203/1.5892=1.6584$; ...
$A15^{eYFP_1}=2.1593/1.5892=1.3667$;	$A15^{eYFP_2}=1.9883/1.5892=1.2584$;	$A15^{eYFP_3}=2.6113/1.5892=1.6528$; ...
$A30^{eYFP_1}=1.6296/1.5892=1.0314$;	$A30^{eYFP_2}=1.9854/1.5892=1.2566$;	$A30^{eYFP_3}=1.8081/1.5892=1.1444$; ...

2) For ETA (labeled light orange in the form below)

First, we calculated the average mean ChETA baseline (B): $(1.7065+1.7104+1.8684+1.6556+1.9478+1.6021)/6=1.7485$. To visualize the fold changes after light stimulation more clearly, we defined the normalized baseline as 1. Then, we divided all data with the average baseline mean (1.7485) to obtain normalized data:

$A0^{ChETA_1}=2.6477/1.7485=1.5129$;	$A0^{ChETA_2}=2.8428/1.7485=1.6338$;	$A0^{ChETA_3}=2.1674/1.7485=1.2456$; ...
$A5^{ChETA_1}=2.1885/1.7485=1.2506$;	$A5^{ChETA_2}=1.7348/1.7485=0.9969$;	$A5^{ChETA_3}=2.839/1.7485=1.6316$; ...
$A15^{ChETA_1}=1.8711/1.7485=1.0692$;	$A15^{ChETA_2}=1.5736/1.7485=0.9043$;	$A15^{ChETA_3}=1.4869/1.7485=0.8546$; ...
$A30^{ChETA_1}=1.8088/1.7485=1.0336$;	$A30^{ChETA_2}=1.6376/1.7485=0.9411$;	$A30^{ChETA_3}=1.4503/1.7485=0.8335$; ...

Figure 6i (raw data)

non-normalized Calcium data (relate to figure 6i)

	eYFP						ETA					
B	1.7371	1.7107	1.6737	1.4568	1.4751	1.4819	1.7065	1.7104	1.8684	1.6556	1.9478	1.6021
A0	1.8321	1.6384	2.5673	2.9009	2.5727	2.7363	2.6477	2.8428	2.1674	3.4043	3.2069	2.1849
A5	1.9359	2.4613	2.6203	2.9181	2.6136	2.6881	2.1885	1.7348	2.839	2.965	2.3127	3.178
A15	2.1593	1.9883	2.6113	2.7331	2.2389	2.0666	1.8711	1.5736	1.4869	1.7133	1.7124	1.5708
A30	1.6296	1.9854	1.8081	1.8279	2.3244	2.246	1.8088	1.6376	1.4503	1.8133	1.7403	1.6969

Normalized Calcium data (relate to figure 6i)

	eYFP						ETA					
B	1	1	1	1	1	1	1	1	1	1	1	1
A0	1.1595	1.0369	1.6249	1.8361	1.6283	1.7318	1.5129	1.6338	1.2456	1.9565	1.8431	1.2557
A5	1.2253	1.5578	1.6584	1.8469	1.6542	1.7013	1.2506	0.9969	1.6316	1.704	1.3291	1.8265
A15	1.3667	1.2584	1.6528	1.7298	1.4171	1.3079	1.0692	0.9043	0.8546	0.9846	0.9841	0.9028
A30	1.0314	1.2566	1.1444	1.1569	1.4712	1.4215	1.0336	0.9411	0.8335	1.0422	1.0002	0.9753

6. In Figure 6', one of the red arrows meant to denote non-infected cells with high PTH expression (upper panel series, red arrow in the lower left part of the image), appears to mark a ChETA-positive (green) cell that also expresses high levels of PTH. This suggests that ChETA expression is not on its own sufficient to suppress PTH production.

Response: Thanks for the careful examination. Indeed, there is a ChETA positive cell showing high PTH expression in Figure 6' of our previous rebuttal letter. This phenomenon may explain why the ChETA transfection rate is almost 50%, whereas the inhibition effect of PTH secretion is only about 39% - not all ChETA positive cells have their PTH secretion completely inhibited. Moreover, other cell types infected by the lentivirus may also contribute to the observed inhibitory effects. We have incorporated this information into the revised discussion.

- On page 28, lines 618- 620 (Discussion):

It is noteworthy that the contribution of pulsatile PTH secretion to the observed effects is unclear and other cell types infected by the lentivirus may contribute the observed effects.

7. In Figure 9', the time course plot would be more informative if included a ChETA-positive cell so show the magnitude of the Fura-2 calcium flux change. Also, the timepoint at which the light stimulus is activated should be indicated on the x-axis.

Response: We now include ChETA⁺ cell (figure 3'b) to show the magnitude of the Fura-2 calcium flux change between uninfected and ChETA cells.

Figure 3'. a, b, Quantification of calcium fluorescence signals with time in uninfected and ChETA-expressing cells. Blue lines indicate the timepoint of blue light stimulation. c. Arrows show an uninfected cell; arrow heads show a ChETA-expressing cell.

8. The quality of the multichannel image overlays in Figure 10' is questionable. For example, the merged image panel of the control cells makes the PTH distribution appear sparse and non-uniform, while the single channel PTH channel for the same field appears uniform. The GFP signal intensity threshold may be too high to allow for meaningful assessment of colocalization with PTH. Can the authors provide a Manders score or similar colocalization metric to quantify the degree of overlap between the green and red color output channels?

Response: Thanks for the comments. We carried out a colocalization analysis (see below). Firstly, it can be observed from the image in low magnification (Figure 4', taken from two other independent experiments) that the rat parathyroid gland tissue is very condensed with and the majority of cells are parathyroid gland cells except for rare connective tissues and endothelial cells. The PTH signal appears uniform, and a clear boundary could be seen between the parathyroid and the surrounding thyroid tissue.

Figure 4' PTH signals in thyroid and parathyroid glands. The picture on the right is an enlarged view of the white box area in the picture on the left

Secondly, when we look at the PTH channel alone in high magnification, we find both 'light spots' and lots of 'round sites' with low intensity. When we overlap the PTH and DAPI image (Figure 5'), we found that the light spots indicate the overexpression of the PTH which are mainly surrounding the DAPI signals. The 'round sites' of low intensity are always be covered by DAPI signals within the merged image, which is actually taken as background signal in this image. When the 'round sites' are covered by nuclei (DAPI) in the merged image, the surrounding PTH overexpression signals would look rather sparse, which might explain the observed discrepancy between single channel and merged image.

Figure 5' PTH signals in parathyroid glands

To precisely colocalize different signals (PTH, GFP, DAPI) in one cell (in cytoplasm or in nucleus or both) and avoid the observing discrepancy, we first used the nucleus signal to define the cell location instead of the 'one-to-one pixel matching' method, then we calculated the mean fluorescence intensity of PTH and GFP of each cell within single channel to quantify the degree of overlap between red and green color. We have included this colocalization metric in Figure 6' and Figure 7'.

Figure 6' Cell Number definition

Cell No.	PTH							48	Cell No.	GFP						27 Sum	Double	27	Double/PTH
	Area	Mean	Min	Max	Sum	PTH+	Area			Mean	Min	Max	Sum	GFP+	Double				
1	466	25.83	3	241		1	1	495	4.805	1	10								
2	456	25.634	5	233		1	2	341	6.328	0	253								
3	732	25.587	5	252		1	3	396	7.671	0	243								
4	612	17.121	5	254		0	4	328	5.547	0	238								
5	638	31.428	10	254		1	5	436	7.645	0	253								
6	506	23.986	4	226		1	6	315	2.721	0	10								
7	540	24.031	4	245		1	7	774	25.879	6	247								
8	556	22.149	8	255		1	8	311	3.48	0	93								
9	711	25.146	5	249		1	9	341	73.116	33	241								
10	540	27.446	4	255		1	10	430	19.189	0	255								
11	640	32.936	3	242		1	11	640	20.538	1	223								
12	424	31.481	1	252		1	12	288	46.931	5	245								
13	470	28.032	5	239		1	13	454	32.333	5	255								
14	400	33.815	7	244		1	14	472	100.788	67	245								
15	508	39.073	11	244		1	15	359	99.925	80	238								
16	472	25.334	7	241		1	16	298	85.701	80	241								
17	532	38.573	11	254		1	17	274	96.894	54	240								
18	528	31.258	6	244		1	18	535	50.062	5	255								
19	577	35.821	6	237		1	19	330	38.833	1	238								
20	568	42.806	11	238		1	20	315	74.219	14	243								
21	508	40.77	11	241		1	21	200	79.48	50	85								
22	489	32.716	12	235		1	22	428	42.369	5	217								
23	638	29.723	3	251		1	23	260	21.346	0	247								
24	472	20.148	11	37		1	24	268	63.433	46	253								
25	568	33.498	9	253		1	25	341	62.578	10	231								
26	344	52.814	10	253		1	26	340	74.415	17	244								
27	421	43.622	6	232		1	27	300	93.347	75	246								
28	470	36.36	10	242		1	28	207	73.952	47	211								
29	356	48.365	8	229		1	29	199	101.613	74	231								
30	542	38.345	11	230		1	30	207	7.676	0	237								
31	420	28.888	6	239		1	31	366	14.25	0	252								
32	420	28.888	6	239		1	32	356	2.141	0	185								
33	820	17.278	8	238		0	33	264	6.132	0	282								
34	326	34.825	2	253		1	34	420	4.124	0	216								
35	436	27.569	0	245		1	35	416	6.86	0	218								
36	364	26.085	10	235		1	36	268	29.943	0	242								
37	472	40.191	8	238		1	37	346	5.957	0	254								
38	430	41.986	5	253		1	38	447	8.97	0	220								
39	414	45.37	10	244		1	39	176	7.507	0	253								
40	389	35.319	10	236		1	40	315	14.898	0	234								
41	366	40.967	8	235		1	41	376	7.486	0	234								
42	356	33.298	8	236		1	42	366	4.962	0	251								
43	384	22.255	5	208		1	43	508	5.304	0	213								
44	262	22.5	7	254		1	44	302	1.085	0	5								
45	356	39.14	4	255		1	45	298	15.115	0	228								
46	447	24.163	11	223		1	46	430	5.084	0	228								
47	599	38.668	10	254		1	47	238	4.867	0	254								
48	688	22.648	6	217		1	48	400	7.227	0	255								
49	774	32.853	6	244		1	49	469	3.908	0	221								
50	648	49.381	8	234		1	50	648	5.5	0	247								

Figure 7' Calculation of signal intensity in single channel

9. Also related to this figure, how were positive cells defined for the graph? Was visual appearance the criterion? Do the authors have a quantitative definition of a positive cell (x-fold over background, something like that?).

Response: Quantification criteria: All cells were defined and located based on the nucleus (DAPI) and cell size was defined based on the distance between the center of neighboring nuclei (Figure 8'). Only cells with an intact nucleus were classified as a cell. When there were either positive or negative GFP/PTH signals within cell areas, cells were counted as 1 or 0, respectively.

Figure 8' Examples of cells defined by DAPI signals.

For GFP or PTH channel signals, five background areas were circled on each image and signal intensities for each were analyzed using ImageJ software. The average and standard deviation of the mean signals were calculated for the image. The threshold was set above the average+2*standard deviation (std) (Figure 9'). The background of the PTH channel was defined as having 'weak round sites', the background of the GFP channel was defined by the control group without virus expression.

Figure 9' Threshold calculation

We then calculated the average signal intensity of each cell. When the average intensity was higher than the background threshold, the cell was classified as positive and counted as '1' during the counting of this channel. Alternatively, '0' was counted when the average signal intensity was lower than the threshold (Figure 10'). Double positive cells were counted only when the cell was positive in both channels.

Figure 10' Positive and Negative definition

Reviewers' Comments:

Reviewer #4:

Remarks to the Author:

The authors have satisfactorily addressed the issues raised in my most recent review.

Response to Reviewer #4

1. The authors have satisfactorily addressed the issues raised in my most recent review.

Response: Thank you so much for all the constructive comments and suggestions. We are glad that our revision addressed your concerns.